# Macrophages form dendrite-like pseudopods to enhance bacterial ingestion

Changyuan Fan[1,2,3], Xinyi Huang [iD][1,2], Jie Mei[1,2,3], Xuemeng Shi[2], Hao Zhang[4], Cong Liang [iD][4], Shuzhi Cui[2], Yifan Xing[2], Biao Cao[5], Wei Liu[6], Huisheng Liu[7], Bo Liu [iD][1,2,3], Wakam Chang[8], Mengle Shao[1,2,3], Gong-Hong Wei[9], Yan-Jun Liu[6], Zheng-Jun Chen[4], Zhaoyu Lin [iD][10✉], Tao Xu[3,7✉] & Yaming Jiu [iD][1,2,3✉]

## Abstract

Macrophages are critical innate immune cells that exhibit remarkable adaptability during pathogen infections. However, the relationship between their morphological plasticity and physiological functions remains largely elusive. Here, we discovered an unprecedented paradigm of macrophage adaptation within a few hours upon severe Gram-negative bacterial infections, characterized by the formation of dendrite-like pseudopods (DLPs). Using in vitro, microfluidic, and in vivo infection models, we demonstrate that these pseudopods enhance bacterial uptake by expanding the macrophage searching radius, thereby bolstering host defense. Mechanistically, Toll-like receptor 4 (TLR4) activation by Gram-negative bacterial lipopolysaccharide (LPS) upregulates the expression of macrophage-specific RhoGEF and ARHGEF3 in an NF-κB-dependent manner. ARHGEF3 localizes to dendrite-like pseudopods and enhances RhoA activity. Consequently, periodic cycles of actin assembly and disassembly propel the elongation of pseudopods, whereas vimentin intermediate filaments stabilize them. Importantly, infusion of DLP-equipped macrophages into Salmonella-infected mice reduced bacterial burden and infection severity. Together, our findings underscore how the dynamic response of macrophages to massive infections can augment immune defense against pathogenic bacteria.

Keywords Macrophage; Dendrite-like Pseudopods; Pathogen Ingestion; Actin Filaments; Vimentin Intermediate Filaments
Subject Categories Cell Adhesion, Polarity & Cytoskeleton; Immunology; Microbiology, Virology & Host Pathogen Interaction

## Introduction

Gram-negative bacteria pose a significant pathogenic threat due to their general antimicrobial resistance, leading to severe diseases and symptoms such as sepsis (Alexandraki and Palacio, 2010; Corsini Campioli et al, 2022; Zabawa et al, 2016). Macrophages serve as frontline sentinels against Gram-negative bacterial infections by deploying phagocytosis and macropinocytosis to eliminate pathogens (Li et al, 2023).

Notably, stimulation with lipopolysaccharide (LPS), a hallmark component of the outer membrane of Gram-negative bacteria, enhances the ability of macrophages to engulf pathogens (Freeman et al, 2018; Yamamoto et al, 2020). Despite understanding macrophage engagement with pathogens, the specific adaptations that macrophages undergo in response to massive Gram-negative bacterial infections remain poorly characterized.

There are implicit correlations between the morphological plasticity and physiological functions of macrophages. For instance, microglia, macrophages in the nervous system, develop dendritic structures for immunological surveillance and apoptotic cell clearance (Nayak et al, 2014). Upon encountering pathogenic bacteria, macrophages undergo substantial deformations driven by rapid cytoskeletal rearrangements (Groves et al, 2008). Actin cytoskeletal-based filopodia and lamellipodia form to capture bacteria within minutes post-infection, using a 'hook-and-shovel' strategy (Davidson and Wood, 2020; Möller et al, 2013). Phagocytic cups then gradually form, envelop, and engulf the bacteria (Krause et al, 2004; Vorselen et al, 2020). Additionally, activated macrophages display high levels of constitutive macropinocytosis (Alpuche-Aranda et al, 1994; Bosedasgupta and Pieters, 2014), in which bacteria are passively internalized along with extracellular fluid, forming macropinosomes. However, these morphological remodeling primarily engage proximal targets through sequential engulfment in a one-at-a-time strategy or engulf them passively.

[1]Shanghai Institute of Materia Medica, Chinese Academy of Sciences, 201203 Shanghai, China. [2]Shanghai Institute of Immunity and Infection, Chinese Academy of Sciences, 200031 Shanghai, China. [3]University of Chinese Academy of Sciences, Yuquan Road No. 19(A), Shijingshan District, 100049 Beijing, China. [4]State Key Laboratory of Cell Biology, Shanghai Institute of Biochemistry and Cell Biology, Center for Excellence in Molecular Cell Science, Chinese Academy of Sciences, 200031 Shanghai, China. [5]State Key Laboratory of Experimental Hematology, Shanghai Institute of Nutrition and Health, Chinese Academy of Sciences, 200031 Shanghai, China. [6]Shanghai Xuhui Central Hospital, Zhongshan-Xuhui Hospital, Shanghai Key Laboratory of Medical Epigenetics, Shanghai Stomatological Hospital, Institutes of Biomedical Sciences, Department of Chemistry, Fudan University, 200032 Shanghai, China. [7]Guangzhou Laboratory, 510005 Guangzhou, China. [8]Department of Biomedical Sciences, Faculty of Health Sciences, University of Macau, Taipa, 999074 Macau, China. [9]MOE Key Laboratory of Metabolism and Molecular Medicine & Department of Biochemistry and Molecular Biology of School of Basic Medical Sciences, and Fudan University Shanghai Cancer Center, Cancer Institutes, Department of Oncology, Shanghai Medical College of Fudan University, 200032 Shanghai, China. [10]State Key Laboratory of Pharmaceutical Biotechnology, Ministry of Education Key Laboratory of Model Animal for Disease Study, Jiangsu Key Laboratory of Molecular Medicine, Model Animal Research Center, National Resource Center for Mutant Mice of China, Nanjing Drum Tower Hospital, School of Medicine, Nanjing University, 210061 Nanjing, China. ✉E-mail: Linzy@nju.edu.cn; xu_tao@gzlab.ac.cn; ymjiu@siii.cas.cn

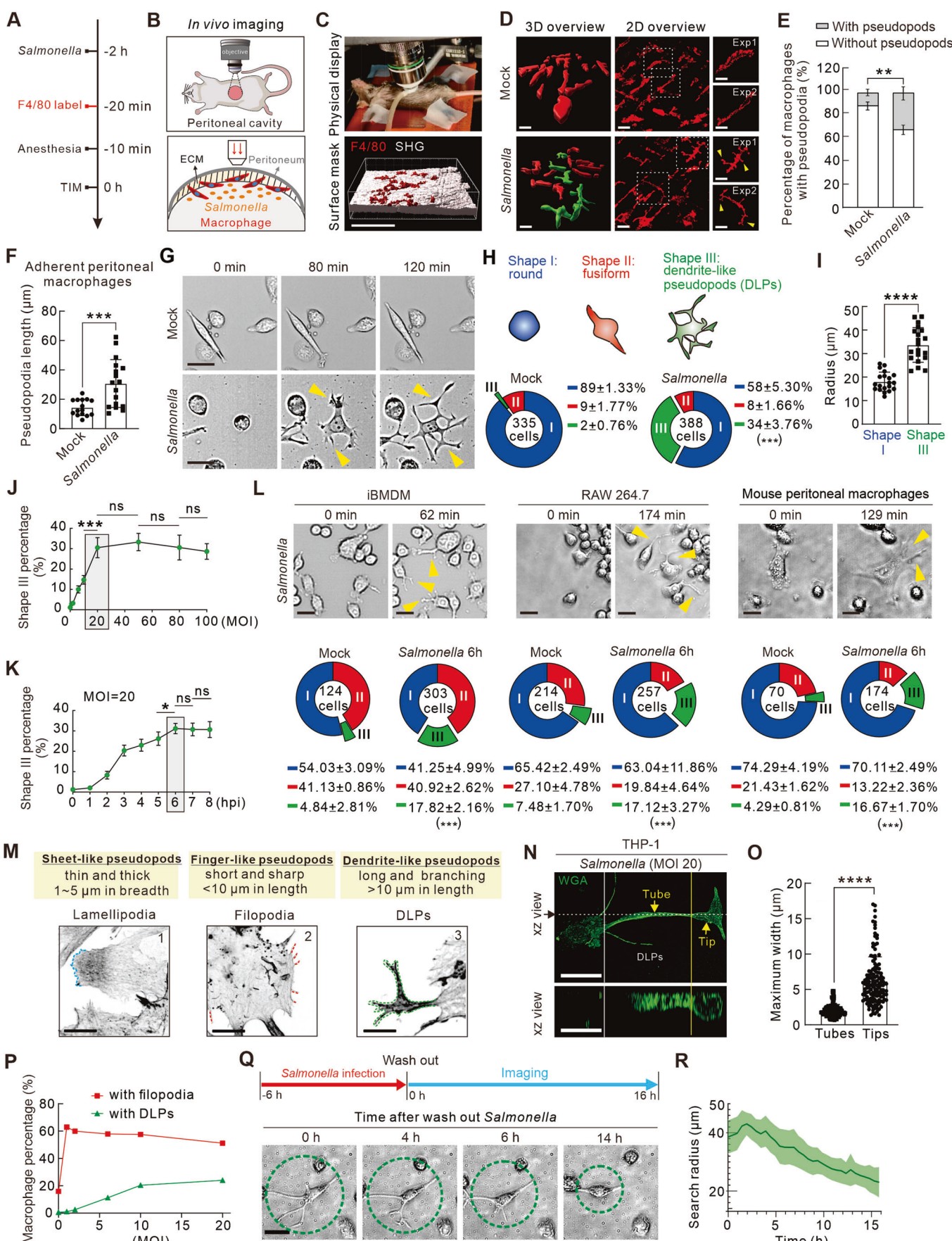

◄ **Figure 1.  Macrophages form DLPs in response to a bacterial challenge.**

(A) Schematic diagram of the two-photo intravital microscopy (TIM) experimental procedure. (B) Illustration of imaging the abdominal pouch in live mouse with TIM. (C) Physical display of TIM device and 3D reconstruction of macrophages on the abdominal wall. Scale bars, 20 μm. (D) Representative TIM imaging of peritoneal macrophages labeled with anti-F4/80 in vivo with or without *Salmonella* challenge. Green cells represent macrophages with obvious pseudopods. Magnified views reveal individual macrophage examples. Scale bars, 10 μm. (E) Quantification of the percentage of macrophages with or without DLPs upon *Salmonella* infection, based on TIM data from 150 cells in 6 independent imaging views from three experiments. (F) Quantification of the DLPs length in adherent peritoneal macrophages with or without *Salmonella* challenge. $n = 18$ cells. (G) Time-lapse imaging of THP-1 macrophages revealing the process of DLPs formation with or without *Salmonella* infection (MOI = 20). Scale bars, 50 μm. (H) Pie chart quantification of the percentage of three shapes of macrophages with or without *Salmonella* infection (MOI = 20). Blue, round shape (I); Red, fusiform shape (II); green, deformed shape with DLPs (III). (I) Quantification of the search radius of shape I and III macrophages. $n = 50$ cells. (J, K) Dosage and temporal-dependent analysis of shape III macrophages at 6 h-post-infection (hpi) under varying MOIs (J) or at different hpi times with a fixed MOI of 20 (K). (L) Representative time-lapse imaging revealing DLPs formation in iBMDM, RAW264.7 and mouse peritoneal macrophages during *Salmonella* infection (MOI = 20). Scale bars, 50 μm. Pie chart quantification of the percentage of three shapes of macrophages with or without *Salmonella* infection (MOI = 20) for 6 h. (M) Immunofluorescence visualization by phalloidin staining, delineating lamellipodia, filopodia and DLPs with dashed lines. Upper panels provide definitions for the three protrusive structures. Scale bars, 10 μm. (N) Representative images of DLPs visualized by WGA staining in *Salmonella* infected THP-1 macrophages. Yellow arrows depict 'tube' and 'tip' structures. Yellow line divides the tube and tip. White line divides the cell body and DLPs. Lower panel shows the orthographic view from the white dash line in the upper panel. Scale bars, 20 μm. (O) Quantification of the maximum width of 'tips' and 'tubes' in (n). $n = 75$ cells. (P) Quantification of the percentage of macrophages with filopodia and DLPs, respectively, in the indicated MOI of *Salmonella* infection at 6 hpi in THP-1 cells. (Q) Upper panel showing the schematic diagram of *Salmonella* withdraw experiment. Lower panel showing the time-lapse imaging of THP-1 macrophages after washing out *Salmonella*. Green circles depict the radius of cell. Scale bars, 20 μm. (R) Quantification of search radius of shape III macrophages after washing out *Salmonella*. The yellow arrowheads depict the DLPs in (D, G, L). Data are presented as mean ± s.d. from three independent experiments. Dots in quantifications presented individual cells. $P = 0.00980$ (E). $P = 0.00090$ (F). $P = 0.00090$ (H). $P = 4.26191E-10$ (I). P values from left to right (J): $P = 0.00700$, $P = 0.50150$, $P = 0.56538$, $P = 0.66530$. P values from left to right (K): $P = 0.02162$, $P = 0.86055$, $P = 0.97863$. P values from left to right (L): $P = 0.00034$, $P = 0.00030$, $P = 0.00026$. $P = 5.32443E-40$ (O). ns $P > 0.05$; *$P < 0.05$; **$P < 0.01$; ***$P < 0.001$; ****$P < 0.0001$: unpaired two-tailed Student's *t* test (E, F, H, I, J, K, L, O). Source data are available online for this figure.

Whether macrophages possess other morphological strategies for the ingestion of distal pathogens to combat massive bacterial infections remains elusive.

Rho GTPase family (Rac1, Cdc42, and RhoA) regulates morphogenesis and remodeling of actin-based structures under various extracellular and intracellular cues, including pathogen stimulation. Rac1 drives Arp2/3-mediated actin polymerization to form lamellipodia and large ruffles for macropinocytosis (Egami et al, 2014), while Cdc42 regulates filopodia formation, and RhoA controls the development of actin stress fibers. All three GTPases converge in phagocytic cups to mediate generalized bacterial engulfment (Anes, 2017; Ernst et al, 1999). Interestingly, while the role of RhoA in infection-induced macrophage remodeling remains unclear, LPS specifically activates RhoA in bone marrow-derived macrophages (BMDMs) (Yang et al, 2014). This LPS-RhoA axis is further amplified in microglia through upregulation of ARHGEF3, an immune- and neuron-restricted guanine nucleotide exchange factor (RhoGEF) (Arthur et al, 2002; Liao et al, 2021). However, whether the mechanistic link between RhoA and its corresponding RhoGEF promotes morphological adaptation of macrophages to Gram-negative pathogens remains elusive.

In this study, we identified dendrite-like pseudopods (DLPs) as macrophage-specific adaptation paradigm for high-efficiency capture and engulfment of Gram-negative bacteria during massive bacterial infections. Unlike classical protrusions to limited proximal targets, DLP operates as dynamic, long-range machinery through coordinated actin bundles and vimentin intermediate filaments, enabling macrophages to ensnare distal pathogens. Mechanistically, LPS activates the TLR4-NF-κB pathway to induce ARHGEF3 expression, which drives RhoA-dependent DLPs formation. Infusion of DLPs-equipped macrophages into bacteria-infected mice significantly boosted bacterial clearance in vivo. Together, our findings highlight a previously overlooked but crucial morphological adaptation of macrophages that enhances pathogen elimination and immune defense capabilities.

## Results

### Macrophages generate dendrite-like pseudopods (DLPs) in response to severe bacterial infections

To elucidate the morphological responses of macrophages to bacterial stimuli, we established a non-invasive intravital imaging model by intraperitoneally challenging mice with *Salmonella*. Peritoneal macrophages were labeled by the mouse macrophage-specific marker, EGF-like module-containing mucin-like hormone receptor-like 1 (EMR1, F4/80) (Fig. 1A) (Dos Anjos Cassado, 2017). Utilizing two-photon microscopy, we observed the morphological responses of these macrophages under basal conditions and following infection (Fig. 1A–C; Movie EV1). In the absence of infection, most macrophages attached to the mesothelium appeared to be embedded in the extracellular matrix, exhibiting a sleek and slender morphology (Fig. 1D). In contrast, *Salmonella* infection increased the proportion of macrophages with elongated and branched pseudopods (Fig. 1D–F), indicating a distinct morphological reconfiguration of macrophages in response to bacterial stimulation in vivo.

To further characterize these deformed macrophages, we established an in vitro visualization pipeline by exposing immortalized THP-1 monocyte-derived macrophages to *Salmonella* infection. Consistent with in vivo observations, macrophages underwent shape transformations, characterized by the emergence of the dendrite-like pseudopods (DLPs) with distinctive bean-shaped distal ends (Fig. 1G; Movie EV2). We categorized these morphological changes into three distinct shapes: round macrophages (shape I), fusiform macrophages (shape II), and DLPs-equipped macrophages (shape III) (Fig. 1H). Under the mock infection condition, the proportions of these shapes were 89%, 9%, and 2%, respectively (Fig. 1H). Upon *Salmonella* infection at a multiplicity of infection (MOI) of 20, the proportion shifted significantly to 58%, 8%, and 34% at 6 h post-infection (hpi) (Fig. 1H). Notably, we

found that ~90% DLPs emerged from type I round, un-polarized macrophages, while ~10% DLPs emerged from type II fusiformed macrophages (Fig. EV1A). The proportion of shape II macrophages remained relatively stable, while the most pronounced changes were a decrease in round macrophages (shape I) and a substantial increase in DLPs-equipped macrophages (shape III).

We further quantitatively characterized the morphological alterations of DLPs-equipped macrophages (shape III) and observed a significant increase in cell radius and area, along with a decrease in circularity and solidity (Figs. 1I and EV1B–D). Notably, the average number of primary pseudopods extending directly from the cell body was consistent across different *Salmonella* MOIs, averaging approximately four per macrophage (Fig. EV1E). To validate these changes, we employed Sholl analysis, a method usually used to assess neuronal dendrite complexity (Binley et al, 2014), which supported our observations by demonstrating an increased number of intersections between the cell periphery and concentric circles (Fig. EV1F).

To determine whether the proportion of DLPs-equipped macrophages is influenced by the bacterial MOI and/or the duration of infection, we conducted a quantitative analysis using *Salmonella* as a model pathogen. The results revealed a dose-dependent increase in the proportion of DLPs-equipped macrophages with *Salmonella* MOI of up to 20, after which the proportion plateaued at ~30% (Fig. 1J). At a constant *Salmonella* MOI of 20, the proportion of DLPs-equipped macrophages increased in a time-dependent manner, peaking at ~30% at 6 hpi (Fig. 1K). These results suggest that the DLPs formation correlates with the severity of infection, reflecting an early cellular response to pathogenic challenge.

To investigate whether non-pathogenic extracellular materials elicit a similar response, we generated cellular debris from Caco2 colonic epithelial cells and exposed macrophages to them. Despite being engulfed (Li et al, 2019), the debris did not induce pseudopod formation (Fig. EV1G). Dendritic cells (DCs), which are named for classical protrusion structures, exhibited shorter and less branched pseudopods upon *Salmonella* infection, appearing different from DLPs in macrophages (Fig. EV1H–J). To assess the prevalence of the DLPs formation, we repeated infection experiments using immortalized murine bone marrow-derived macrophages (iBMDM), RAW264.7, and primary mouse peritoneal macrophages. *Salmonella* consistently triggered a visibly distinct DLP-positive macrophage subpopulation after persistent infection in all tested macrophages (Fig. 1L), which were not observed in other cell types, such as cancer cells (Fig. EV1K). These results indicate that the DLPs formation is a specific phenomenon in macrophage responses to Gram-negative bacteria stimuli.

Filopodia and lamellipodia are typical protrusive structures in macrophages used to capturing pathogens (Krause et al, 2004). We characterized and compared these structures with the DLPs identified, noting that, unlike the finger-like or sheet-like protrusions of filopodia and lamellipodia, DLPs exhibited a more elongated, branched, and tentacle-like topology (Fig. 1M). Additionally, DLPs often exhibited non-uniform width, with slender tubular 'tubes' and enlarged 'tips', further emphasizing their tentacle-like morphology (Fig. 1N,O). Different from filopodia, which are known for their frequent and rapid retraction, DLPs demonstrated a more persistent but slower growth pattern (Fig. EV1L,M). Especially, at 6 hpi, filopodia were fully stimulated

in ~60% of macrophages at an MOI below 1, whereas the DLPs required an MOI of 3 or higher to be triggered, peaking at a proportion of ~30% at an MOI of 20 (Fig. 1P). This threshold difference underscores the specificity of DLPs as a macrophage response to severe bacterial infection. Furthermore, DLPs were with enriched focal adhesions compared to cell body (Fig. EV1N,O) indicating its ability to anchor to the matrix. Microenvironment adhesiveness may contribute to DLPs formation. We also observed that the DLPs can recede within 14 h following bacterial clearance (Fig. 1Q,R), indicating that their formation is a reversible response to pathogen infection.

Together, these findings suggest that the DLPs observed are specialized morphological adaptations of macrophages in response to pathogenic bacterial stimuli. After an extensive literature search and to the best of our knowledge, the DLPs in macrophages, which are 2–3 times longer than the cell body and feature several multi-level branches, have not been thoroughly studied previously.

## Contact with bacteria is required for directional DLPs formation

To clarify the directional specificity of the DLPs, we designed a microfluidic system to controllably direct bacterial interactions with macrophages. This system comprises three parallel main chambers linked perpendicularly by an array of 10 μm-wide microchannels, designed to permit the passage of *Salmonella* (~1–2 μm), while excluding THP-1 macrophages (~30 μm) (Fig. 2A). To validate the unidirectional diffusion of substances, a fluorescence tracer, FITC-Dextran, was applied. Fluorescence intensity measurements confirmed a gradient extending into the central chamber over the observation period (Fig. EV1P). *Salmonella* and THP-1 macrophages were seeded into the upper and middle chambers, respectively, under the premise of unidirectional bacterial engagement from above. Intriguingly, the DLPs consistently extended towards the bacteria-laden microchannels with minimal deviation angles of less than 30 degrees, indicative of a directed growth response (Fig. 2B–E). In contrast, macrophage cell bodies occasionally occupied the lower microchannels (Fig. 2D,E; Movie EV3), suggesting a spatial preference aligned with the presence of bacteria. We quantified the proportion of shapes to macrophages near the junction between the microchannels and the middle chamber, DLP-positive macrophages were significantly more prevalent near the junction facing the infection (Fig. 2F). Particularly, we found that in response to bacterial infection, most DLPs extended toward the upper microchannel. In non-infected cases, the few DLPs observed rarely extended toward the lower microchannel (Fig. 2G). the morphology of the DLPs within the microchannel closely resembled that observed in conventional in vitro assays, both in shape and length (Fig. 2F,H), underscoring the physiological relevance of our findings.

To elucidate the necessity of direct bacterial contact for DLPs formation, we employed a transwell-based contact assay. THP-1 macrophages were plated in the lower chamber and bacteria were added to the upper chamber. The interaction between macrophages and bacteria was modulated using transwell membranes with distinct pore sizes (Fig. 2I). Notably, the use of a 0.4 μm pore size membrane, which precludes bacterial penetration, abolished the DLPs formation (Fig. 2J,K). In contrast, membranes with 8 μm pores facilitated bacterial diffusion, leading to robust DLPs development (Fig. 2J,K). These results revealed the directional nature of DLPs induction by bacterial infection and highlighted the

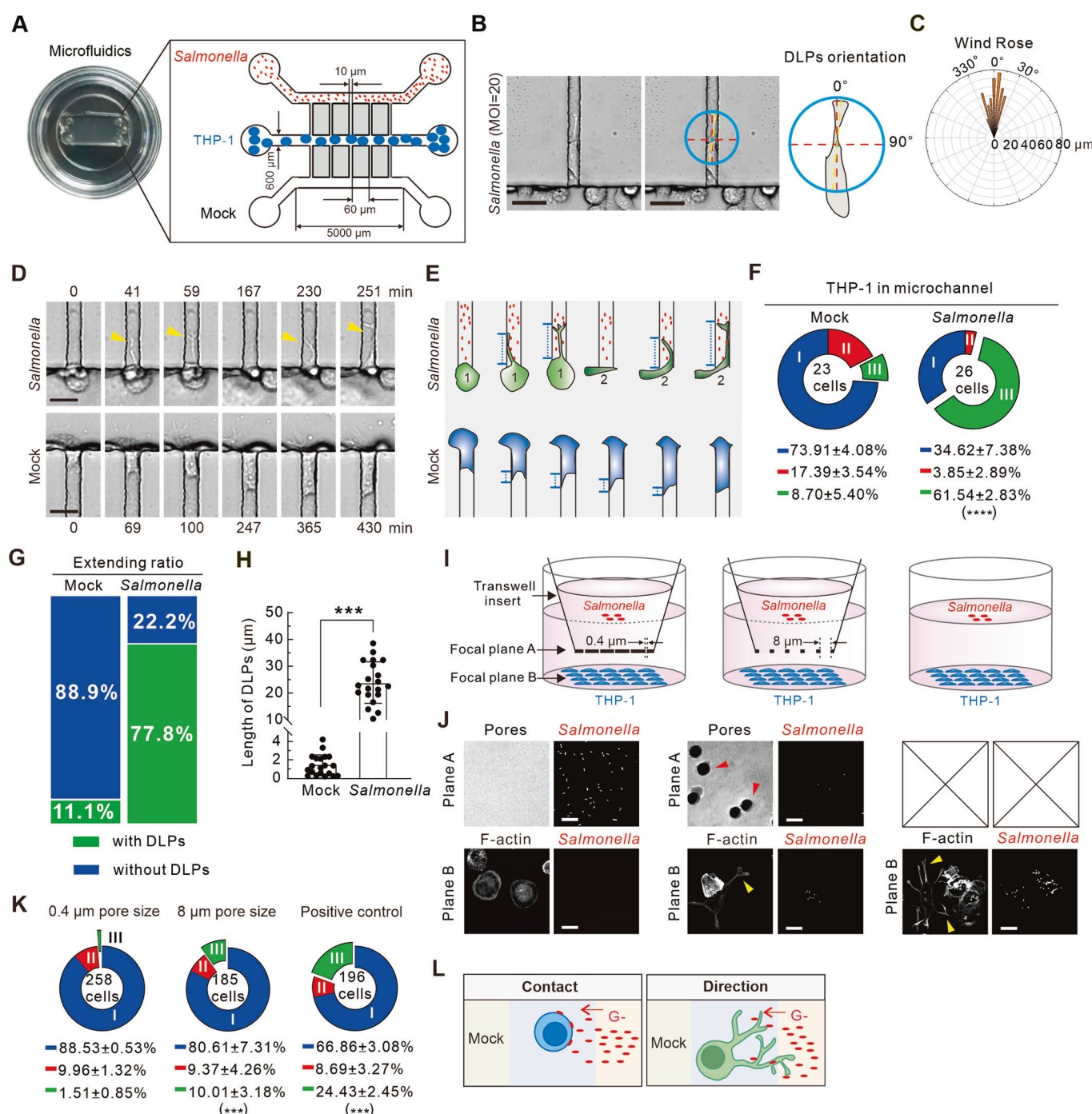

necessity of direct contact between macrophages and bacteria (Fig. 2L). This finding further indicates that intrinsic bacterial components, rather than their secreted factors, are essential prerequisites for the DLPs formation.

## Cyclic actin assembly waves drive the DLPs construction and vimentin intermediate filaments stabilize them

Drastic changes in cell morphology often involve cytoskeletal rearrangements (Fletcher and Mullins, 2010). Immunofluorescence

analysis revealed a higher concentration of actin bundles and vimentin, the primary intermediate filament (IF) (Peuhu et al, 2017), in DLPs than in the cell body, while tubulin exhibited a relatively uniform distribution across both regions (Fig. 3A,B).

To investigate the role of the actin cytoskeleton in DLPs morphogenesis, we conducted real-time imaging of actin dynamics to uncover a novel stepwise cyclic growth pattern of extending DLPs (Fig. 3C). This pattern was delineated into three stages (Figs. 3C,D and EV2A; Movie EV4): (1) *the growing stage*, characterized by rapid actin assembly into bundles to fill a just-

**Figure 2. DLPs formation requires the contact with bacteria and they elongate directionally towards bacteria.**

(A) Photo image and schematic diagram of the microfluidics. (B) Representative bright field images of shape III macrophages within microfluidics. The blue circle depicts the DLPs. The yellow dashed line depicts the length of DLPs. The red line depicts the vertical axis. The angle between the red line and the yellow line is the DLPs orientation. (C) Wind rose analysis depicting the directional orientation of 20 DLPs. (D) Representative bright field time-lapse imaging of THP-1 macrophages in microfluidics infected with *Salmonella* (MOI = 20) from upper channel. Scale bars, 20 μm. (E) Schematic diagram showing the observed macrophage deformation in microfluidics corresponding in (D). (F) Pie chart quantification of the percentage of three shapes of macrophages located in the junction between the upper vertical microchannel (*Salmonella* infection for 6 h)/the lower vertical microchannels (Mock) and the middle horizontal chamber. Blue, round shape (I); Red, fusiform shape (II); Green, deformed shape with DLPs (III). (G) Percentage of extending ratio of DLP-positive macrophages to microchannels. (H) Quantification of the DLPs length in Mock and *Salmonella* infected macrophage in the microfluidics assay. *n* = 20 cells. (I) Schematic diagram showing the optimized transwell model for the contact assay. (J) Representative images showing the membrane pores (depicted by red arrowheads) in bright field, GFP-tagged bacterial in upper A and lower B focal planes, and macrophages morphology with phalloidin staining. Scale bars, 20 μm. (K) Pie chart quantifications of the percentage of three shapes in macrophages in the contact assay. (L) Schematic diagram illustrating that the directional growth of DLPs require the contact with bacteria. The yellow arrowheads depict the DLPs in (D, J). Data are presented as mean ± s.d. from three independent experiments. *P* = 0.00024 (F). *P* = 2.04912E-15 (H). *P* values from left to right (K): *P* = 0.00216, *P* = 0.00019. ***P* < 0.001: unpaired two-tailed Student's *t* test (F, H). one-way ANOVA with Sidak's analysis (K). Source data are available online for this figure.

grown nascent DLP, where DLP length remains almost unchanged while F-actin intensity gradually increases; (2) *the steady stage*, the longest period (over half of the cycle) during which both the protrusive structure and the internal actin bundles remained relatively stable; and (3) *the collapsing stage*, during which the DLPs extended further, accompanied by rapid disassembly of actin bundles indicated by decreased F-actin intensity. A new cycle commenced with rapid actin re-assembly, culminating in the development of DLPs, thus highlighting a cyclic actin assembly-disassembly mechanism essential for DLPs morphogenesis.

To further elucidate the cytoskeleton-associated mechanisms of DLPs formation, we first applied various inhibitors to modulate actin dynamics before and during *Salmonella* infection (Fig. 3E). These inhibitors include Latrunculin B, which induces overall actin disassembly (Diring et al, 2019); Blebbistatin, which disrupts actomyosin bundles by inhibiting myosin ATPase activity (Zhang et al, 2017); SMIFH2, which inhibits linear actin nucleation and polymerization by targeting formin (Nishimura et al, 2021); NP-G2-044, which inhibits actomyosin bundles by targeting the crosslinker fascin (Wang et al, 2021); and CK666, which reduces actin filament branching by inhibiting the Arp2/3 nucleator complex (Henson et al, 2015). We verified the optimal drug concentrations for cell viability and bacterial proliferation (Fig. EV2B,C), and their effects on the actin network through immunofluorescence (Fig. EV2D). Treatment with Blebbistatin, SMIFH2, and NP-G2-044 resulted in the disappearance of actin bundles; LatrunculinB treatment caused actin filaments disorder; CK666 treatment led to the complete disappearance of podosomes, highly ordered structures composed of branched actin (Henson et al, 2015) (Fig. EV2D). Interestingly, we found that the disruption of linear actin bundles, rather than branched actin, impedes DLPs formation, implicating the essentiality of myosin, formin, and fascin (Fig. 3F). Furthermore, we explored the involvement of microtubules and IFs in DLPs formation. Treatment with nocodazole, which destabilizes the microtubules, had negligible effects on DLPs (Fig. EV2E–H). CRISPR/Cas9-mediated vimentin knockout (VIM KO) in THP-1 macrophages resulted in an inability to form DLPs-equipped macrophages, characterized instead by the continuous initiation of bleb-like 'growth cones' (Fig. 3G–I). Similarly, simvastatin, an inhibitor targeting the vimentin network (Trogden et al, 2018), failed to induce DLPs formation (Fig. EV2I–L). Collectively, these findings suggest that assembly and disassembly of bundled actin and vimentin are crucial for DLPs morphogenesis.

Vimentin IFs function in many cellular morphogenesis processes by working synergistically with actin network (Fletcher and Mullins, 2010). Considering the high abundance of vimentin in DLPs, we examined the actin dynamics in VIM KO THP-1 macrophages. Despite the absence of successful DLPs formation, 'budding' protrusions were sequentially formed and retracted from distinct regions in the cell body, accompanied by actin bundle assembly and disassembly (Fig. 3J; Movie EV5). These budding cycles in VIM KO macrophages were similar to those in wild-type THP-1 cells but lacked sustained DLPs elongation (Figs. 3I–K and EV2M). Collectively, these results indicate that vimentin IFs are essential for stabilization and elongation of the DLPs. The synergistic interplay between actin and vimentin IFs is critical for DLPs morphogenesis.

## LPS-activated TLR4 signaling stimulates the DLPs formation

Extensions of macrophages under innate immune stimulation have been previously reported (Williams and Ridley, 2000). However, there is no investigation up to date on the formation requirements and the underlying mechanisms of the DLPs we discovered, which are much longer than cell body and with several multi-level branches. To ascertain the ubiquity and specificity of DLPs, we exposed THP-1 macrophages to a spectrum of bacterial strains. Notably, Gram-negative bacteria, including *Salmonella*, *E. coli*, and *Shigella*, consistently induced DLPs, in which the morphology of the DLPs remained consistent, a response not observed with Gram-positive bacteria such as *Bacillus* and *Listeria* (Fig. 4A), suggesting that there are certain commonalities among Gram-negative bacteria in promoting DLPs formation.

To dissect the underlying mechanisms of Gram-negative bacteria triggered DLPs, we employed lipopolysaccharide (LPS), a key component of the outer membrane of Gram-negative bacteria (Galanos et al, 1984; Galanos et al, 1985; Maldonado et al, 2016), and lipoteichoic acid (LTA), a major cell wall constituent of Gram-positive bacteria (Siegel et al, 2016). We observed the DLPs formation only with LPS treatment, but not with LTA (Fig. 4B; Movie EV6). Both the proportion and morphological characteristics of shape III macrophages (DLPs-equipped) stimulated by LPS (~38%, Fig. 4C–G) were comparable to those observed with *Salmonella* infection (~34%; Figs. 1G–I and EV1A–C).

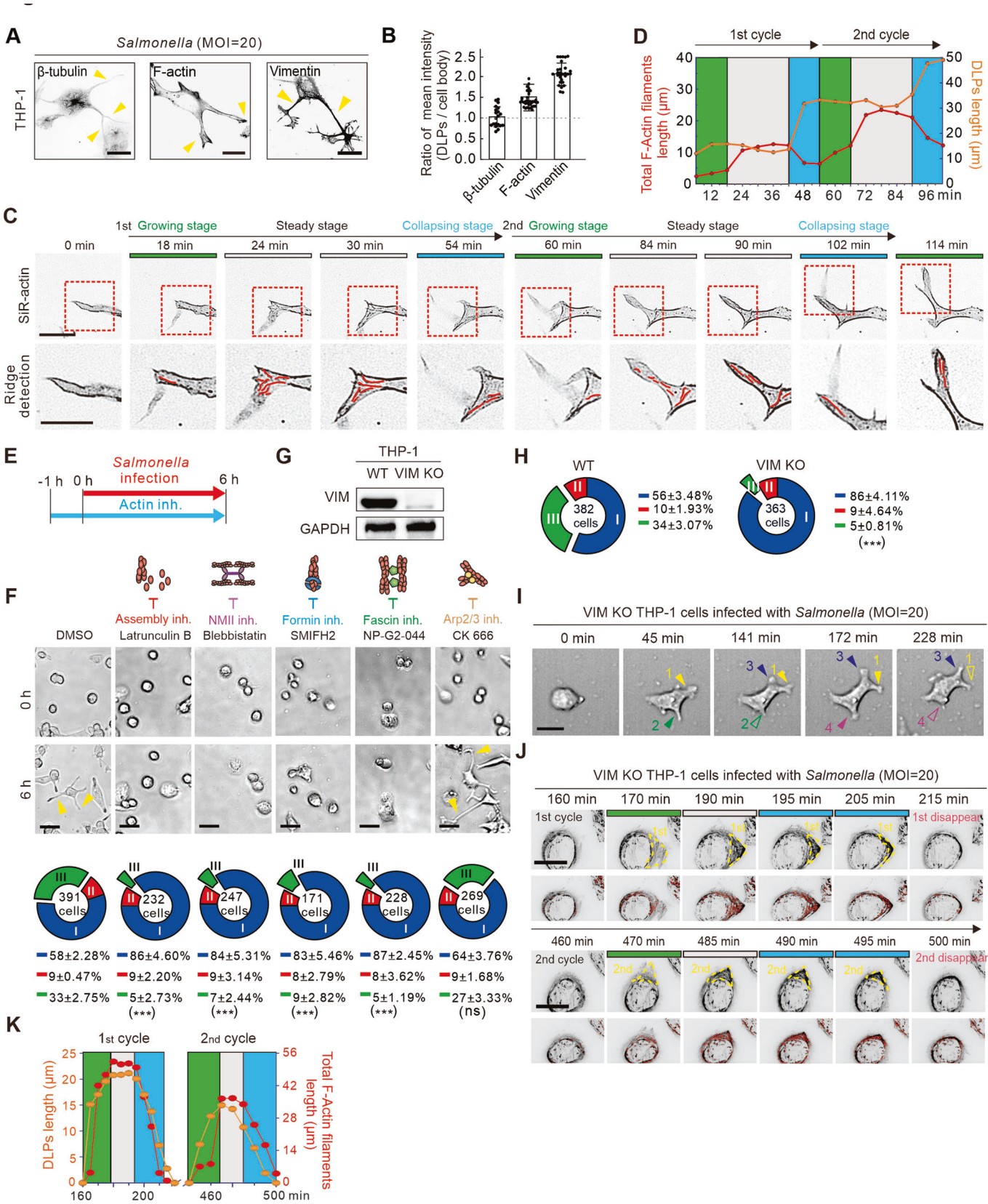

**Figure 3.  Cyclic actin assembly waves drive the DLPs construction and vimentin intermediate filaments stabilize the DLPs.**

(A) Immunostaining of endogenous β-tubulin, F-actin and vimentin in *Salmonella* infected THP-1 macrophages. Scale bar, 20 μm. (B) Quantification of the fluorescence intensity ratio of β-tubulin, F-actin and vimentin in DLPs versus the cell body in *Salmonella* infected THP-1 macrophages. n = 20 cells. (C) Representative time-lapse images of SiR-actin labeled actin filaments in *Salmonella* infected THP-1 macrophages (MOI = 20). Scale bar, 20 μm. Magnified views showing the DLPs. Red lines depicting the actin filaments detected by Ridged detection. Scale bar, 20 μm. (D) Quantification of the DLPs length and actin filaments length over time in infected THP-1 macrophages in (C). Growing, steady and collapsing stages are denoted by green, gray and blue, respectively. (E) Schematic diagram of actin inhibitors treatment assay. (F) Upper panels showing time-lapse imaging of actin inhibitors treated THP-1 macrophages during *Salmonella* infection (MOI = 20) for 6 h. The targeting sites of inhibitors are shown in the upper panels. Pie charts in the lower panels show the percentage of macrophage shapes. Scale bar, 20 μm. DMSO, n = 391 cells; Latrunculin B treatment, n = 232 cells; Blebbistatin treatment, n = 247 cells; SMIFH2 treatment, n = 171 cells; NP-G2-044 treatment, n = 228 cells; and CK666 treatment, n = 269 cells. (G) Western blot analysis of vimentin in wild-type (WT) and vimentin knockout (VIM KO) THP-1 macrophages. (H) Pie charts showing the percentage of three shapes in macrophages upon *Salmonella* infection for 6 h. WT, n = 382 cells; VIM KO, n = 363 cells. (I) Time-lapse images of VIM KO THP-1 macrophages during *Salmonella* infection (MOI = 20). Arrowheads in distinct colors depict the unstable DLPs. Scale bars, 20 μm. (J) Time-lapse images of SiR-actin labeled actin filaments in *Salmonella* infected VIM KO THP-1 macrophages (MOI = 20). Yellow dashed lines outline the budding DLPs. (K) Quantification of the DLPs length and actin filaments length over time in infected VIM KO THP-1 macrophage in (J). Growing, steady and collapsing stage are denoted by green, gray and pink, respectively. The yellow arrowheads depict the DLPs in (A, F). Data are presented as mean ± s.d. from three independent experiments. Dots in quantifications presented individual cells. *P* values from left to right (F): *P* = 0.00052, *P* = 0.00053, *P* = 0.00102, *P* = 0.00018, *P* = 0.08446. *P* = 0.00022 (H). ns *P* > 0.05; ***P* < 0.001: unpaired two-tailed Student's *t* test (H). one-way ANOVA with Sidak's analysis (F). Source data are available online for this figure.

Additionally, LPS sensitively stimulated the DLPs formation in both iBMDMs and RAW264.7 macrophages (Fig. 4H,I), consistent with the response in THP-1 cells (Fig. 4B) and the response to *Salmonella* infection (Fig. 1L).

Toll-like receptor 4 (TLR4) is a canonical LPS receptor on macrophages that mediates the activation of downstream signaling cascades, such as NF-κB signaling (Lu et al, 2008). TLR4 activation was confirmed by increased nuclear localization and phosphorylation of p65, indicative of NF-κB signaling downstream of TLR4 (Zhou et al, 2022) (Fig. EV3A,B). Notably, TLR4 knockout (TLR4 KO) THP-1 cells generated by CRISPR-Cas9 method showed a significant inhibition of the DLPs formation (Fig. 4J–L). The TLR4 inhibitor TAK-242 effectively inhibited the LPS-induced activation of NF-κB signaling (Fig. 4M). THP-1 macrophages were pretreated with TAK-242 at a concentration that spared cell viability and bacterial proliferation prior to and during *Salmonella* infection (Fig. EV3C,D). This intervention significantly attenuated the DLPs formation (Fig. 4N,O). Together, these findings suggest that LPS-activated TLR4 signaling is crucial for orchestrating the DLPs construction in response to Gram-negative bacterial infection (Fig. 4P).

## Omics profiling reveals the relevance of ARHGEF3-RhoA in the DLPs formation

To uncover the molecules that participate in LPS-activated TLR4 signaling and subsequent DLPs formation, we conducted a strategy to isolate and characterize P- (macrophages with minimal DLPs) and P+ (macrophages with excessive DLPs) cell colonies, followed by RNA sequencing to identify differential gene expression patterns (Fig. 5A). Considering the restricted proliferative capacity of differentiated THP-1 macrophages, we selected iBMDM (immortalized murine bone marrow-derived macrophages enabling unlimited proliferation) for screening. Two parallel cell populations (#1 and #2) derived from the same batch of iBMDM macrophages were used. Parallel #1 underwent LPS stimulation to confirm the reproducibility of DLPs formation in this batch of cells. Once validated, parallel #2 was subjected to monoclonal cell seeding in a 96-well plate, followed by LPS stimulation to screen for P+ and P− colonies based on morphological criteria. The specific criteria used to identify P− vs. P+ colonies were based on the percentage of cells

that were adapted to the type III morphology after LPS stimulation. More than 40% of type III cells were considered P + , while less than 10% of type III cells were considered P−. The specific clones selected for RNA sequencing represented the most extreme phenotypes within these groups, with the sequenced P+ clone showing ~52% type III cells and the P- clone showing only ~1%. Colonies with significant variance in their capacity for DLPs production were identified as 19 P− colonies with minimal DLPs and 5 P+ colonies with robust DLPs formation.

RNA sequencing was subsequently performed on P− and P+ macrophage colonies under both LPS-stimulated and non-stimulated conditions (Fig. 5A). The significant upregulation of genes associated with innate immune responses confirmed the sensitivity of these macrophages to LPS stimulation (Fig. EV4A). Differential gene expression analysis revealed that the P+ population uniquely upregulated 464 genes following LPS stimulation, whereas the P− population upregulated 217 genes (Fig. 5B). Additionally, 330 genes were significantly upregulated in both cell types (Fig. 5B).

Further profiling revealed that genes specifically upregulated in P+ macrophages appeared to be sensitive to immune stimuli (Fig. 5C). Whereas, P− cells showed minimal upregulation of classical pro-inflammatory cytokines, such as IL-1β, upon LPS stimulation (Fig. EV4B). Moreover, surface marker analysis revealed that characteristics of M1 macrophages (CD64, CD32) are significantly upregulated in P+ cells upon LPS stimulation, but not in P− cells (Fig. 5D). We also validated the expression of some surface markers in P + / P− cells with LPS stimulation. The quantitative results of immunofluorescence were consistent with the transcriptomic profiling results, showing a significant increase in the expression of related surface markers (CD64, CD32, CD14) in P+ cells compared to P− cells (Fig. EV4C–E).

Strikingly, our transcriptomic analysis revealed a significant enrichment of cytoskeletal components, particularly genes associated with the actin network, among those upregulated in P+ macrophages in response to LPS stimulation (Fig. 5E). This observation directed our focus toward the potential regulation of actin dynamics in P+ cells. Rho GTPases are central regulators of dynamic cytoskeletal assembly (Burridge and Guilluy, 2016; Kim et al, 2018), which are controlled by multidomain guanine nucleotide exchange factors (RhoGEFs) and GTPase-activating

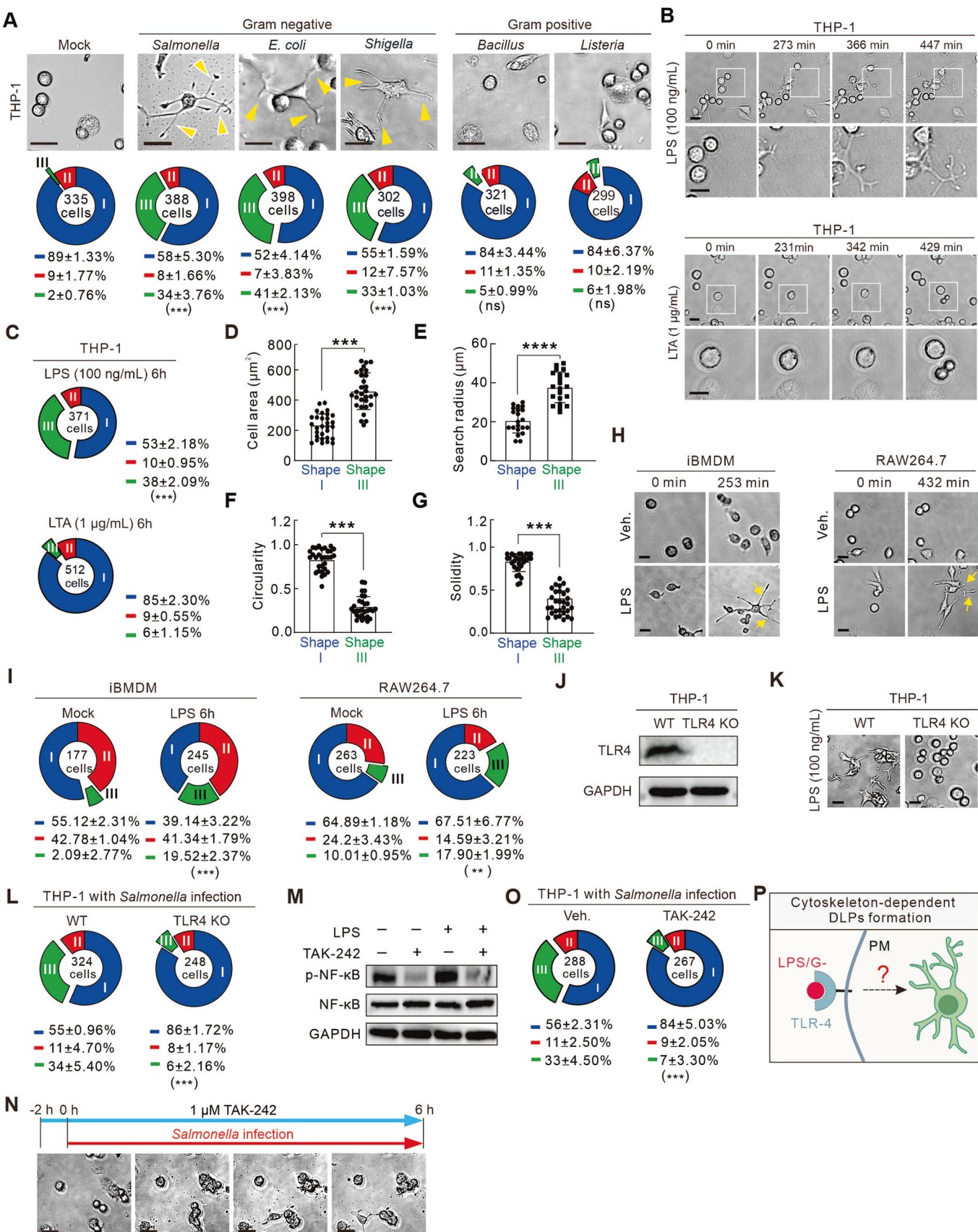

**Figure 4. LPS activated TLR4 stimulates the DLPs formation.**

(A) Representative phase-contrast imaging of THP-1 macrophages following infection with distinct bacteria at 6 hpi (MOI = 20). Scale bars, 50 μm. Pie chart in the lower panel showing the quantification of the percentage of three shapes of macrophages with different bacterial infection (MOI = 20) for 6 h. Yellow arrows depict the DLPs. (B) Time-lapse imaging of THP-1 macrophages showing the DLPs formation upon treatment of 100 ng/mL LPS and 1 μg/mL LTA, respectively. Lower panels are magnified regions. Scale bar, 20 μm (in cell images) and 2 μm (in the magnified images). (C) Pie charts representing the percentage of three shapes in macrophages treated with LPS and LTA for 6 h, respectively. (D–G) Quantification of the cell area (D), searching radius (E), circularity (F) and solidity (G) of shape I and shape III macrophages. $n = 25$-30 cells. (H) Representative time-lapse images of iBMDMs and RAW264.7 with or without LPS stimulation. Yellow arrows depict the DLPs. Scale bar, 20 μm. (I) Pie chart quantification of the percentage of iBMDMs and RAW264.7 treated with LPS for 6 h. (J) Western blot analysis of TLR4 in wild-type (WT) and TLR4 knockout (TLR4 KO) THP-1 macrophages. (K) Representative imaging of THP-1 WT and TLR4 KO macrophage with LPS stimulation. Scale bars, 20 μm. (L) Pie charts showing the percentage of three shapes in macrophages in wild-type (WT) and TLR4 knockout (TLR4 KO) THP-1 macrophages infected with *salmonella* for 6 h. WT, $n = 324$ cells; TLR4 KO, $n = 248$ cells. (M) Western blot analysis of NF-κB (Ser536) phosphorylation following LPS, TAK-242 and combined LPS and TAK-242 treatment. (N) Schematic diagram (upper panel) and representative time-lapse imaging (lower panels) of THP-1 macrophage during *Salmonella* infection (MOI = 20) under TAK-242 treatment. Scale bars, 20 μm. (O) Pie charts showing the percentage of three shapes in macrophages following vehicle (Veh.) and TAK-242 treatment. Veh., $n = 288$ cells; TAK-242 treatment, $n = 267$ cells. (P) Schematic diagram of LPS-triggered DLPs formation. Data are presented as mean ± s.d. from three independent experiments. Dots in quantifications presented individual cells. $P$ values from left to right (A): $P = 0.00029$, $P = 6.77908E-05$, $P = 4.29608E-05$, $P = 0.07380$, $P = 0.08446$. $P = 0.00006$ (C). $P = 4.66028E-12$ (D). $P = 1.86775E-24$ (E). $P = 4.80168E-23$ (F). $P = 1.78717E-09$ (G). $P$ values from left to right (I): $P = 4.41414E-05$, $P = 0.01250$. $P = 0.00313$ (L). $P = 0.00645$ (O). ns $P > 0.05$; **$P < 0.01$; ***$P < 0.001$; ****$P < 0.0001$: unpaired two-tailed Student's $t$ test (C, D–G, I, O). one-way ANOVA with Sidak's analysis (A). Source data are available online for this figure.

proteins (RhoGAPs) (Müller et al, 2020). Due to the enrichment of parallel actin bundles in DLPs, we thus sorted out all emerging Rho GEFs and GAPs from the transcriptomics data, among which ARHGEF3 increased most significantly in P+ macrophages (Fig. 5F). This alteration was further confirmed by qRT-PCR and Western blotting at the transcriptional and translational levels (Figs. 5G,H and EV4F–H; ARHGEF11, ARHGAP18, and ARHGAP10 were used as negative controls). Notably, ARHGEF3 is specifically abundant in macrophages, with moderate expression in platelets and neuronal tissues (Arthur et al, 2002; Khaliq et al, 2021). Consistent with iBMDM, LPS stimulation induced a time-dependent increase in ARHGEF3 levels in THP-1 macrophages (Fig. 5I), a response not found by LTA challenge (Fig. EV4I). ARHGEF3 is known to selectively activate RhoA, but not RhoC, RhoG, Rac1 or Cdc42 (Khaliq et al, 2021), suggesting that RhoA-regulated actin assembly and disassembly are critical for DLPs formation. This was further supported by the delayed upregulation of GTP-bound active RhoA following LPS stimulation, which paralleled the increase in the ARHGEF3 levels (Fig. 5I).

To elucidate the mechanisms underlying the upregulation of ARHGEF3, we utilized JASPAR (jaspar.elixir.no) to predict transcription factor binding at the ARHGEF3 locus. Our analysis revealed a predicted NF-κB binding site in the ARHGEF3 promoter, a well-characterized downstream transcription factor of TLR4 signaling (Fig. EV4J). Further validation using a chromatin immunoprecipitation (ChIP) assay and a luciferase reporter system confirmed that ARHGEF3 transcription is modulated by NF-κB in response to LPS stimulation (Figs. 5J and EV4K,L). This upregulation was attenuated by application of the TLR4 inhibitor (Fig. 5K). Pyrrolidine dithiocarbamate (PDTC), a recognized NF-κB antagonist (Chuang et al, 2009), was utilized to assess the role of NF-κB in ARHGEF3 expression, with an effective but non-toxic concentration (Fig. EV4M). Western blot analysis demonstrated diminished ARHGEF3 expression under LPS-PDTC co-treatment relative to LPS treatment alone (Fig. EV4N), suggesting that NF-κB is the key factor through which LPS-TLR4 boosts ARHGEF3 expression. Consistent with these findings, the TLR4 inhibitor predominantly decreased both the protein level of ARHGEF3 and the active level of RhoA stimulated by LPS (Fig. 5K).

To verify the necessity of ARHGEF3 for DLPs formation, a small interference RNA-mediated knockdown assay was performed in THP-1 macrophages (Fig. EV4O). The proportion of DLPs was dramatically decreased upon knockdown of ARHGEF3 (Fig. 5L), but not upon ARHGEF11 and ARHGAP10 knockdown (Fig. EV4P). These results further highlight the unique role of ARHGEF3 in regulating DLPs formation. RhoA and ROCK are important downstream effector molecules of ARHGEF3, directly influencing the assembly of actin bundles (Fig. 5M). We have transfected the constitutively active RhoA (CA, L63) and the dominant negative RhoA (DN, N19) (Jeon et al, 2010) transiently in THP-1 macrophages to regulate the activity of RhoA, which boosts and compromises the DLPs formation, respectively (Fig. 5N). Consistently, inhibition of RhoA using Y-27632 confirmed the dependency of DLPs formation on RhoA activity (Figs. 5O and EV4Q–S). This effect was comparable to that achieved inhibiting TLR4 pathway (Fig. 4L,O).

As a RhoGEF, ARHGEF3 exhibits a distinct cellular spatial distribution, primarily localized to the cell membrane and nucleus (Singh et al, 2021). Previous biochemical studies revealed that the plasma membrane targeting by interaction with PI(4,5)P$_2$ is necessary for ARHGEF3 to achieve its maximum GEF activity (Singh et al, 2021). It is worth mentioning that TLR4 signaling induction necessitates the delivery of TLR4 to PIP$_2$-rich domains of the plasma membrane (Kagan and Medzhitov, 2006). This prompted us to investigate whether TLR4 is involved in ARHGEF3 distribution. We stably expressed HA-tagged TLR4 (TLR4-HA) in iBMDMs and confirmed its interaction with the endogenous ARHGEF3 via reciprocal co-immunoprecipitations (Fig. 5P). Interestingly, ARHGEF3 was recruited to DLPs rather than to the cell body in LPS-stimulated THP-1 macrophages (Fig. 5Q,R). Furthermore, inhibition of TLR4 activity by TAK242 obviously decreased ARHGEF3 intensity in the regions where the initial grow cones of DLPs locate. Consistently, we observed similar ARHGEF3 and TLR4 co-localization in the DLPs of iBMDMs and RAW264.7 macrophages (Fig. EV4T,U). In addition, the co-localization of ARHGEF3 and TLR4 was enriched in DLPs with a puncta pattern, commonly observed at the tips or branching points of DLPs, rather than in the cell body (Figs. 5Q and EV4V,W). Thus, it was evident

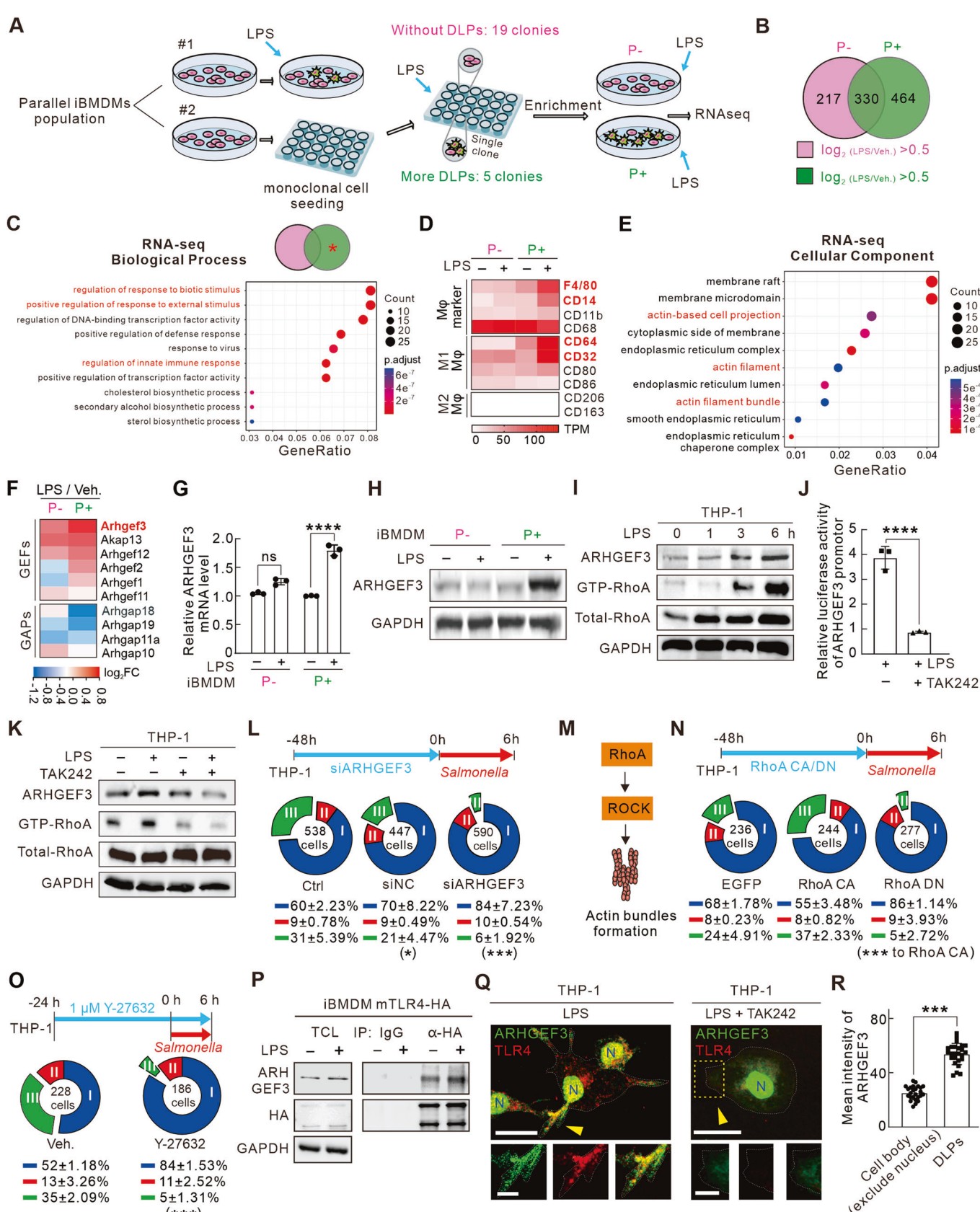

**Figure 5.  Omics profiling reveals critical roles of ARHGEF3 and RhoA in DLPs formation.**

(A) Schematic diagram showing the screening and sorting of none-DLPs (P−) and more-DLPs (P + ) iBMDM macrophages. (B) Illustration of upregulated genes in P− and P+ iBMDMs with LPS treatment. (C) Biological process enrichment analysis of upregulated genes in P+ but not in P− iBMDMs with LPS treatment. The top ten enriched terms are shown in the bubble chart. (D) Heatmap showing the transcriptional levels of surface markers of macrophages with LPS stimulation in P+ and P− iBMDMs by RNA sequencing. (E) Cellular component enrichment analysis of upregulated genes in P+ but not in P− iBMDMs. The top 10 enriched items are shown in the bubble chart. In (C) and (E), the colored bubbles displayed from red to blue indicating the descending order of P. adjust value. The sizes of the bubbles are displayed from small to large in ascending order of gene counts. The x and y axis represent the gene ratio and the GO terms, respectively. (F) Heatmap showing the transcriptional levels of RhoA GEFs and GAPs by RNA sequencing. (G) ARHGEF3 transcriptional levels were measured by RT-qPCR in P− and P+ iBMDM macrophages. (H) Western blot analysis of ARHGEF3 protein levels upon LPS stimulation in P− and P+ iBMDM macrophages. (I) Western blot analysis of ARHGEF3 and the active levels of RhoA in THP-1 macrophages treated with LPS for indicated times. (J) Quantification of the relative luciferase activity upon LPS or LPS + TAK242 treatment. (K) Western blot analysis of ARHGEF3 and the active levels of RhoA upon LPS, TAK-242 or combined LPS and TAK-242 treatment. (L) Upper panel showing the diagram of ARHGEF3 knockdown procedure in THP-1 macrophages. Pie charts in lower panels illustrating the percentage of three shapes with RNAi treatment and salmonella infection for 6 h. Ctrl (no transfection), n = 758; siNC, n = 384 cells; siARHGEF3, n = 590 cells. (M) Diagram of the RhoA-ROCK signaling pathway. (N) Upper panel showing the diagram of RhoA CA/DN plasmid transfection procedure in THP-1 macrophages. Pie charts in lower panels illustrating the percentage of three shapes with transfection and salmonella infection for 6 h. EGFP (Control), n = 236; RhoA CA, n = 244 cells; RhoA DN, n = 277 cells. (O) Upper panel showing the diagram of Y-27632 treatment procedure. Pie charts in lower panels illustrating the percentage of three shapes with or without Y-27632 treatment and with Salmonella infection for 6 h. Veh., n = 228 cells; Y-27632 treatment, n = 186 cells. (P) Co-immunoprecipitation (Co-IP) verified the interaction between mTLR4 and ARHGEF3 in iBMDM cells stably expressed mTLR4-HA. The protein complex was enriched using Protein A/G agarose beads. (Q) Representative images of THP-1 macrophages with DLPs stimulated by LPS without or with TAK242 pretreatment and stained with ARHGEF3 and TLR4 antibodies. White dash lines depict the outline of the cell. Cropped views depict DLPs regions. Yellow arrows depict the DLPs. Bars, 20 μm (in cell images) and 5 μm (in the magnified images). (R) Quantification of the mean intensity of ARHGEF3 in cell body (except nucleus) and the DLPs of THP-1 macrophages treated with LPS. Data are presented as mean ± s.d. from three independent experiments. Dots in quantifications presented individual cells. P values from left to right (G): P = 0.28437, P = 0.00013. P = 0.00035 (J). P values from left to right (L): P = 0.01918, P = 0.00050. P = 0.00057 (N), P = 7.67308E-05 (O). P = 3.63749E-20 (R). ns P > 0.05; ***P < 0.001; ****P < 0.0001: unpaired two-tailed Student's t test (G, J, O, R), one-way ANOVA with Sidak's analysis (L, N), two-way ANOVA with Sidak's analysis (F), Benjamini–Hochberg (BH) analysis (C, E). Source data are available online for this figure.

that the signals of ARHGEF3 and TLR4 exhibited consistent patterns and were stronger in DLPs, indicating that TLR4 also guides the spatial localization of ARHGEF3, thereby promoting DLPs formation.

## DLPs enhance the efficiency of bacterial ingestion by macrophages and promote the clearance of bacteria in vivo

Macrophages, as sentinels of the innate immune system, are tasked with capturing and eliminating pathogens (Freeman et al, 2018; Yamamoto et al, 2020). To investigate the role of DLPs in pathogen ingestion, we used a fluorescently labeled virulent strain of Salmonella. Notably, fluorescence labeling did not compromise the infectivity of Salmonella or the proportion of DLPs formation in THP-1 cells (Fig. EV5A,B). Three-dimensional imaging analysis revealed that bacteria were frequently found within the regions occupied by DLPs (Fig. 6A). Further, live-cell imaging provided detailed bacterial capture process by DLPs (Fig. 6B; Movie EV7). Initially, tips of DLPs expanded as the bacteria approached, potentially maximizing contact. Subsequently, bacteria were phagocytosed from the tips of DLPs. Instead of immediate retraction, DLPs enabled the retrograde transport of the bacterium towards the cell body. By meticulously tracking the movement of individual Salmonella within DLPs, we determined that the average transport speed was ~0.12 μm/min (Fig. EV5C), aligning with the kinetics of actin-dependent trafficking (Ponti et al, 2004). Moreover, we observed a progressive decrease in the Salmonella fluorescence signal during its retrograde transportation within DLPs (Fig. 6B), a phenomenon that typically accompanies bacterial engulfment and degradation (Jiang et al, 2021). As expected, bacteria in DLPs and the cell body were found within vacuoles that are positive for the lysosome-specific marker LAMP1 (Fig. 6C) (Nguyen and Yates, 2021), suggesting that the bacteria, regardless of their subcellular location, are targeted for a similar lysosomal

degradation destination. This process is distinct from the rapid retraction of filopodial structures, which do not internalize bacteria but instead pull them to the cell body surface for subsequent phagocytosis (Krause et al, 2004) and is distinct from the pseudopodia in the process of macropinocytosis because of no large vacuoles visible inside macrophages and most bacteria engulfed by lysosomes.

Defining the role of DLPs in bacterial capture necessitated the identification of an optimal time window for measuring the intracellular bacterial load. By quantifying colony-forming units (CFU), we observed a significant increase in bacterial count from 6 to 12 hpi, followed by a decrease from 12 to 24 hpi (Fig. EV5D,E). This pattern corresponds to the early replication phase and subsequent degradation of Salmonella in macrophages, aligning with previously reported in vivo Salmonella replication (Hoffman et al, 2021). Therefore, we measured the bacterial load at 6 hpi, a critical pre-replication juncture. Our findings revealed that shape III DLPs-equipped macrophages, internalized twice the number of bacteria compared to shape I macrophages at distinct MOIs (Fig. 6D–G). To verify these findings in a more physiological context, we examined the bacterial load in both human peripheral blood mononuclear cell (PBMC)-derived macrophages and murine bone marrow-derived macrophages. Both types of macrophages with DLPs exhibited enhanced bacterial absorption (Fig. 6H,I). To confirm that the DLPs enhance the phagocytic capability of macrophages, we interfered the DLPs formation using the regulatory molecules we have identified. As expected, VIM KO (in THP-1 and iBMDM cells), ARHGEF3 knockdown, TLR-4 inhibition and RhoA-ROCK inhibition (in THP-1 cells) significantly reduced bacterial uptake (Figs. 6J±L and EV5G–I). Furthermore, the enhanced bacterial capture by P+ macrophages, as indicated by the increased bacterial load with higher multiplicities of infection (MOIs), highlighted the robust response of DLPs to severe bacterial challenge (Fig. 6M).

To further elucidate the physiological significance of DLPs in bacterial clearance by macrophages, we established in vivo model of

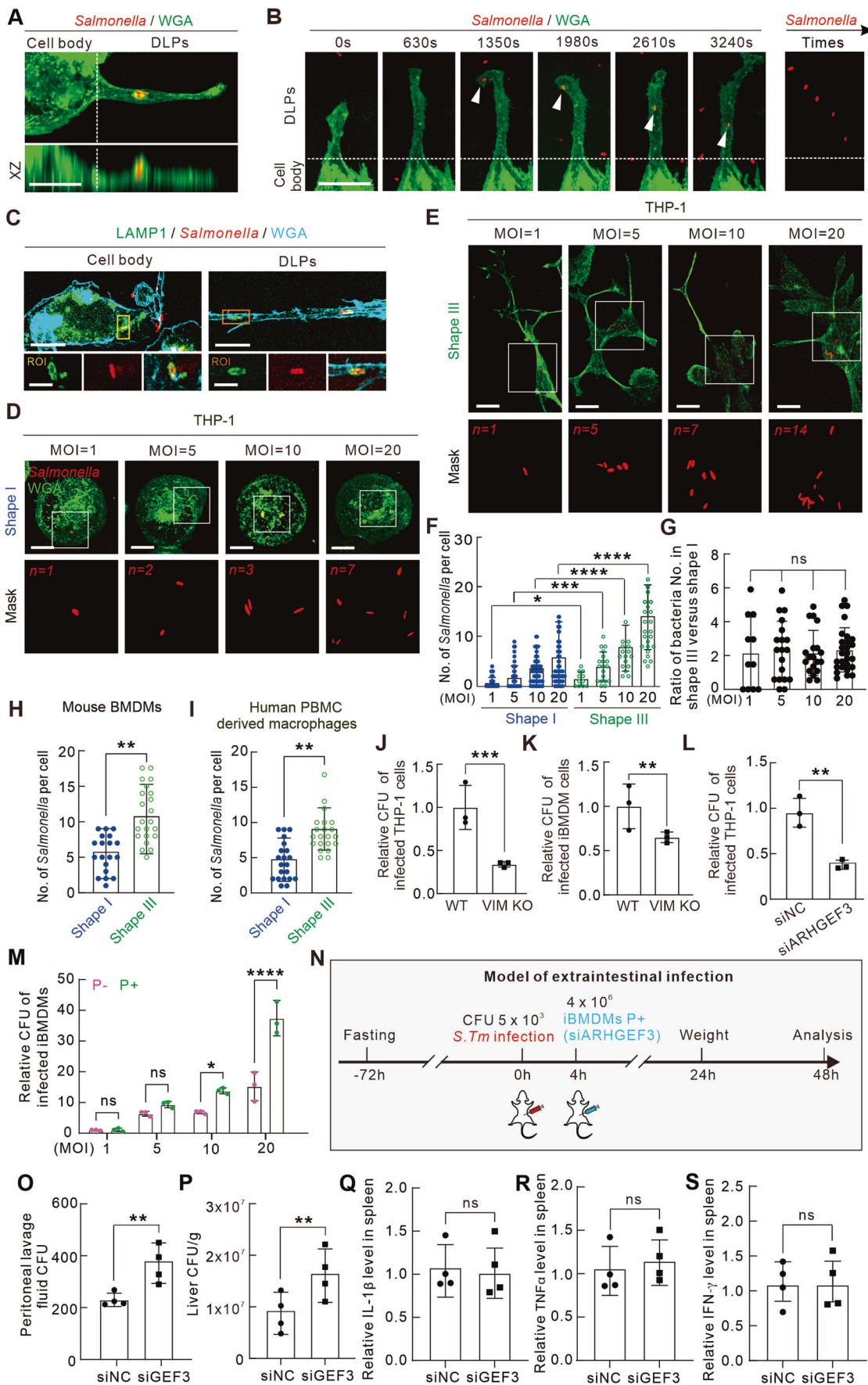

◄

**Figure 6. DLPs enhance the efficiency of bacterial ingestion of macrophages and promote the clearance of bacteria in vivo.**

(A) Representative images of mCherry-tagged *Salmonella* infected THP-1 macrophages stained with WGA. White dash line divides DLPs from the cell body. Lower panel shows the orthographic view of the upper panel image. Scale bars, 10 μm. (B) Time-lapse imaging of mCherry-tagged *Salmonella* infected THP-1 macrophages stained with WGA. White dash line depicts the cell body and DLPs. White arrowheads depict the position of an inward moving *Salmonella* bacterium along DLPs. Scale bars, 20 μm. (C) Immunofluorescence staining of endogenous LAMP1 and WGA in mCherry-tagged *Salmonella* infected THP-1 macrophages. The magnified views depict ROIs in cell body and in DLPs. Scale bars, 10 μm (in cell image) and 5 μm (in the magnified images). (D, E) Representative images of mCherry-tagged *Salmonella* in THP-1 macrophages with or without DLPs. Cell outlines were visualized with WGA staining. White squares depict ROIs containing ingested *Salmonella*. ROIs are magnified below, with ingested *Salmonella*. Scale bars, 10 μm. (F) Quantification of the number of *Salmonella* per infected cell in shape I and III macrophages at distinct MOIs. $n = 20$–30 cells. (G) Quantification of the ratio of bacteria numbers in shape III versus in shape I macrophages at distinct MOIs. $n = 15$–25 cells. (H, I) Quantification of the number of *Salmonella* per infected cell in shape I and III mouse BMDMs (J) and human PBMC derived macrophages (K). $n = 20$–25 cells. (J-L) Quantification of the bacterial load upon *Salmonella* infection in THP-1 WT and VIM-KO macrophages (J), iBMDMs WT and VIM-KO macrophages (K) and THP-1 macrophages with RNAi of negative control or ARHGEF3 (L). (M) Quantification of the bacterial load in P− and P+ iBMDM macrophages at distinct MOIs. (N) Schematic diagram of extraintestinal infection mouse model incorporating macrophage re-infusion. (O, P) Quantification of bacterial load indicated by colony forming unit (CFU) in peritoneal lavage fluid (O) and liver (P) infected with *Salmonella*. (Q–S) Quantitative RT-qPCR analysis of the expression levels of inflammatory markers in the spleen. Data are presented as mean ± s.d. from three independent experiments. Dots in quantifications presented individual cells. *P* values from left to right (F): $P = 0.03057$, $P = 0.00013$, $P = 0.00024$, $P = 4.00394E$-06. $P = 0.016278$ (H), $P = 0.00043$ (I), $P = 0.01133$ (J), $P = 0.00855$ (K), $P = 0.00477$ (L), *P* values from left to right (M): $P = 0.79525$, $P = 0.16105$, $P = 0.06649$, $P = 0.00034$. $P = 0.00769$ (N), $P = 0.00362$ (O). ns $P > 0.05$; *$P < 0.05$; **$P < 0.01$; ***$P < 0.001$; ****$P < 0.0001$: unpaired two-tailed Student's *t* test (H–L, O–S), one-way ANOVA followed by a Tukey's post hoc test (G), two-way ANOVA with Sidak's analysis (F, M). Source data are available online for this figure.

macrophage re-infusion during bacterial infection. Mice were infected intraperitoneally with *Salmonella*, followed by the intraperitoneal administration of two types of P+ macrophages (one of P+ cells with impaired expression of ARHGEF3 by RNAi, another of P+ cells treated with scrambled siRNA, which served as negative control) at 4 hpi. Two days later, tissues were collected to assess the bacterial load, thereby evaluating the impact of macrophage re-infusion on *Salmonella* infection (Fig. 6N). We assessed the bacterial load using two criteria: (1) (1) peritoneal lavage fluid, reflecting the primary infection site due to intraperitoneally infection, and (2) liver tissue, indicating secondary infection spread due the openness of the abdominal cavity to internal organs. The re-infusion of siNC P+ macrophages significantly reduced the bacterial burden at both the primary and secondary sites compared to that of siARHGEF3 P+ macrophages (Fig. 6O,P). Additionally, another in vivo model was also employed, following the same procedure as previously described, with the difference of re-infusion of P+ and P− cells (Fig. EV5J). As expected, the re-infusion of P+ cells, compared to P- cells, significantly reduced the bacterial burden (Fig. EV5K,L). Transcriptional analysis of inflammatory markers in the spleen, which were upregulated in response to *Salmonella* infection, showed a relative reduction in the P+ re-infusion group compared to P- re-infusion (Fig. EV5M–O). However, re-infusion of P+ cells with impaired ARHGEF3 expression, compared to negative control, did not significantly alter the inflammatory level within the tissue (Fig. 6Q–S), suggesting the protective effect of infection is not simply due to differential cytokine responses of P + /P− but likely to differential bacterial uptake capacity caused by ARHGEF3 regulation. These findings confirm that DLPs can suppress the progression and dissemination of infection in vivo.

## Discussion

Macrophages are important innate immune cells dedicated to pathogen digestion upon infection. Our study revealed an unprecedented efficient 'fishing nets' hunting paradigm of macrophages to fight against severe bacterial infection: (1) upon encountering Gram-negative bacteria, macrophages initiate the formation of dendrite-like pseudopods (DLPs) with blunt, bean sprout-like tips; (2) the function of DLPs is to enhance the internalization of bacteria digestion; (3) the underlying molecular mechanism of DLPs formation involves the LPS-TLR4-NF-κB-ARHGEF3-RhoA-cytoskeletal assembly. Furthermore, we revealed that the cyclic turnover of linearized actin and pronounced accumulation of vimentin are essential components for building and stabilizing DLPs, respectively. Collectively, our study uncovered a new avenue by which macrophages capture pathogens, shedding light on the intricate processes involved in their immune responses (Fig. 7).

Distinguishing DLPs from other polarized cellular substructures, such as filopodia and lamellipodia, is straightforward based on their distinct morphological, compositional, formational, and functional characteristics. Morphologically, DLPs appeared as tentacle-like structures with secondary branches and blunt tips, often extending over dozens of microns (Fig. 1G,L). However, filopodia and lamellipodia are needle-like and sheet-like structures, respectively, which are much thinner and shorter, typically spanning just a few microns (Ozawa et al, 2020; Rottner et al, 2017). Compositionally, DLPs were characterized by the dominance of linearized actin bundles and a substantial presence of vimentin filaments (Fig. 4A). The presence of vimentin in DLPs may provide physical support for its broader and more robust structure. In contrast, filopodia and lamellipodia are primarily actin-enriched structures composed of branched actin filaments, lacking intermediate filaments content (Burnette et al, 2007). In terms of formation process, filopodia and lamellipodia can form rapidly and under low levels of bacterial infection (Leijnse et al, 2015; Mannherz et al, 2007), but DLPs only form in response to substantial bacterial stimulus, indicating their distinct requirement for severe bacterial infection. Additionally, DLPs that need the transcriptional, translational and translocation regulation of ARHGEF3 upon LPS stimulation, form slowly and are more stable substructures than filopodia and lamellipodia, with a longer turnover time and maintained structure for several hours with bacterial stimulation (Fig. EV1L,M). In contrast, the turnover time for filopodia and lamellipodia is typically within minutes (Fig. EV1L,M). Functionally, DLPs display a broader radius and improved capacity for contacting and capturing bacteria, whereas filopodia and lamellipodia function as 'hook-and-shovel'-like

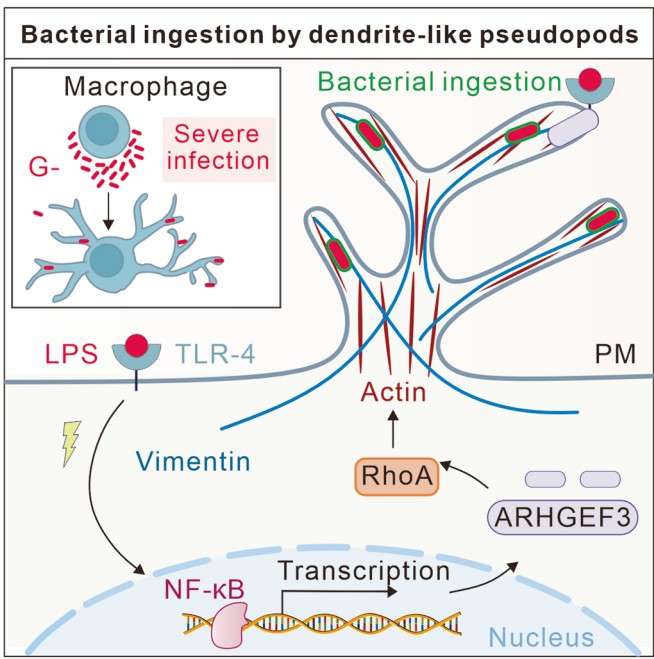

**Figure 7.** Schematic overview of the role of DLPs in enhancing the uptake of Gram-negative bacteria by macrophages.

machinery for picking up macrophage surface-bound bacteria adjacently (Krause et al, 2004). Together, these distinctions underscore that DLPs are unique cellular substructures with a clearly defined pro-phagocytic function, bolstered by the structural support from actin and vimentin, which has not been comprehensively documented in previous research.

Distinct macrophage types were examined to assess DLPs prevalence. All of them demonstrated the capability to construct DLPs in response to Gram-negative bacterial infections. Primary mouse peritoneal macrophages RAW264.7 also formed DLPs, albeit with a lower proportion, shorter length, and fewer branches, possibly due to their degree of differentiation and adhesiveness (Fig. 1M). Most DLP-positive THP-1 macrophages emerge from Shape I round, un-polarized THP-1 macrophages, while a few from Shape II fusiformed THP-1 macrophages (Fig. EV1A), and DLPs were enriched with focal adhesions compared to cell body (Fig. EV1N,O), which suggests macrophages with lower differentiation and adhesiveness might be more likely to form DLPs when exposed to bacterial stimuli. Notably, the timing of DLPs formation remained consistent among different macrophages. DLPs typically became visible 1–2 h post bacterial infection, and the proportion of macrophages with DLPs plateaued at ~5–6 h (Fig. 1K). This suggests that DLPs formation is an early event following bacterial infection, aligned with its proposed function of enhancing hunting capabilities.

Gram-positive and Gram-negative bacteria stimulate macrophages differently, primarily due to variations in receptor activation and morphological deformation. In terms of receptor activation, the lipopolysaccharide (LPS) of Gram-negative bacteria is a potent immunostimulant that binds to the Toll-like receptor 4 (TLR4) on the surface of macrophages, leading to an intense inflammatory response and the production of pro-inflammatory cytokines like

TNF-α and IL-1β (Elson et al, 2007). In contrast, Gram-positive bacteria activate macrophages through cell wall components such as peptidoglycan and teichoic acid, which bind to TLR2, resulting in a comparatively weaker immune response (Elson et al, 2007; Takeuchi et al, 1999). Notably, in terms of morphological deformation, Gram-negative bacteria typically induce more significant changes in the shape and activation state of macrophages (Malainou et al, 2023). It is worth mentioning that the effectors of *Salmonella* type III secretion system remodel the actin cytoskeleton and induce membrane protrusions and ruffling during infection (Hernandez et al, 2004; Patel and Galán, 2006). To clarify whether the type III secretion system is involved in the formation of DLPs. We generated *Salmonella* strains lacking SPI-1 (SipB mutant) or SPI-2 (PhoP mutant). By infecting THP-1 macrophages with these mutants, we observed that DLPs still formed in similar shape and proportion compared to wild-type strain (Appendix Fig. S1). Additionally, the actin rearrangements and membrane ruffling caused by the type III secretion show features of disordered actin perturbations and slight membrane ruffling morphology (Patel and Galán, 2005), which are distinctly different from DLPs, indicating type III secretion system not regulating DLP formation. For the feature of Gram-negative bacteria, one explanation is that LPS stimulation triggers the assembly of the Piezo1 and TLR4 complex to remodel F-actin organization (Geng et al, 2021). In our study, we found that LPS not only triggers a robust immune response but also promotes the DLPs formation in macrophages, enhancing the uptake and clearance of Gram-negative bacteria. Thus, our findings supplement the differential morphological responses of macrophages to Gram-positive and Gram-negative bacteria, which is crucial for understanding the host response to various bacterial infections.

It is interesting to note that preconditioning with Gram-negative bacteria or LPS can enhance phagocytosis by macrophages and reduce the bacterial load in organs and systemic inflammatory levels compared to preconditioning with Gram-positive bacteria (Corsini Campioli et al, 2022; Gao et al, 2023). A previous study proposed that LPS-induced rearrangement of plasma membrane phospholipids and increased motility of CD44 enriches phagocytic receptors, thereby enhancing phagocytosis (Corsini Campioli et al, 2022). In addition to receptor distribution, we found that LPS binding to TLR4 activated the NF-κB-ARHGEF3-RhoA pathway, promoting actin filament assembly and DLPs formation in macrophages, which enhanced their capacity to capture bacteria. These findings contribute to the understanding of this mechanism by highlighting a morphological adaptation that enhances bacterial uptake, a factor that may have been previously overlooked.

Although ARHGEF3 is broadly expressed in macrophages (Arthur et al, 2002), its specific functions have received little attention. It is known that ARHGEF3 can control histone deacetylase (HDAC) inhibitor-induced macrophage differentiation via RhoA-dependent pathways in U937 macrophage (D'Amato et al, 2015). LPS treatment significantly increases ARHGEF3 expression in microglia, which are functionally similar to macrophages in the brain and spinal cord and capable of engulfing and digesting pathogens and cellular debris (Liao et al, 2021). These findings thus suggest that ARHGEF3 may have a broad regulatory impact on various aspects of the host immune system. Our study highlights the previously underexplored role of ARHGEF3 in macrophages during infection, indicating its potential as a target

for anti-infection drug development. The RhoGEF GEF-H1 activator Plinabulin has shown significant efficacy as a clinical drug for neutropenia (Kashyap et al, 2019). However, no activators targeting ARHGEF3 have been identified by far, developing of which targeting macrophages to increase infection resistance possesses a potential clinical implication.

Remarkably, intraperitoneal transfusion of P+ (DLPs-equipped) macrophages with impaired ARHGEF3 expression in mice significantly reduced bacterial infection load but not inflammation level (Fig. 6O–S), which indicates that beyond the inflammatory cytokine secretion, the phagocytic capacity of macrophages themselves also plays a crucial role in immune defense against bacterial infections in vivo. Taken together, our discovery of DLPs reveals a previously overlooked subcellular substructure in macrophages, designed to enhance their bacterial capture efficiency, underscoring the intimate connection between dynamic morphological changes and the physiological roles of macrophages.

# Methods

### Reagents and tools table

| Reagent/resource | Reference or source | Identifier or catalog number |
|---|---|---|
| **Experimental models** | | |
| C57BL/6J (*M. musculus*) Jackson Lab #000664; RRID: | C57BL/6J (*M. musculus*) Jackson Lab #000664; RRID: | C57BL/6J (*M. musculus*) Jackson Lab #000664; RRID: |
| THP-1 | ATCC | TIB-202 |
| THP-1 TLR4-KO | Prof. Guangxun Meng | N/A |
| THP-1 VIM-KO | This study | N/A |
| iBMDM | Prof. Hong Tang | N/A |
| iBMDM VIM-KO | This study | N/A |
| iBMDM TLR4-HA EGFP | This study | N/A |
| RAW264.7 | Prof. John Eriksson | N/A |
| Mouse Peritoneal macrophages | C57BL/6J (*M. musculus*) Jackson Lab #000664; RRID: | N/A |
| U2OS | Prof. Pekka Lappalainen | N/A |
| MoDC | PBMC | N/A |
| PBMC-derived macrophages | PBMC | N/A |
| *Salmonella Typhimurium* (S. Tm; strain LT2) | Prof. Haihua Ruan | N/A |
| *Salmonella Typhimurium* (S. Tm; strain LT2)(mCherry-tagged) | Prof. Haihua Ruan | N/A |
| *E. coli* EPEC | Prof. Yanjie Chao | N/A |
| *Shigella* | Prof. Yanjie Chao | N/A |
| *Bacillus* | Prof. Jean-Marc Collard | N/A |
| *Listeria* | Prof. Jean-Marc Collard | N/A |
| **Recombinant DNA** | | |
| RhoA (CA, L63) | Addgene | #15900 |
| RhoA (DN, N19) | Addgene | #15901 |
| pMSCV2.2-TLR4-HA-EGFP | Prof. Bo Liu | N/A |
| pSpCas9-2A-GFP | Addgene | #48138 |
| pSpCas9-2A-GFP-sgVIM1 | This study | N/A |
| pSpCas9-2A-GFP-sgVIM2 | This study | N/A |

| Reagent/resource | Reference or source | Identifier or catalog number |
|---|---|---|
| pGL3 basic | Promega | #PAE1751 |
| Pgl3 prom-ARHGEF3 | This study | N/A |
| **Antibodies** | | |
| Rabbit vimentin antibody | Cell Signaling | #5741 |
| Rabbit p-NF-κB antibody | Abcam | #ab51074 |
| Mouse β-actin antibody | Proteintech | #66009-1-Ig |
| Rabbit LAMP1 antibody | Cell Signaling | #99437 |
| Rabbit TLR4 antibody | Proteintech | #19811-1-AP |
| Mouse β-tubulin antibody | Sigma | #T4026 |
| Rabbit ARHGEF3 antibody | Abmart | #TD4434 |
| PE Anti-F4/80 | Abcam | #ab105156 |
| Rabbit CD64 antibody | Proteintech | #84518-2-RR |
| Rabbit CD32 antibody | Proteintech | #65057-1-Ig |
| Rabbit CD14 antibody | Proteintech | #17000-1-AP |
| Rabbit NF-κB p65 antibody | Cell Signaling | #8242 |
| Rabbit phospho-NF-κB p65 (Ser536) antibody | Cell Signaling | #3033 |
| Mouse myosin light chain antibody | Sigma | #M4401 |
| Rabbit Phospho-myosin light chain 2 (Thr18/Ser19) antibody | Cell Signaling | #3674 |
| Mouse vimentin antibody | Cell Signaling | #49636 |
| Mouse GAPDH antibody | Sigma | #G8795 |
| Rabbit RhoA (67B9) antibody | Cell Signaling | #2117 |
| Rabbit HA antibody | Sigma | #H6908 |
| NF-κB p65 specific antibody | Cell Signaling | #8242 |

| Reagent/resource | Reference or source | |
|---|---|---|
| **Oligonucleotides and other sequence-based reagents** | Forward | Reverse |
| RNAi ARHGEF3 (human) | GCACACACCAAA UGAUAAUTT | AUUAUCAUUUGGU GUGUGCTT |
| RNAi ARHGEF11 (human) | ACCGUACAAACCACCA AAUCCUCUG | CAGAGGAUUUGG UGGUUUUGUACGGU |
| RNAi ARHGAP10 (human) | GCUGGAAGA UGGUGACAAUU | AUUGUCACC AUCUUCCAGCTT |
| Negative control | UUCUCCGAACGUGU CACGUTT | ACGUGACACGUU CGGAGAATT |
| RNAi ARHGEF3 (mouse) | GCGGCUCCUUCAC UUGGAAUU | UUCCAUGUAAAG UAGCCGCTT |
| Mouse ARHGEF3 | AAGCAGGACC ACAGAGTGC | TGCGACTGAAGGGTGA TTCTAA |
| Mouse ARHGAP18 | TCGGGAGTTGT GCTAACTGC | GGCCATATCTGCGAC TGGAG |
| Human GAPDH | GCATCCTGCACCA CCAACTG | GCCTGCTTCACCAC CTTCTT |
| Mouse GAPDH | CCTTCCGTGTTCC TACCCC | GCCCAAGATGCCCT TCAGT |
| Human ARHGEF11 | CGCTGTGTCATTATC CAAAAGGA | CATGGTGCCGTTG ACTTTGA |
| Mouse ARHGEF11 | GGTGAATGACTCGGA AGTGTATC | TCCCTTGTCCTGATGC CTCT |
| Human ARHGEF10 | CCCAGCGGAAGTTT GCTCAT | ACAGCTCCAAGTTGC TCTTTTC |
| Mouse ARHGAP10 | CAGCCCCTGGAG TTTAGCG | GGTTCTTGCCGTCTTTG ATGAG |
| Mouse TNFα | CATCTTCTCAAAATT CGAGTGACAA | TGGGAGTAGACAAGG TACAACCC |

| Reagent/resource | Reference or source | Identifier or catalog number |
|---|---|---|
| Mouse IL-1β | CTCTCCAGCCCA AGCTTCCTTGTGC | GCTCTCATCAGGACAG CCCAGGT |
| Mouse IFN-γ | TCAAGTGGCATAGATG TGGAAGAA | TGGCTCTGCAGGAT TTTCATG |
| **Chemicals, enzymes and other reagents** | | |
| RPMI-1640 | Gibco | #11875093 |
| fetal bovine serum (FBS) | Gibco | #10270-106 |
| Dulbecco's Modified Eagle's Medium (DMEM) | Biological Industries | #06-1055-57-1A |
| Brewer thioglycollate medium | BD Biosciences | #221742 |
| Human M-CSF | MCE | HY-P7050A |
| Mouse M-CSF | MCE | HY-P7085A |
| Human GM-CSF | Proteintech | #HZ-1002 |
| Human IL-4 | MedChemExpress | #HY-P70445 |
| LPS | InvivoGen | #5969-44-01 |
| Human IL-6 | Peprotech | #P05231 |
| 35 mm glass-bottomed dishes | MatTek Corporation | #P35G-1.5-14-C |
| fibronectin | Sigma | #F2006 |
| SiR-actin | Cytoskeleton | #CY-SC001 |
| WGA | Invitrogen | #W11261 |
| Lipofectamine™ LTX | Invitrogen | #15338100 |
| Ketamine hydrochloride | Sigma-Aldrich | #K4138 |
| Xylazine hydrochloride | Sigma-Aldrich | #K4138 |
| Polydimethylsiloxane (PDMS) | MOMENTIVE | RTV615 |
| 24-well transwell plate | NEST | 725121/725311 |
| protease inhibitors | Beyotime | #P1045 |
| Protein A/G PLUS-Agarose | Santa Cruz Biotechnology | #sc-2003 |
| proteinase K | Sigma-Aldrich | #39450-01-6 |
| QIAamp DNA mini kit | Qiagen | #51304 |
| Bio-RadGene Pulser Electroporation buffer | Bio-Rad | #1652676 |
| LipoRNAi MAX transfection reagent | Invitrogen | #13778100 |
| EZ-press RNA Purification Kit | EZBioscience | #B0004DP |
| Color Reverse Transcription Kit | EZBioscience | #A0010CGQ |
| 2× Color SYBR Green qPCR Master Mix (ROX2 plus) | EZBioscience | #A0012-R2 |
| blebbistatin | Sigma | #b0560 |
| Latrunculin B | Santa Cruz | #sc203318 |
| SMIFH2 | Millipore | #344092 |
| NP-G2-044 | Sigma | #2962 |
| CK666 | Sigma | #sml0006 |
| Nocodazole | Sigma | #SML1665 |
| LTA | InvivoGen | #tlrl-pslta |
| Simvastatin | Selleck | #S1792 |
| TAK242 | Selleck | #S7455 |
| LPS | Sigma | #L2630 |
| PDTC | Beyotime | #S1808 |
| Y-27632 | Selleck | #S6390 |
| Cell Counting Kit-8 | Beyotime | #C0040 |

| Reagent/resource | Reference or source | Identifier or catalog number |
|---|---|---|
| Active Rho Detection Kit | CST | #8820 |
| **Software** | | |
| GraphPad Prism | https://www.graphpad.com | |
| ImageJ | https://imagej.net/ij/ | |
| Imaris 10.0 | Oxford Instruments | |
| CellSens Dimension system | Olympus | |
| FV31S-SW software | Olympus | |
| **Other** | | |
| Synergy H1 Hybrid Multi-Mode Reader | BioTek | |
| Sonics Vibra Cell Processors | Sonics | #VC 505 |
| QuantStudio 1 system | Thermo | |

## Cell culture

THP-1 human monocytic leukemia cells were cultured in Roswell Park Memorial Institute medium (RPMI-1640) (#11875093, Gibco) supplemented with 10% fetal bovine serum (FBS) (#10270-106, Gibco), 10 U/mL penicillin, 10 μg/mL streptomycin, 100 μM nonessential amino acids, 1 mM sodium pyruvate, and 55 mM mercaptoethanol at 37 °C and 5% $CO_2$ with a density of $1 \times 10^6$ cells/mL. For differentiation into adherent macrophages, suspended THP-1 cells were treated with 100 ng/mL phorbol 12-myristate 13-acetate (PMA) for 48 h.

RAW 264.7 murine macrophages, immortalized murine bone marrow-derived macrophages (iBMDMs), and human osteosarcoma (U2OS) cells were maintained in high-glucose (4.5 g/L) Dulbecco's Modified Eagle's Medium (DMEM) (#06-1055-57-1 A, Biological Industries) supplemented with 10% FBS, 100 U/mL penicillin, 100 μg/mL streptomycin, and 4 mM L-glutamine (later referred as complete DMEM) at 37 °C and 5% $CO_2$.

Peritoneal elicited macrophages (PEMs) were collected from mice 4 days after intraperitoneal (i.p.) injection of 3% Brewer thioglycollate medium (#221742, BD Biosciences). The harvested liquid was centrifuged at 1000 rpm, and the cell pellet was resuspended and washed three times with phosphate-buffered saline (PBS). Cells were incubated with CD11b and F4/80 antibodies on ice for 30 min in the dark and then resuspended in 500 μL PBS with 3% FBS, sorted by flow cytometry and used immediately for further infection experiments.

For culturing bone marrow-derived macrophages (BMDMs), the femur and tibia were collected from mice of each genotype, and bone marrow cells were flushed with complete DMEM. Erythrocytes were removed via treatment with red blood cell lysis buffer, and cell suspensions were filtered through a 40-μm cell strainer to remove clumps. Single-cell suspensions were cultured for 1 h at 37 °C. Non-adherent cells were then re-plated in complete DMEM with 25% recombinant mouse M-CSF (carrier-free, 50 ng/mL) conditioned medium. For complete differentiation, BMDMs were cultured for an additional 8 days, with the medium replaced every 2 days.

For culturing human peripheral blood mononuclear cell (PBMC)-derived macrophages, PBMCs were isolated from whole

blood using density gradient centrifugation and enriched for CD14+ monocytes via magnetic cell separation with anti-CD14 antibodies. These monocytes were plated in tissue culture plates with 25% medium conditioned by recombinant human M-CSF (carrier-free, 20 ng/mL). For complete differentiation into macrophages, cells were cultured for an additional 8 days with the medium replaced every 2 days.

For generating monocyte derived dendritic cells (MoDCs), PBMCs were isolated from whole blood using density gradient centrifugation and enriched for CD14+ monocytes via magnetic cell separation with anti-CD14 antibodies. These monocytes were treated with 100 ng/mL GM-CSF (#HZ-1002, Proteintech) and IL-4 (#HY-P70445, MedChemExpress) for 5 days, with media and cytokines replaced on day 5 to generate moDCs. These moDCs were then stimulated with IL-6 (P05231, Peprotech) and Lipopolysaccharide (LPS) (#5969-44-01, InvivoGen) for 24 h to promote DC maturation and activation before analysis on day 8.

## Bacterial strains and infection

*Salmonella typhimurium* (*S.Tm*; strain LT2) was provided by Dr. Haihua Ruan (Tianjin University of Commerce, China). Fluorescence-tagged *Salmonella* strains were generated via electroporation of strain LT2. Heat-killed bacteria were prepared by culturing *Salmonella* in Luria-Bertani medium at 65 °C for 20 min (Block et al, 2008), followed by centrifugation at 6000 rpm for 5 min to precipitate the bacteria. Enteropathogenic *E. coli*, EPEC and *Shigella flexneri* were provided by Dr. Yanjie Chao (Chinese Academy of Sciences, China). *Bacillus thuringiensis* and *Listeria monocytogenes* were provided by Dr. Jean-Marc Collard (Chinese Academy of Sciences, China).

For infection of THP-1, RAW264.7 and iBMDMs macrophages, the bacteria were cultured overnight at 37 °C in LB broth, washed in PBS, and transferred 1:33 to 2 mL fresh LB medium and grown for another 2 h to logarithmic phase. Macrophages were washed once in pre-warmed PBS and cultured at 37 °C in RPMI-1640 containing 10% FBS, and then infected with suspended *Salmonella* in serum-free, antibiotic-free medium for 30 min. The supernatant medium with bacteria was then collected, centrifuged and resuspended in medium with 10% FBS and 2 μg/mL gentamicin. This gentamicin concentration effectively prevents bacterial overgrowth while maintaining infectivity throughout the incubation period. Cells were re-infected with these gentamicin-added bacteria. This time point counts as 0 h of infection. Unless otherwise specified, *Salmonella* was added to macrophages at a multiplicity of infection (MOI) of 20. Infections were performed at 37 °C.

## Sholl analysis

Sholl analysis is a commonly used quantitative method in neuronal studies to characterize the morphological characteristics of an imaged neuron (Binley et al, 2014). Specifically, Fiji ImageJ (version 1.53 f, Wayne Rasband, National Institutes of Health NIH) was used for image processing. Raw images were converted to 8-bit grayscale. The ImageJ line tool was used to draw a line from the cell body center to the longest extension. The starting radius was set to encompass the cell body, and the ending radius was set to the distance from the cell center to the longest protrusion. The radius interval was determined between the two analysis radii. The analysis produced intersection points and concentric circles on the image.

## Immunofluorescence microscopy

Immunofluorescence (IF) experiments were performed as previously described (Zhang et al, 2019). Briefly, cells were fixed with 4% paraformaldehyde in PBS for 15 min, washed twice with PBS, permeabilized with 0.02% Triton X-100 in PBS for 5 min, and washed twice more with PBS. Fixed cells were blocked with 5% BSA for 30 min before incubation with the following primary antibodies: rabbit vimentin antibody (#5741, Cell Signaling, dilution 1:500); rabbit p-NF-κB antibody (#ab51074, Abcam, dilution 1:200); mouse β-actin antibody (#66009-1-Ig, Proteintech, dilution 1:500); rabbit LAMP1 antibody (#99437, Cell Signaling, dilution 1:500); rabbit TLR4 antibody (#19811-1-AP, Proteintech, dilution 1:500); mouse β-tubulin antibody (#T4026, Sigma, dilution 1:500); rabbit ARHGEF3 antibody (#TD4434, Abmart, dilution 1:200); PE Anti-F4/80 (ab105156, Abcam, dilution 1:20). Rabbit CD64/CD32/CD14 antibody (#84518-2-RR, # 65057-1-Ig, #17000-1-AP, Proteintech, dilution 1:500), Samples were washed with PBS for three times and incubated with Alexa Fluor-conjugated secondary antibodies (Thermo Fisher Scientific) for 1 h at room temperature, followed by three washes with PBS. The imaging data were obtained with an Olympus SpinSR10 Ixplore spinning disk confocal microscope with a UplanApo 100×/1.5 Oil objective (Olympus Corporation). The pixel size was optimized for maximum resolution, calculated as 65 nm. Super-resolution images were acquired using structured illumination microscopy, which is based on Olympus IX83 microscopy equipped with an ApoN 100X/1.7 Oil TIRF objective, multiband dichroic mirror (DM, ZT405/488/561/640-phase R), Sapphire 488LP-200; 561LP-200; and MRL-III-640-150 as light sources. An acoustic optical tunable filter (AOTF, AA Opto-Electronic) controlled laser illumination power. Nine raw frames illuminated with a periodic pattern and shifted through three phases per orientation angle were reconstructed to generate one super-resolution image.

## Live-cell imaging

For bright-field live-cell imaging, cells were seeded in a 6-well plate and cultivated at 37 °C in 5% $CO_2$. Images were captured using Olympus IX73 inverted microscope with the UplanFL 10×/0.3 objective (Olympus Corporation).

For fluorescence live-cell imaging, 35 mm glass-bottomed dishes (MatTek Corporation) were coated with 10 μg/mL fibronectin (#F2006, Sigma) in PBS for at least 3 h at 37 °C, washed with PBS twice, and immersed in cell culture medium without phenol red before seeding cells. Time-lapse images of cells were acquired with the Olympus cellSens Dimension system, consisting of an Olympus Ixplore SpinSR10 spinning disk confocal and a Yokogawa CSU-W1 confocal scanner. Appropriate filters, a heated sample environment (37 °C), controlled 5% $CO_2$ and a UplanApo 100×/1.5 Oil objective (Olympus Corporation) was used. Z drift compensation (ZDC) was used to maintain a single focal plane throughout image acquisition. For labeling F-actin, cells were incubated with 0.2 μM SiR-actin (#CY-SC001, Cytoskeleton) for 6 h. The addition of 10 μm

verapamil improved the staining efficiency. For labeling WGA, the prewarmed (37 °C) WGA (#W11261, Invitrogen)-containing medium (10 μg/mL) was applied cells for 2 h, then replaced with fresh medium and followed bacterial infection to observe the cells using a fluorescence microscope.

## Cell debris production

Cell debris was generated using ultrasonication (Wang and Yuan, 2016). Briefly, Caco2 cells were resuspended in 1 mL DMEM and subjected to ultrasonication (VC 505, Sonics Vibra Cell Processors) at 4 °C for 15 min with 30% power efficiency. Debris was then pelleted by centrifugation at 800 rpm for 10 min. The resulting cell debris ranged from 1 to 5 μm in size, comparable to bacteria.

## Transfection of plasmids

THP-1 macrophages were transiently transfected with constitutively active RhoA (CA, L63) (#15900, Addgene), dominant negative RhoA (DN, N19) (#15901, Addgene) or EGFP (negative control) by using Lipofectamine™ LTX (#15338100, Invitrogen), and then bright-field live-cell imaging was conducted.

## Two-photon intravital microscopy (TIM)

TIM was performed on 8- to 10-week-old mice, weighing ~20 g. Mice were injected with *Salmonella* 20 min before imaging. Anesthesia was induced with ketamine hydrochloride (200 mg/kg, Sigma-Aldrich) and xylazine hydrochloride (10 mg/kg, Sigma-Aldrich). After anesthesia, the tail vein was cannulated for additional anesthetic administration. A heated stage was used to maintain body temperature at 37 °C.

The ventral abdominal wall was aseptically cleaned with 70% ethanol, and the fur and skin were removed, avoiding the peritoneal cavity. If the peritoneal cavity was accidentally opened, the mouse was euthanized immediately. Staining antibody was injected intraperitoneally with a 28 G needle, and the injection site was sutured to immobilize the peritoneum onto the coverslip. The mouse was placed in a semi-prone position, and gentle pressure was applied with a cotton tip to stabilize imaging while preserving peritoneal fluid movement. After TIM, all mice were euthanized immediately.

For imaging, mice were placed supine on a heated stage to maintain body temperature at 37 °C. The abdominal wall was cleaned with 70% ethanol and injected with staining antibodies (anti-mouse F4/80). After 10 min incubation on a heated stage, the peritoneum was incised in an inverted a U-shaped from below the xyphoid process to inguinal ligament, creating a perfused flap of peritoneum and the rectus abdominis muscles. The flap was washed with 1 mL of PBS and placed on the glass coverslip of the heated stage for imaging of the abdominal wall or epigastric blood vessels.

Intravital peritoneal imaging was performed as previously described (Zindel et al, 2021). Exposed tissues were visualized with an FVMPE-RS upright microscope (Olympus Corporation) using MaiTai DeepSee (690–1040 nm) and Insight DeepSee (690–1300 nm) lasers. All images shown here were acquired with a 25×1.05 NA objective, 2× digital zoom, and water as the immersion medium. The MaiTai laser was set to 870 nm to excite second harmonic waves and GFP collagen visualization. The Insight laser was set to 1120 nm for PE excitation to visualize macrophages. Emission wavelengths were detected using ultra-sensitive photomultiplier tubes (GaSP, Hamamatsu) for 410–560 nm (second harmonic signal) and 575–645 nm (PE). FV31S-SW software controlled the microscope. After TIM, all mice were euthanized immediately.

Most mouse peritoneal macrophages labeled with F4/80 and observed utilizing two-photon microscopy adhered to the peritoneal wall and aligned with the ECM network, exhibiting sleek and slender morphology. We defined 'with pseudopods' as when a macrophage derived three or more protrusions from two sides of its elongated axis, and each protrusion was over five microns in length. Reciprocally, we defined 'without pseudopods' as when a macrophage showed less than two protrusions.

The 3D overview is obtained from the automated surface reconstruction in Imaris 10.0 software (Oxford Instruments). To ensure objectivity, double blind manual quantification was used for calculating the percentage of macrophages with pseudopods. Briefly, one person numbered and mixed the Mock and infected data, and other person performed the outline analysis and calculated the percentage of the macrophages with pseudopods. The 3D reconstruction in Fig. 1D is from the automated surface reconstruction in Imaris 10.0.

## Mice infection

S.*Tm* (strain LT2) was cultured overnight at 37 °C and then subcultured the next day in fresh LB at 37 °C until reaching an $OD_{600}$ of 0.9–1.0. Male 8-weeks-old C57/B6J mice ($n = 4$ per group) were housed in a specific pathogen free (SPF) facility with controlled temperature (18–22 °C), humidity (50–60%), and a 12-h dark/light cycle. After 72 h of rest, mice were intraperitoneally injected with $5 \times 10^3$ S.*Tm*. At 4 hpi, mice received an intraperitoneal injection of $4 \times 10^6$ P+/P−/P+ siNC/P+ siARHGEF3 iBMDMs. Mice for tissue and organ collection were anesthetized and euthanized at 48 hpi, while those for body weight measurements were euthanized at 72 hpi. The liver and peritoneal lavage fluid were collected for Colony Forming Units (CFU) analysis, and the spleen was harvested for cytokine expression analysis using real-time quantitative PCR.

## Microfluidic device

The microfluidic device was fabricated from Polydimethylsiloxane (PDMS) (RTV615, MOMENTIVE) using soft lithography from a patterned SU-8 silicon wafer. Glass coverslips were plasma bonded to the PDMS layer. Each device consisted of three main chambers with dimensions of 600 μm in width, 200 μm in height, and 5000 μm in length. The micro-channel was 10 μm in width, 200 μm in height, and 600 μm in length. All channels were coated with 10 μg/mL fibronectin for 1 h at 37 °C before use.

## Transwell assay

Transwell assay was performed using a 24-well transwell plate (0.4/8 μm pore size polyethylene terephthalate (PET) membrane, translucent, TC, sterilized, 725121/725311, NEST). *Salmonella* was

seeded in the upper chambers at an MOI of 20 in Roswell Park Memorial Institute (RPMI) 1640 medium (Gibco) supplemented with 10% FBS (Sigma-Aldrich), 2 mM L-glutamine and 2 µg/mL gentamycin. The lower chamber contained 500 µL of RPMI 1640 medium, while the upper chamber contained 300 µL. THP-1 cells were seeded in the lower chambers to assess their response to *Salmonella*. Plates were then incubated for 6 h. After incubation, PET membranes were washed three times with PBS, cut off, and fixed in 4% paraformaldehyde for 30 min. Imaging was performed using an Olympus SpinSR10 Ixplore spinning disk confocal microscope with a UplanApo 100×/1.5 Oil objective (Olympus Corporation).

## Co-immunoprecipitation

iBMDMs were transfected with pMSCV2.2-TLR4-HA-EGFP (#60206, Addgene) by lentivirus and sorted for GFP-positive cells by flow cytometry, which assumed as stable transfection after enrichment and verification by macroscopy. For chemical treatment, LPS was added to the culture medium. After 6 h, cells were collected, washed with cold PBS, and lysed with lysis buffer (50 mM Tris-HCl, pH 8.0; 150 mM NaCl; 0.5% NP-40; 1 mM EDTA; protease inhibitors from Beyotime, #P1045). Cell lysates were centrifuged at 10,000×*g* for 10 min at 4 °C, and the supernatant was collected. Protein concentration was determined using the BCA Protein Assay Kit (#P0010, Beyotime). A 100 µL aliquot of the supernatant was reserved for input detection. The remaining supernatant, containing equal protein amounts, was incubated overnight at 4 °C with Protein A/G PLUS-Agarose (#sc-2003, Santa Cruz Biotechnology) and ARHGEF3 antibody (#TD4434, Abmart) or HA antibody (#H6908, Sigma). Beads were washed four times with lysis buffer and eluted with 40 µL SDS loading buffer at 98 °C for 5 min for further western blotting analysis.

## CFU assay

THP-1 macrophages were incubated with fresh overnight cultures of *Salmonella* (MOI = 20) for the specified time points at 37 °C. Cells were washed three times with PBS to remove extracellular bacteria, resuspended in distilled $H_2O$ to lyse macrophages, and diluted aliquots were plated on LB agar. CFUs were counted after overnight incubation at 37 °C (Day et al, 2009).

## Chromatin immunoprecipitation (ChIP) assay

Briefly, $1 \times 10^6$/mL cells were fixed in 1% formaldehyde in PBS for 10 min at room temperature. The cross-linking reaction was quenched by adding 1/20 volume of 2.5 M glycine. Cells were then incubated in ChIP lysis buffer (50 mM Tris-HCl pH 8.0, 5 mM EDTA, 150 mM NaCl, 1% NP-40, 0.1% SDS, with protease inhibitor) for 30 min on ice. Chromatin was sheared by sonication to yield fragments suitable for immunoprecipitation. Cross-linked chromatin samples were incubated with the NF-κB p65 specific antibody (#8242, Cell Signaling) or a control normal rabbit IgG (#2729, Cell Signaling) overnight at 4 °C on a rotator. Subsequently, protein A/G-conjugated agarose beads (#sc-2003, Santa Cruz Biotechnology) were added and incubated overnight in the rotator at 4 °C. The beads were collected and washed three times.

To elute DNA fragments, immunocomplexes were incubated with elution buffer (50 mM Tris-HCl, pH 8.0, 10 mM EDTA, 1.0% SDS) for 2 h at 65 °C. A portion of the eluted immunocomplexes was reserved as ChIP sample, while the remainder was treated with proteinase K (#39450-01-6, Sigma-Aldrich) overnight at 55 °C. Finally, DNA was purified using the QIAamp DNA mini kit (#51304, Qiagen) and subjected to qPCR analysis for detection of target DNA.

## CRISPR knockout cell line generation

VIM KO cells were generated using CRISPR/Cas9 methods based on pSpCas9-2A-GFP vector (#48138, Addgene) with two targets (Jiu et al, 2017) by electroporation using Bio-RadGene Pulser Electroporation buffer (#1652676, Bio-Rad). Primers for vimentin target 1 were 5'-CACCGTGGACGTAGTCACGTAGCTC-3' and 5'-AAACGAGCTACGTGACTACGTCCAC-3'. Primers for vimentin target 2 were 5'-CACCGCAACGACAAAGCCCGCGTCG-3' and 5'-AAACCGACGCGGGCTTTGTCGTTGC-3'. Transfected THP-1 or iBMDMs were collected 24 h post-transfection and sorted with FACS Aria II (BD Biosciences) using low intensity GFP-expression pass gating. Sorted cells were plated in 96-well plates with RPMI 1640 medium for THP-1 or DMEM for iBMDM, both containing 20% FBS. Cells were cultivated for two weeks and selected for the absence of vimentin protein expression by western blot. TLR4 KO cells were generated using CRISPR/Cas9 methods based on lentiCRISPRv2 vector (Addgene #98290) by lentivirus with two targets. Primers for TLR4 target 1 were 5'-CACCGGA-CATCTTATTCCACATATC-3' and 5'-AAACGATATGTGGAA-TAAGATGTCC-3'. Primers for TLR4 target 2 were 5'-CACCGCATACTCCTAATTATTAAGC-3' and 5'-AAACGCT-TAATAATTAGGAGTATGC-3'. TLR4 KO cell lines were selected by puromycin (1 µg/mL) pressure.

## Gene knockdown by RNA silencing

siRNAs were transfected into THP-1 macrophages with LipoRNAi MAX transfection reagent (Invitrogen, #13778100). Experiments were performed 2 days post-transfection. siRNAs were used at 10 pmol per well.

## Real-time RT-PCR

Total cellular RNA was extracted by EZ-press RNA Purification Kit (#B0004DP, EZBioscience) according to the manufacturer's protocols. Total RNA was reverse transcribed by Color Reverse Transcription Kit (#A0010CGQ, EZBioscience). Real-time RT-PCR was carried out by 2× Color SYBR Green qPCR Master Mix (ROX2 plus) (#A0012-R2, EZBioscience) in QuantStudio 1 system (Thermo). All readings were normalized to the level of GAPDH.

## Chemical treatment

The following drugs were used: blebbistatin (#b0560, Sigma; 10 µM for 1 h), Latrunculin B (LatB) (#sc203318, Santa Cruz; 0.5 µM for 1 h), SMIFH2 (#344092, Millipore;15 µM for 1 h), NP-G2-044 (#2962, Selleck; 1 µM for 1 h), CK666 (#sml0006, Sigma; 40 µM for 1 h), Nocodazole (#SML1665, Sigma; 1 µM for 1 h), simvastatin (#S1792, Selleck; 1 µM for

12 h), LTA (#tlrl-pslta, InvivoGen; 1 µg/mL for up to 10 h), TAK-242 (#S7455, Selleck; 1 µM for 2 h), Y-27632 (#S6390, Selleck; 1 µM for 24 h), LPS from *Escherichia coli* O111:B4 (LPS, #L2630, Sigma; 100 ng/mL for 10 h), PDTC (#S1808, Beyotime; 50 µM for 1 h).

## Luciferase assay

The promoter region (2000 bp upstream) of ARHGEF3 containing NF-κB binding sites was amplified and inserted into the pGL3 basic vector (Promega), to generate pGL3-prom-ARHGEF3. iBMDMs at 80% confluence were transfected this construct, treated after 48 h, and luciferase activity was quantified using the Synergy H1 Hybrid Multi-Mode Reader (BioTek).

## Cell viability assay

Cell viability was measured using Cell Counting Kit-8 (#C0040, Beyotime). Briefly, cells were plated at a density of $2 \times 10^3$ cells per 100 µL in a 96-well culture plate and incubated with 10 µL of CCK-8 reagent at 37 °C for 0.5-4 h. Absorbance at 450 nm was measured using a Synergy H1 Hybrid Multi-Mode Reader (BioTek).

## Bacterial activity detection

*Salmonella* strains were cultured overnight at 37 °C in LB broth. The cultures were subsequently transferred into fresh LB broth with chemical treatment the following day. After 6 h of treatment, OD600 of bacterial cultures was measured using Biotek Synergy H1 to assess the bacterial viability.

## Active RhoA detection

The detection of GTP-RhoA level is based on the specific binding of the Rho-binding domain (RBD) of Rhotekin to GTP-RhoA with the commercial active Rho detection Kit (#8820, CST). Briefly, GTP-RhoA in cell lysates is affinity-purified using Rhotekin-RBD/GST fusion protein-coated glutathione affinity beads. Then, Western blotting analysis with a RhoA-specific antibody quantifies the activated RhoA level in cells.

## Western blotting

Cell lysates were prepared by washing cells once with PBS and scraping them into RIPA lysis buffer (50 mM Tris pH 7.4, 150 mM NaCl, 1% Triton X-100, 1% sodium deoxycholate and 0.1% SDS) supplemented with 1 mM phenylmethanesulfonylfluoride or phenyl-methylsulphonyl fluorid (PMSF), 10 mM dithiothreitol (DTT), 40 µg/mL DNase I and 1 µg/mL each of leupeptin, pepstatin, and aprotinin, all conducted at 4 °C. Protein concentrations were determined using the BCA Protein Assay kit. Equal amounts of total cell lysates were mixed with Laemmli Sample Buffer (LSB), boiled, and separated on 12.5% SDS-PAGE gels. Proteins were transferred to nitrocellulose membranes using the Trans-Blot Turbo transfer system (Bio-Rad) following the Mini TGX gel transfer protocol. Membranes were blocked in 5% BSA for 1 h, followed by overnight incubation at 4 °C with primary antibodies: rabbit NF-κB p65 antibody (#8242, Cell Signaling, dilution 1:1000); rabbit phospho-NF-κB p65 (Ser536) antibody (#3033, Cell Signaling, dilution 1:1000); mouse myosin light chain antibody (#M4401, Sigma, dilution 1:1000); rabbit

Phospho-myosin light chain 2 (Thr18/Ser19) antibody (#3674, Cell Signaling, dilution 1:500); mouse vimentin antibody (#49636, Cell Signaling, dilution 1:1000); rabbit TLR4 antibody (#19811-1-AP, Proteintech, dilution 1:500); rabbit RhoA (67B9) antibody (#2117, Cell Signaling, dilution 1:1000); rabbit ARHGEF3 antibody (#TD4434, Abmart, dilution 1:1000); mouse GAPDH antibody (#G8795, Sigma, dilution 1:1000). Primary and secondary antibodies were diluted in fresh blocking buffer. Primary antibody incubated with PVDF membranes overnight at 4 °C and secondary antibodies incubated with PVDF membrane for 1 h at room temperature, respectively. Proteins bands were detected using SuperSigna West FemtoMaximum Sensitivity Substrate (Thermo Fisher Scientific).

## RNA sequencing and transcriptome analysis

Total RNA was extracted from monoclonal iBMDM with or without LPS stimulation using TRIzol™ Reagent (#15596026, Invitrogen) according to the manufacturer's instruction. RNA libraries were prepared according to the mRNA Library Preparation (DNBSEQ) BGI-NGS-JK-RNA-001 A0 protocol. Sequencing data were filtered with SOAPnuke (Li et al, 2008), which included: (I) Removal of reads containing sequencing adapter; (II) Filtering out reads with a low-quality base ratio (base quality less than or equal to 15) exceeding 20%; (III) Removal of reads containing more than 5% unknown base ('N' bases). Clean reads were obtained and stored in FASTQ format.

Subsequent analysis and data mining were performed on the Dr. Tom Multi-Omics Data Mining System (https://biosys.bgi.com). Bowtie was used to align the clean reads to the gene set, which included known and novel coding and noncoding transcripts (Langmead and Salzberg, 2012). The expression level of genes was calculated using RSEM (v1.3.1) (Li and Dewey, 2011). Differential expression analysis utilized DESeq2 (v1.4.5) with criteria set at |log2FoldChange| >0.5 (Love et al, 2014) and Q value ≤0.05 (or FDR ≤0.001). Visualization of gene expression differences across samples was facilitated by generating heatmaps with pheatmap (v1.0.8).

To elucidate biological significance, annotated differentially expressed genes underwent GO (Gene Ontology) (http://www.geneontology.org/) and KEGG (Kyoto Encyclopedia of Genes and Genomes) (https://www.kegg.jp/) enrichment analysis using Phyper (https://en.wikipedia.org/wiki/Hypergeometric_distribution) based on the hypergeometric test. Statistical significance of enriched terms and pathways was determined by Q value adjustment with a stringent threshold (Q value ≤ 0.05) (Cheng et al, 2004).

## Ethics statement

All animal experiments were conducted following institutional ethics requirements under the animal user permit (No. P2024011) approved by the Animal Care Committee of Shanghai Institute of Immunity and Infection, Chinese Academy of Sciences.

## Statistical analysis

Statistical analyses were performed by unpaired Student's *t* test, one-way analysis of variance or two-way ANOVA using the GraphPad Prism v9 software, and *p* values were indicated by *$P < 0.05$, or **$P < 0.01$, or ***$P < 0.001$, or ****$P < 0.0001$. The histogram data were presented as mean ± SD.

## Data availability

Bulk RNA-seq data have been deposited into the China National GeneBank DataBase (CNGBdb), with the accession number of CNP0007104 and the link https://db.cngb.org/search/project/CNP0007104. This study did not generate new codes. Any additional information required to reanalyze the data reported in this paper is available from the corresponding author upon request.

The source data of this paper are collected in the following database record: biostudies:S-SCDT-10_1038-S44318-025-00515-z.

## Peer review information

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

## Acknowledgements

This study was supported by National Natural Science Foundation of China (32222022, 92354301); Natural Science Foundation of Shanghai (23ZR1470900); R&D Program of Guangzhou National Laboratory (GZNL2023A03004); Key Research and Development Program, Ministry of Science and Technology of China (2024YFC2310003, 2022YFC2303502). We thank Dr. Pekka Lappalainen (University of Helsinki, Finland) and Dr. Weijun Pan (Chinese Academy of Sciences, China) for helpful suggestions on the study and comments on the manuscript. We thank the Medical Science Data Center of Fudan University for technical support.

## Author contributions

**Changyuan Fan**: Writing—original draft; Project administration.
**Xinyi Huang**: Formal analysis; Validation; Writing—review and editing.
**Jie Mei**: Methodology; Project administration. **Xuemeng Shi**: Project administration.
**Hao Zhang**: Project administration. **Cong Liang**: Project administration.
**Shuzhi Cui**: Validation; Visualization. **Yifan Xing**: Writing—review and editing.
**Biao Cao**: Validation. **Wei Liu**: Validation. **Huisheng Liu**: Visualization.
**Bo Liu**: Visualization. **Wakam Chang**: Validation. **Mengle Shao**: Visualization.
**Gong-Hong Wei**: Visualization. **Yan-Jun Liu**: Validation. **Zheng-Jun Chen**: Visualization. **Zhaoyu Lin**: Visualization; Writing—review and editing.
**Tao Xu**: Supervision. **Yaming Jiu**: Conceptualization; Data curation; Supervision; Funding acquisition; Investigation; Visualization; Writing—original draft; Writing—review and editing

Source data underlying figure panels in this paper may have individual authorship assigned. Where available, figure panel/source data authorship is listed in the following database record: biostudies:S-SCDT-10_1038-S44318-025-00515-z.

## Disclosure and competing interests statement

The authors declare no competing interests.

# Expanded View Figures

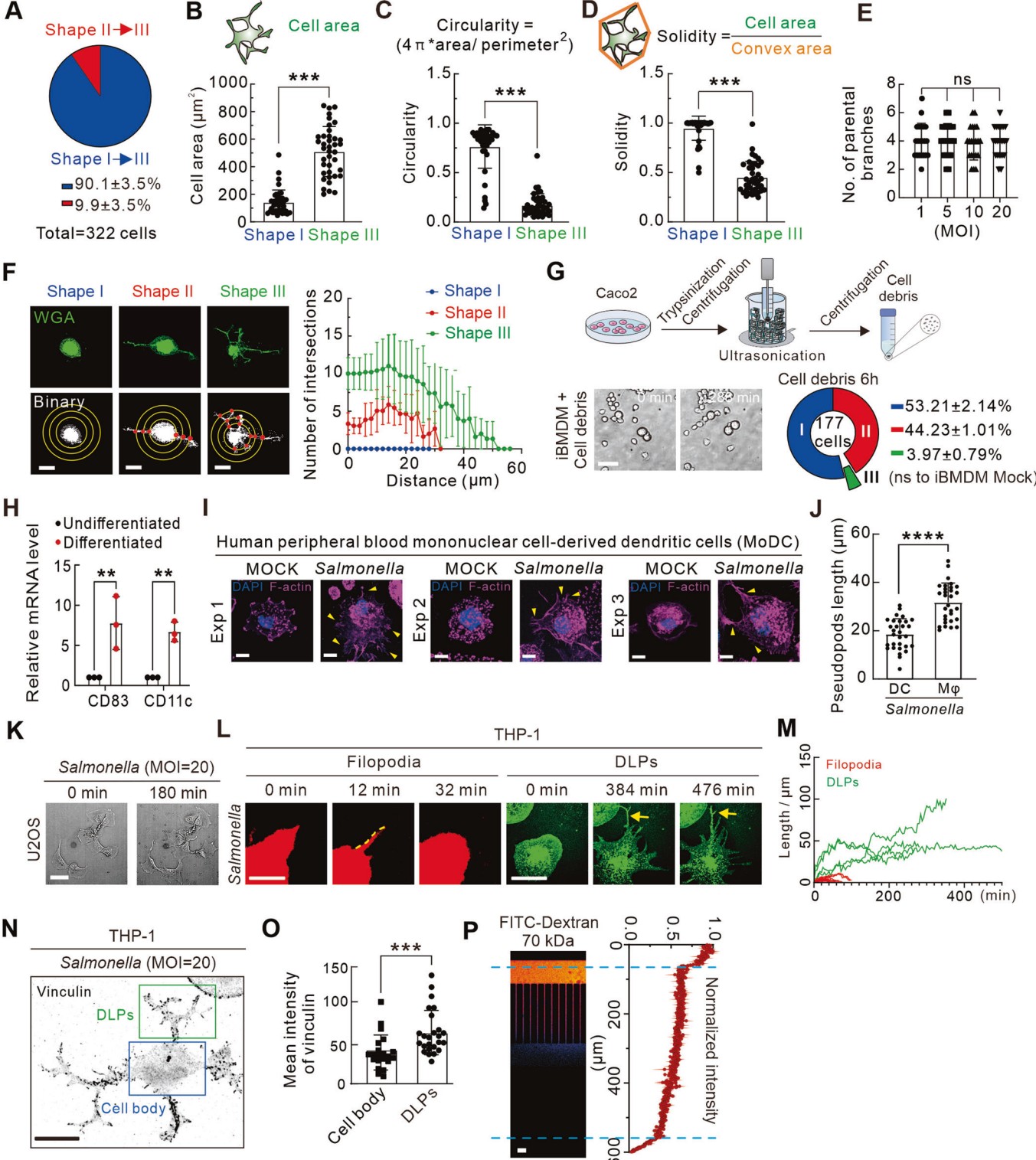

**Figure EV1. Macrophages construct dendrite-like pseudopods (DLPs) in response to bacterial infection.**

(A) Pie chart quantification of the percentage of shape III macrophages derived from shape I/II macrophages with Salmonella infection (MOI = 20). Blue, shape I to shape III; Red, shape II to shape III. The percentages presented in the form of mean ± s.d. (B-D) Quantification of cell area (B), circularity (C) and solidity (D) of shape I and shape III macrophages. *n* = 50 cells. (E) Quantification of the number of primary pseudopods derived from THP-1 cell body upon *Salmonella* infection with distinct MOIs. (F) Representative images visualized by WGA in shape I, shape II and shape III THP-1 macrophages and processed by Binary and Sholl analysis. Scale bar, 20 µm. Right panel shows the quantification of the number of intersections along DLPs based on Sholl analysis. *n* = 20 cells. (G) Schematic diagram of cell debris extraction (upper panel) and the time-lapse images of iBMDMs stimulated with cell debris (lower left panel). Scale bar, 50 µm. Pie chart quantification of the percentage of three shapes of macrophages with cell debris stimulation for 6 h (lower right panel). The percentages presented in the form of mean ± s.d. (H) Quantifications of the transcriptional levels of dendritic cell markers (CD83 and CD11c) by RT-qPCR. (I) Representative images of MoDCs derived from Human peripheral blood mononuclear cell-derived dendritic cells with and without *Salmonella* infection (MOI = 20) for 6 h. Scale bar, 10 µm. (J) Quantification of DLPs length in *Salmonella* infected MoDCs and macrophages. *n* = 30 cells. (K) Representative time-lapse images of U2OS cells upon *Salmonella* infection (MOI = 20). Scale bar, 10 µm. (L) Immunofluorescence of THP-1 macrophages infected with *Salmonella* and stained with WGA. Scale bar, 5 µm (filopodia panel) and 20 µm (DLPs panel). (M) Quantification of filopodia and DLPs length upon *Salmonella* infection. (N) Invert confocal images visualized by vinculin staining in *Salmonella* infected THP-1 macrophages. Blue and green boxes depicting the region of cell body and DLPs, respectively. Bars, 20 µm. (O) Quantification of the mean intensity of vinculin in ROIs of cell body and DLPs in (N). *n* = 28 cells. (P) Representative fluorescent imaging revealing the diffusion of 0.5 mg/mL 70 kDa FITC-Dextran in the microchannel 6 h after loading. Line profile on the right panel illustrates the fluorescence intensity of FITC-Dextran along the microchannels. Scale bars, 60 µm. Data are presented as mean ± s.d. from three independent experiments. Dots in quantifications presented individual cells. $P = 2.38284E\text{-}18$ (B). $P = 2.16145E\text{-}24$ (C). $P = 2.74828E\text{-}25$ (D). *P* values from left to right (E): $P = 0.89574$, $P = 0.60961$, $P = 0.787850183$, $P = 0.51197$, $P = 0.67727$, $P = 0.78709$. *P* values from left to right (H): $P = 0.00233$, $P = 0.00128$. $P = 2.07762E\text{-}09$ (J). $P = 0.00093$ (O). ns $P > 0.05$; **$P < 0.001$; ***$P < 0.001$: unpaired two-tailed Student's *t* test (B–D, G, J). one-way/two-way ANOVA with Sidak's analysis (E, H). Source data are available online for this figure.

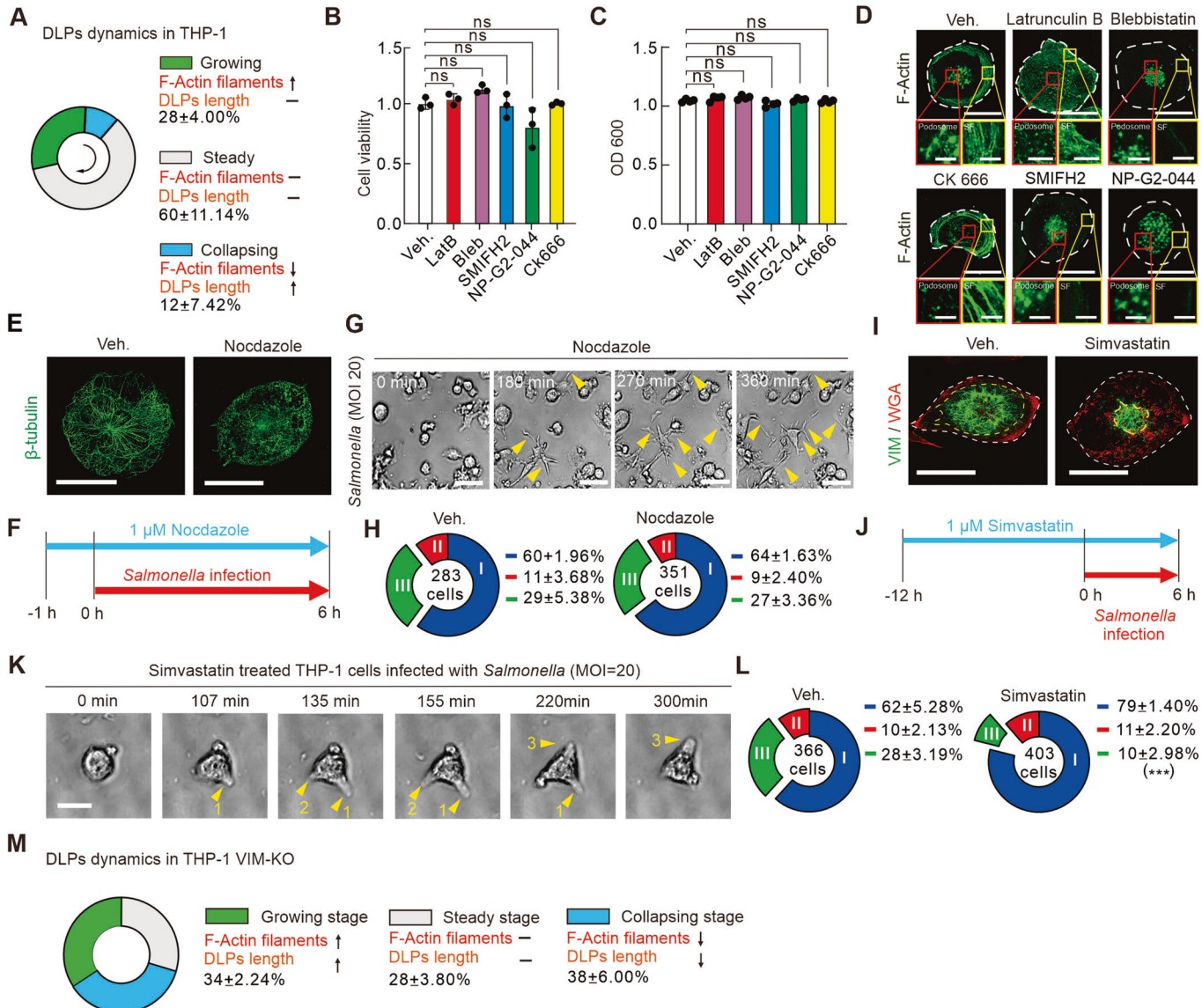

**Figure EV2. DLPs formation is dependent on cytoskeleton.**

(A) Pie chart shows the percentage of time spent in each stage of DLPs in wild-type THP-1 macrophages. (B) Quantification of the cell toxicity of the actin inhibitors. (C) Quantification of bacterial growth rate following the treatment of the actin inhibitors. (D) Representative images of actin filaments visualized by phalloidin in Veh., 0.5 μM Latrunculin B (LatB), 10 μM Blebbistatin, 40 μM CK666, 15 μM SMIFH2 and 1 μM NP-G2-044 treated THP-1 macrophages, respectively. Red and yellow magnified regions represent the podosome and contractile stress fibers (SF), respectively. Scale bars, 10 μm (in cell images) and 5 μm (in the magnified images). White dash lines outline cell contours. (E) Effects of 1 μM Nocodazole treatment on THP-1 macrophages for 1 h. The endogenous microtubule network was visualized by β-tubulin antibody staining. Scale bars, 20 μm. (F) Diagram of the Nocodazole treatment procedure. (G) Time-lapse images of THP-1 macrophages during *Salmonella* infection for 6 h with Nocodazole treatment (MOI = 20). Scale bars, 20 μm. (H) Pie charts show the percentage of three shapes in macrophages infected with *salmonella* for 6 h. Veh. $n = 283$ cells; Nocodazole treatment, $n = 351$ cells. (I) Representative images of endogenous vimentin in vehicle and 1 μM simvastatin treated cells. White and yellow dashed lines depict the regions of cell outline and vimentin network, respectively. Scale bars, 10 μm. (J) Schematic diagram of simvastatin treatment procedure. (K) Time-lapse images of THP-1 macrophages during *Salmonella* infection for 6 h with simvastatin treatment (MOI = 20). Scale bars, 20 μm. Yellow arrowheads depict DLPs, while the hollow arrowheads depict the diminishing DLPs. (L) Pie charts show the distribution of macrophage shapes following vehicle and simvastatin treatment with *salmonella* infection for 6 h. Veh., $n = 366$ cells; simvastatin treatment, $n = 403$ cells. (M) Pie chart shows the percentage of time spent in each stage of DLPs in VIM KO THP-1. Data are presented as mean ± s.d. from three independent experiments. $P = 0.00040$ (L). ns $P > 0.05$. ***$P < 0.001$: unpaired two-tailed Student's $t$ test (B, C, H, L). Source data are available online for this figure.

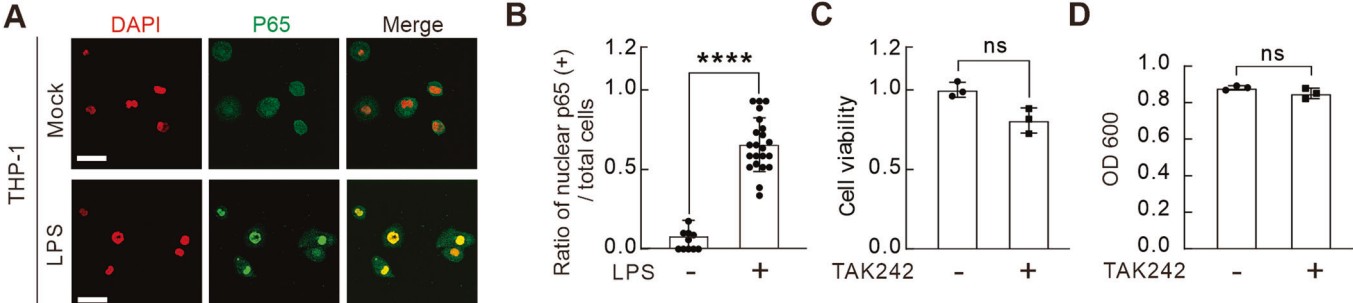

**Figure EV3.  TLR4 is a key factor for DLPs formation.**

(A) Representative images show the localization of NF-κB/P65 in THP-1 macrophages under mock or LPS-treated conditions. Scale bar, 20 μm. (B) Quantification of the nuclear P65 ratio relative to total cell count in (A). (C) Quantification of cell toxicity followingTAK-242 treatment. (D) Quantification of bacterial growth with TAK-242 treatment. Data are presented as mean ± s.d. from three independent experiments. Dots in quantifications presented individual cells. $P = 2.27158\text{E-}07$ (B). ns $P > 0.05$; ****$P < 0.0001$: unpaired two-tailed Student's *t* test (B–D). Source data are available online for this figure.

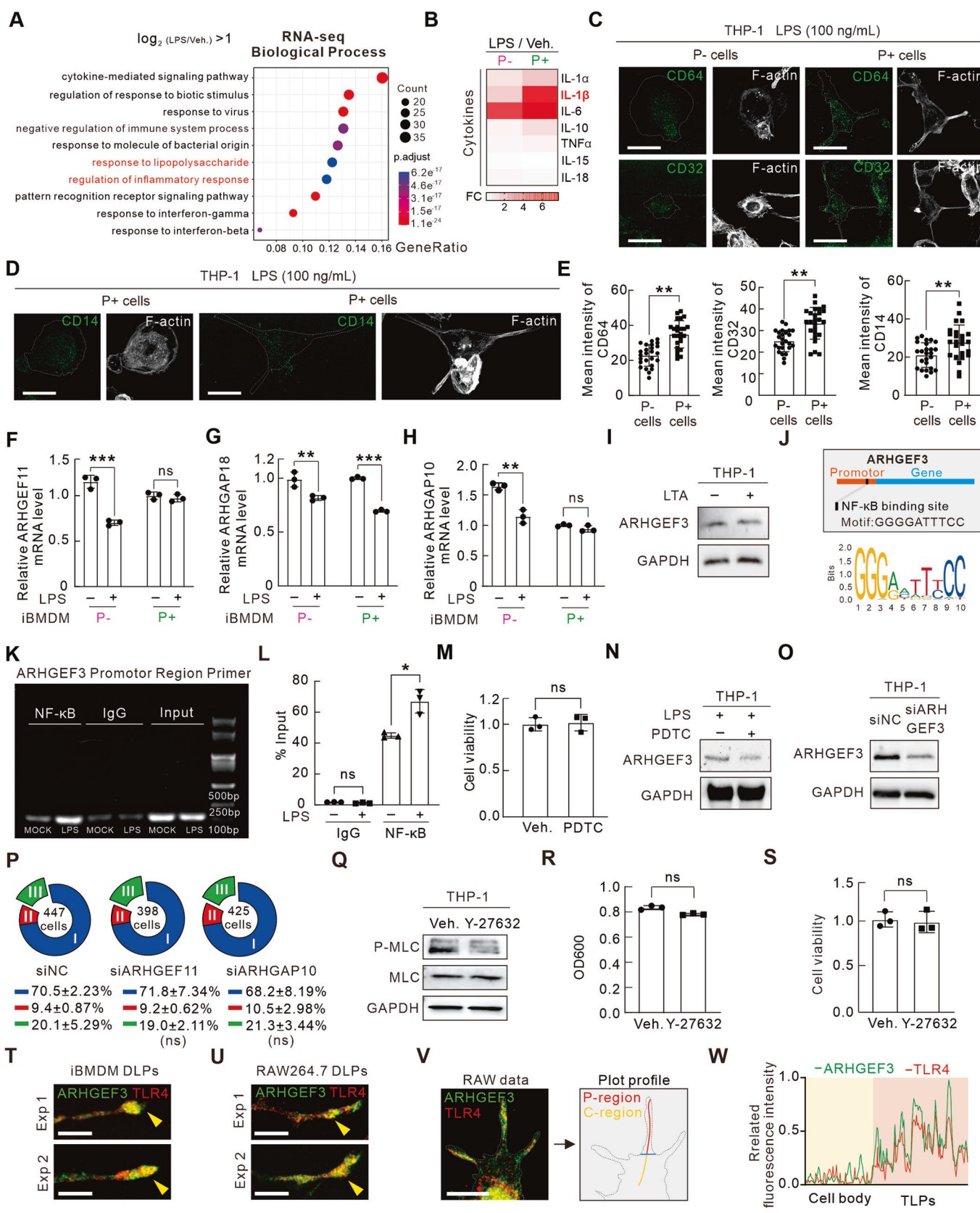

**Figure EV4.   ARHGEF3 and RhoA is essential in DLPs formation.**

(A) Biological process enrichment analysis of significantly upregulated genes in P + /P− iBMDMs challenged with LPS (count with average TPM). The top 10 enriched items are shown in the bubble chart. The colored bubbles from red to blue indicates the descending order of *P*. adjust value. The sizes of the bubbles are displayed from small to large in ascending order of gene counts. The *x* and *y* axis represent the gene ratio and the GO terms, respectively. (B) Heatmap shows the transcriptional levels of inflammatory cytokines by RNA sequencing. (C) Representative images showing the F-actin visualized by phalloidin and macrophage surface markers CD64 and CD32 in P − and P+ iBMDMs with LPS treatment for 6 h. Bars, 10 μm. (D) Representative images showing the F-actin visualized by phalloidin and macrophage surface markers CD14 in P− and P+ iBMDMs with LPS treatment for 6 h. Bars, 10 μm. (E) quantification of the mean intensity of macrophage surface markers CD64, CD32, and CD14 in P− and P+ iBMDMs in (C) and (D). (F–H) Quantifications of ARHGEF11, ARHGAP18, ARHGAP10 transcriptional levels by qRT-PCR in P- and P+ iBMDMs upon LPS stimulation. (I) Western blot analysis of ARHGEF3 upon LTA treatment. (J) Schematic diagram showing predicted NF-κB binding site in the ARHGEF3 promoter. (K, L) RT-qPCR analysis of DNA pulled down with IgG or NF-κB antibody upon LPS stimulus. (M) Quantification of the cell toxicity upon PDTC treatment. (N) Western blot analysis of ARHGEF3 upon LPS, or combined LPS and PDTC treatment. (O) Western blot analysis demonstrating ARHGEF3 knockdown efficiency in THP-1 macrophages. (P) Pie charts show the percentage of three shapes in macrophages with RNAi treatment of negative control, ARHGEF11 and ARHGAP10 and with *Salmonella* infection for 6 h. (Q) Western blot analysis of pMLC and MLC upon Y-27632 treatment in THP-1 macrophages. (R) Quantification of bacterial growth rate upon Y-27632 treatment. (S) Quantification of cell toxicity upon Y-27632 treatment with *Salmonella* infection for 6 h. (T, U) Representative images of DLPs in iBMDMs (T) and RAW264.7 (U) stimulated by LPS and stained with ARHGEF3 and TLR4 antibodies. Yellow arrows depict the DLPs. (V) The distribution of ARHGEF3 and TLR4 from DLPs to cell body. P-region, the specific pseudopods region; C-region, Cell body region. (W) Line profiles of related fluorescence intensity of ARHGEF3 and TLR4 in (V). Data are presented as mean ± s.d. from three independent experiments. Dots in quantifications presented individual cells. *P* values from left to right (E): $P = 0.00288$, $P = 0.00189$, $P = 0.00425$. *P* values from left to right (F): $P = 0.00054$, $P = 0.45742$. *P* values from left to right (G): $P = 0.00150$, $P = 0.00067$. *P* values from left to right (H): $P = 0.00243$, $P = 0.14849$. *P* values from left to right (L): $P = 0.39404$, $P = 0.03801$. ns $P > 0.05$; *$P < 0.05$; **$P < 0.01$; ***$P < 0.001$: one-way ANOVA with Sidak's analysis (P), two-way ANOVA with Sidak's analysis (F–H, L), unpaired two-tailed Student's *t* test (E, M, R, S), Benjamini–Hochberg (BH) analysis (A). Source data are available online for this figure.

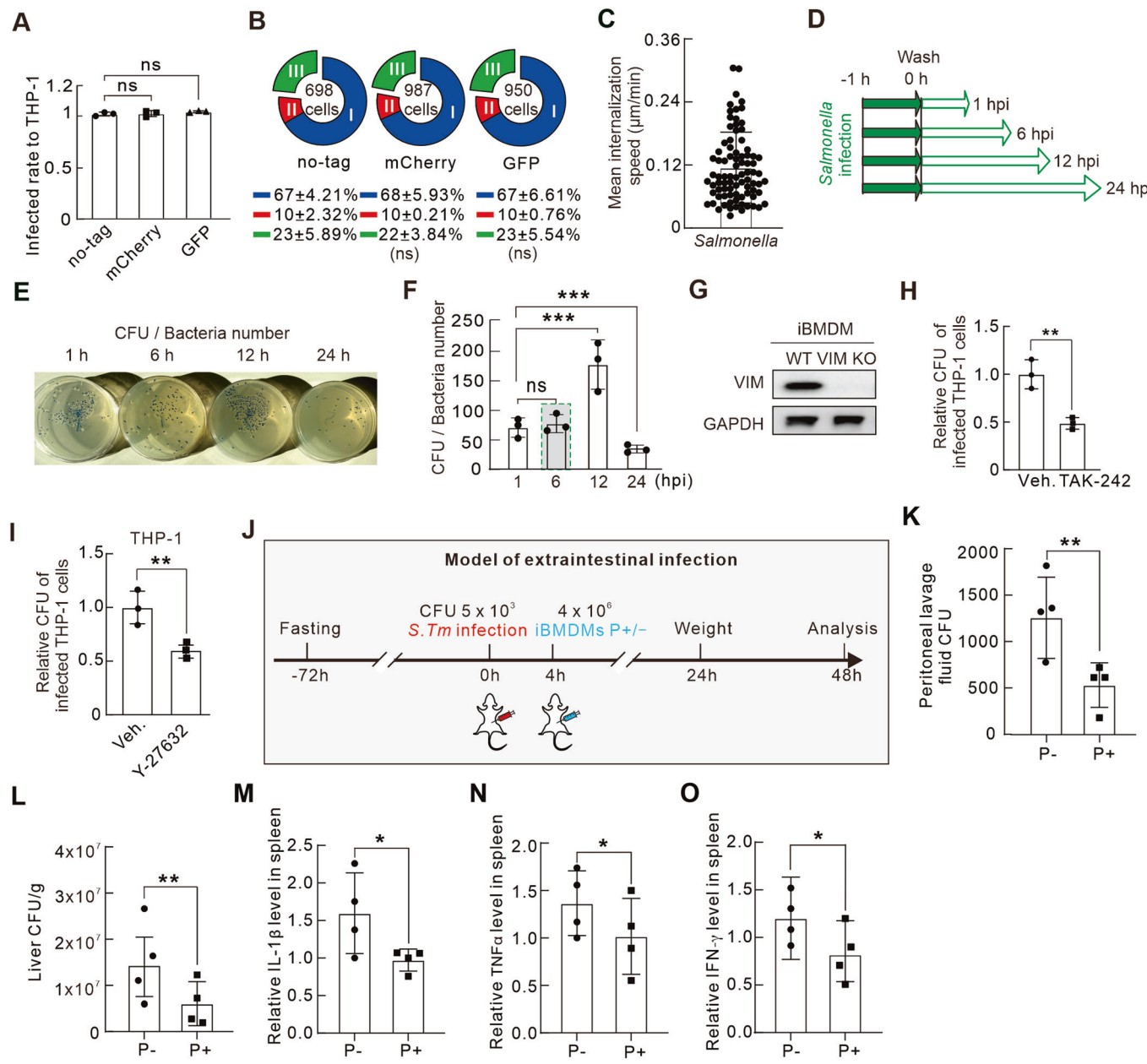

**Figure EV5.  Cytoskeleton and related signaling are essential in DLPs formation.**

(A) Quantification of infection rates in THP-1 macrophages with fluorescence-tagged or non-tagged *Salmonella*. (B) Pie charts show the percentage of three shapes of macrophages with *Salmonella* infection for 6 h in (A). (C) Quantification of the average speed of inward moving *Salmonella*. $n = 78$ cells. (D) Schematic diagram illustrating the CFU assay performed at distinct infection durations. (E) Representative images of the CFU assay at different time point post *Salmonella* infection of THP-1 macrophages. (F) Quantification of intracellular *Salmonella* over time in THP-1 macrophages. (G) Western blot analysis of vimentin in WT and VIM KO iBMDM. (H, I) Quantification of the bacterial load upon *Salmonella* infection in THP-1 macrophages treatment with TAK242 or Y-27632. (J) Schematic diagram of extraintestinal infection mouse model incorporating macrophage re-infusion. (K, L) Quantification of bacterial load indicated by colony forming unit (CFU) in peritoneal lavage fluid (K) and liver (L) infected with *Salmonella*. (M–O) Quantitative RT-qPCR analysis of the expression levels of inflammatory markers in the spleen. Data are presented as mean ± s.d. from three independent experiments. Dots in quantifications presented individual cells. *P* values from left to right (F): $P = 0.65388$, $P = 0.00094$, $P = 0.00094$. $P = 0.00558515$ (H). $P = 0.00124$ (I). $P = 0.00805$ (K). $P = 0.00243$ (L). $P = 0.03700$ (M). $P = 0.04913$ (N). $P = 0.04496$ (O). ns $P > 0.05$; *$P < 0.05$; **$P < 0.01$; ***$P < 0.001$: unpaired two-tailed Student's *t* test (H, I, K–O), one-way ANOVA followed by a Tukey's post hoc test (A, F). Source data are available online for this figure.

