## [Peer Review File · The EMBO Journal]

Macrophages form dendrite-like pseudopods to enhance bacterial ingestion

Changyuan Fan, Xinyi Huang, Jie Mei, Xuemeng Shi, Hao Zhang, Cong Liang, Shuzhi Cui, Yifan Xing, Biao Cao, Wei Liu, Huisheng Liu, Bo Liu, Wakam Chang, Mengle Shao, Gong-Hong Wei, Yan-Jun Liu, Zhengjun Chen, Zhaoyu Lin, Tao Xu, and Yaming Jiu

Corresponding authors: Yaming Jiu (ymjiu@siii.cas.cn) , Zhaoyu Lin (linzy@nju.edu.cn), Tao Xu (xu_tao@gzlab.ac.cn)

Review Timeline:

Submission Date:	16th Dec 24
Editorial Decision:	6th Feb 25
Revision Received:	7th Apr 25
Editorial Decision:	10th Jun 25
Revision Received:	24th Jun 25
Accepted:	10th Jul 25

Editor: Ieva Gailite

Transaction Report:

Dear Yaming,

Thank you for submitting your manuscript for consideration by the EMBO Journal. I sincerely apologise for the protracted review process for your manuscript due to delays in reviewer report submission. We have now received three referee reports on your manuscript, which are included below for your information.

As you will see from the comments, in particular reviewers #1 and #3 find the presented identification of macrophage pseudopod-like protrusions in response to high bacterial concentration novel and interesting. However, all reviewers also indicate multiple important concerns regarding the experimental approach, data analysis and presentation that would need to be addressed before they can support publication of the manuscript. In particular, all reviewers agree that analysis of ARHGEF3-depleted macrophages in the infiltration experiments would be needed. Furthermore, reviewer #2 finds that clearer evidence needs to be provided for the proposed role of the pseudopods in bacterial uptake. Additionally, reviewers ask for a better characterisation and definition of the DLPs and their porpoed specificity to macrophages.

Based on the interest expressed by the reviewers, I would like to invite you to address the comments of all referees in a revised version of your manuscript. Please note that a strong referee support will be required for the acceptance of the revised manuscript.

I should add that it is The EMBO Journal policy to allow only a single major round of revision and that it is therefore important to resolve the main concerns at this stage. I think that it would be helpful to discuss the revision in more detail via email or phone/videoconferencing. Please also let me know if you find that particular issues will not be addressable in the revised version, in which case I would be happy to discuss alternative publication possibilities at EMBO Press.

We generally allow three months as standard revision time, which can be extended to six months in the case of major revisions. Should you foresee a problem in meeting this deadline, please let us know in advance to discuss an extension. As a matter of policy, competing manuscripts published during this period will not negatively impact on our assessment of the conceptual advance presented by your study. However, please contact me as soon as possible upon publication of any related work to discuss the appropriate course of action.

When preparing your letter of response to the referees' comments, please bear in mind that this will form part of the Review Process File and will therefore be available online to the community. For more details on our Transparent Editorial Process, please visit our website: <https://www.embopress.org/page/journal/14602075/authorguide#transparentprocess>. Please also see the attached instructions for further guidelines on preparation of the revised manuscript.

Please feel free to contact me if have any further questions regarding the revision. Thank you for the opportunity to consider your work for publication, and I look forward to discussing your revision with you.

With best wishes,

Ieva

We realize that it is difficult to revise to a specific deadline. In the interest of protecting the conceptual advance provided by the work, we recommend a revision within 3 months (7th May 2025). Please discuss the revision progress ahead of this time with the editor if you require more time to complete the revisions.

Referee #1:

In the accompanying manuscript by Fan et al, the authors identify the appearance of macrophages that exhibit dendritic-like pseudopods in the presence of high numbers of bacteria, such as Salmonella. This is an interesting finding and it is surprising that these shapes have not been observed before. The authors carefully describe the dynamic appearance of these dendritic-like pseudopods in macrophages from various sources and across numerous datasets. Mechanistically, they identify the dependence of dendritic-like pseudopods on high numbers of gram-negative bacteria, TLR4 receptor signalling, the vimentin intermediate filaments, the linear actin cytoskeleton, as well as myosin, RhoA, and ARHGEF3. Overall, this manuscript is very well-written and reports an interesting finding relevant to the fields of macrophage biology, cell shape, and the cytoskeleton. Therefore, it would be a suitable fit for this journal, provided that the authors address the following aspects, including critical controls, quantifications, and statistical analysis.

Major points:

(i) Macrophages typically have elongated and branched cell types even before a bacterial infection (see e.g., see images from the intravital imaging in Fig.1d, mock). Although it is very clear from the entire manuscript that cells with dendritic-like pseudopods appear upon high numbers of bacteria, it is surprising to see that the cells - from which they appear before infection - are mostly very round throughout the datasets shown in the manuscript (e.g. Fig. 1g, l; Fig. 2f; etc...). Is this surprising round shape a specific property of the imaging slides that the authors used? For instance, does the adhesiveness of the microenvironment play a role in the appearance of macrophages with dendritic-like pseudopods? Would macrophages with a more natural elongated and branched shape also develop additionally dendritic-like pseudopods? The authors should also mention this aspect in the manuscript.

(ii) The percentage of macrophages with dendritic-like pseudopods in the entire population of cells is around 30% in most of the results. Are these cells a specific subpopulation of macrophages within the population? Could the authors test this by using macrophage surface markers in combination with immunofluorescence stainings?

(iii) Furthermore, the occurrence of cells exhibiting dendritic-like pseudopods is only around 30% of cells in the population at a MOI of 20 or higher. In contrast, an MOI of 5 only results in this shape in about 10% of the cells. Is an MOI of 20 physiologically

relevant?

(iv) The authors claim that the appearance of dendritic-like pseudopods is a specific reaction in macrophages and does not appear in other cell types. This argument is mostly based on experiments with THP-1-derived dendritic cells (Extended Data Fig. G-i). However, these data also show the appearance of protrusions in these cells, albeit shorter. While these protrusions have not exactly the same shape as in macrophages, it cannot be concluded from these data that dendritic-like pseudopods are not relevant to dendritic cells. In particular, THP-1-derived dendritic cells may also not be an ideal model for physiologically appearing dendritic cells. Thus, the authors should at least dampen their interpretation.

(v) Figure 3 characterises the cytoskeletal dependence and the dynamics of the dendritic-like pseudopods. However, the images on the dynamics shown in Figure 3 d and j - which are also the basis for quantifications shown in Figure 3d and k - show protrusion types with a wide non-branched shape that is entirely different from the described thin and branched dendritic-like pseudopods in the rest of the manuscript (see. e.g. Fig 3a). This is surprising, as it suggests that Figure 3 is based on the characterisation of a different protrusion type other than dendritic-like pseudopods, questioning the results from these datasets.

(vi) Statistics: many of the results are depicted in cake-like graphs (e.g. Fig. 1h, 2g, 3f,h etc...) that nicely illustrate the percentage of cells with dendrite-like pseudopods. However, they all lack information about statistical variance (error bars) and significance (statistical tests). It will be critical to add this information to all of these graphs of the manuscript to be able to judge the variability in cell shape types between different experiments (at least 3 replicates).

Minor points:

(i) Intravital imaging of macrophage shapes: it remains unclear from the text and methods how the shapes were defined as having pseudopods or not (Fig. 1e). Further, were these shapes manually quantified or automatically in a non-biased manner? The latter would be particularly important, given the image resolution shown in Fig.1d, which makes it difficult to distinguish individual cells and thus may cause a bias in manual quantification.

(ii) The data on the shapes of macrophages upon their exposure to cellular debris from Caco2 colonic epithelial cells lack a quantification (Extended Fig. 1F).

(iii) Single images of the appearance of cells with dendritic-like pseudopods in macrophages of different sources are shown in Figure 1I. This is an important dataset but lacks a quantification. Ideally, the quantification should be performed over time, as it is surprising that very specific time points (e.g., 62 min for BMDMs and 174 min for RAW cells) are shown.

(iv) The results from the microfluidic setup (Fig. 2c, d) need a quantification beyond the length of dendrite-like pseudopods: the number of dendrite-like pseudopods in mock vs. Salmonella conditions.

(v) Figure 3: definition of growing and collapsing stages: to this reviewer, it is not clear how the growing and collapsing stages were defined, as neither the growth nor the collapse are really evident in the figures and movies. Moreover, it is surprising that the protrusions grow and collapse so slowly over multiple hours, as macrophages normally can extend and retract protrusions within a few minutes.

(vi) Figure 3F: This dataset lacks a control (solvent of inhibitors; e.g. DMSO).

(vii) Figure 3J and I lack a control (WT control to VIM KO). Along this line: do VIM KOs already have an altered shape before the infection with Salmonella?

(viii) Figure 4A: this critical dataset lacks a quantification, which be essential to see the effects of gram-negative and -positive bacteria.

(ix) Figure 4h, i: also these datasets lack a quantification.

Referee #2:

The manuscript by Changyuan Fan et al., describes membrane protrusions induced by Gram-negative bacteria and LPS in macrophages. The authors use a lot of different approaches to investigate the signaling axis implicated in this process and identify a RhoGEF, ARHGEF3, acting upstream of NF- κ B.

The authors present a study that looks compelling, with a lot of experiments that rely on multiple approaches (both in vitro and in vivo, using different types of macrophages).

However, there is a major question that is never addressed concerning the invasive capacity of the bacteria used, which are Salmonella Typhimurium. These bacteria have been extensively studied as the archetypal invasive bacteria: they are known to inject in the host cell, via a secretion system, effectors that were described to remodel the actin cytoskeleton and induce membrane protrusions and ruffling. This large body of literature is totally ignored (work from the laboratory of Jorge Galan for

instance).

In addition, the structures described seem to have several shapes depending on the experiments, from filopodia-like structures, to large membrane extensions and cone-shaped structures (figure 3).

The timing of the phenomenon observed is also unclear, as bacteria often induce quick membrane remodeling, while the process described here requires de novo transcription.

There is no proof that the described structures are involved in bacterial uptake. The two populations, P- and P+, used in vivo (figure 6), have distinct properties that could explain differences in bacterial survival.

In addition, there are major concerns regarding how data from multiple experiments are presented, as well as lack of controls and quantifications in several instances. Methods sometimes unclear or missing. English needs some work.

Additional major concerns:

1. General comments:

- Figures for which a quantification of the observed phenotype over a population of cells should be provided: Figure 1I, Figure 3c-d, Figure 4a, Figure 4h-i.
- For many figures, the legend states "n= x cells": from how many independent experiments do they come from? Statistics lacking in all the pie charts with percentages. Ideally, the percentages from each experiment should be displayed and statistical tests should be done on them.
- Figure for which controls (untreated cells) are missing: Figure 3f (inhibitors could have an effect on macrophage shape even in the absence of Salmonella infection), Figure 3h (VIM KO could have an effect on macrophage shape even in the absence of Salmonella infection), Figure 4k-l (TLR4 KO could have an effect on macrophage shape even in the absence of Salmonella infection), Figure 4o (inhibitor could have an effect on macrophage shape even in the absence of Salmonella infection).

2. Specific questions/comments:

- Supplementary Figure S2a-c could be in the main figure, as it is the quantification corresponding to the observations made in the images shown in Figure 2c.
- Figure 2c: Could the authors provide a quantification of the percentage of pseudopods extending in the upper vs lower channel in mock vs Salmonella conditions?
- line 278: "Collectively, these findings suggest that cyclic assembly and disassembly of bundled actin and vimentin are crucial for DLPs morphogenesis." The cyclic aspect was only observed here, whether or not it is crucial was not tested.
- Figure 5a: why a first round of LPS stimulation here? Why not seed the cells directly in the 96-well plate to establish the different clones? The two rounds of LPS stimulation might affect gene expression. In addition, what specific criteria were used to identify the P- vs P+ colonies? Was it based on the percentage of cells that adopt the type III morphology after LPS stimulation? If so, what was the threshold?
- Supplementary Figure S5c: could be in the main figure?
- Figure 6o-s: The authors conclude from the re-infusion experiment that DLPs help control bacterial infection; however, P- and P+ macrophages differ not only in their ability to form DLPs but also in other regards, as demonstrated by the RNA-seq analysis in Figure 5. Therefore, the conclusion should be more nuanced to reflect the possibility that other mechanisms might be at play to explain why the re-infusion of P+ macrophages is more efficient than the one of P- macrophages at controlling the infection. Alternatively, to demonstrate that DLPs are definitely involved, the authors could perform re-infusion experiments with macrophages silenced for ARHGEF3, if they can also show that the silencing does not affect some key parameters such as pro-inflammatory cytokine production.

3. Methods that are unclear or missing:

- The methods don't indicate how the bacteria were cultured.
- The methods for LPS and LTA stimulation are missing from the Material and Methods section. Especially, the duration of the stimulation is not clear in the legend of Figure 4.
- WGA staining is not described in the Material and Methods section.
- It is not clear in the Material and Methods how GTP-RhoA levels were assessed by western blot, was it with an antibody specific to the GTP-bound form of the protein?
- line 676 (bacterial infection), paragraph starting with "For infection of THP-1, RAW264.7 and iBMDMs macrophages..." How the infection was performed is not clear. How long were the macrophages exposed to the bacteria before being cultured with gentamicin?
- line 747 (transfection): How was the transfection performed? Using what reagent?
- line 821 (co-immunoprecipitation): How were iBMDMs stably transfected?
- line 867 (CRISPR KO): How were THP-1 and iBMDMs transfected? Using what reagent?
- line 904 (luciferase assay): More details on the experimental procedure should be provided regarding the cloning of the promoter of ARHGEF3.
- TIM: How was the 3D reconstruction done?
- Figure legends: It is not always clear for how long were the infections done.

Other concerns:

1. Inaccuracies:

- Figure 1d: It is not mentioned in the legend why some macrophages are green in the 3D reconstruction.

- Figure 5p: in the text, refers to a co-IP experiment, but actually microscopy images in the figure (5p and 5r seem to be inverted)
- line 436: "indicating that TLR4 also guides the spatial localization of ARHGEF3, thereby promoting DLPs formation" Could also be the other way around.
- line 655 (differentiation of human macrophages): "These monocytes were plated in tissue culture plates with 25% medium conditioned by recombinant mouse M-CSF (carrier-free, 20 ng/mL)" Why mouse M-CSF to differentiate human cells?
- In the legend of Figure 1, Figure 3 and Figure 4, the authors state: "Data are presented as mean {plus minus} s.d. from three independent experiments." This statement is not correct for most of the graphs presented in those figures, where dots represent individual cells.
- line 1321 (Figure 3 legend): Were the macrophages analyzed in Figure 3h infected with Salmonella? This information should be stated in the legend.
- line 1371: "Schematic diagram shows that NF- κ B binds to ARHGEF3 promoter." That is incorrect, the diagram shows that there is a predicted NF- κ B binding site in the ARHGEF3 promoter.
- There are 7 supplementary videos in the supplementary figure legends but only 6 files. The file for supplementary movie 6 fits the legend of supplementary video 7.

2. writing style:

- Introduction is sometimes unnecessarily wordy.
- line 72: "morphogenetic remodeling paradigms"? Does the term "morphogenetic" really apply in this context? Maybe "morphological" instead?
- line 165: "and exposed it to macrophages" should be corrected to "and exposed macrophages to them"
- line 151: "There results indicate that an increase in..."
- line 207: "while excluded THP-1" should be corrected to "while excluding THP-1"
- line 342: "After a 24-hour recovery interval, single-cell colonies were established by sorting individual macrophages into 96-well plates, followed by enrichment in the second round of LPS stimulation, colonies with significant variance in their capacity for DLPs production were identified as 19 P- colonies with minimal DLPs and 5 P+ colonies with robust DLPs formation." Clarity would be improved by breaking this sentence into two.
- line 381: "implicating" should be corrected to "suggesting that"
- line 400: "suggesting that NF- κ B is the key cause of LPS-TLR4 to boost ARHGEF3 expression" should be corrected to "suggesting that NF- κ B is the key factor through which LPS-TLR4 boosts ARHGEF3 expression"
- line 415: "inhibition of RhoA signaling using Y-27632, an inhibitor of..."
- line 789 (mice infection): Only male mice were analyzed for the re-infusion experiments, why?
- line 794: What does "partially euthanized" mean?
- line 894: "The following drugs were used at a specified doses and times"
- line 943: "Primary and secondary antibodies were diluted in fresh blocking buffer for overnight at 4{degree sign}C and for 1 h at room temperature, respectively."
- line 949: "iBMDM monoclonals" should be replaced by "monoclonal iBMDMs"
- Title of Fig. 1 should be: "Macrophages form DLPs in response to a bacterial challenge"
- line 1253: "from three separate sets of experimental mice" Do the authors mean from three experiments?
- line 1299 (Figure 1 legend): "Schematic diagram illustrates that macrophages directional growth requires the contact with bacteria."
- There are sentences written in the present tense in most figure legends, which are a bit unnatural to read (for example, "Representative images visualize the membrane..."). These sentences should be corrected to use the gerund (for example, "Representative images showing the membrane...") or to avoid the use of a verb altogether.
- line 1296: "membrane pore" should be plural.
- lines 1313 and 1329: "stage" should be plural.
- line 1335: "TLPs" should be DLPs.
- line 1370: "Western blot analysis of ARHGEF3 and the active levels of RhoA treated with LPS for indicated time in THP-1 macrophages" should be corrected to: "Western blot analysis of ARHGEF3 and the active levels of RhoA in THP-1 macrophages treated with LPS for indicated times".
- line 1378: "The diagram of RhoA-ROCK signaling pathway" should be corrected to: "Diagram of the RhoA-ROCK signaling pathway".

Referee #3:

This manuscript presents evidence for a previously undescribed response of macrophages to infection with Gram-negative bacteria. While many studies have reported that macrophages rapidly take up bacteria via "capture" by filopodia and subsequent engulfment in actin-rich phagocytic cups, here the authors describe a later, more prolonged response in which cells extend long, branched, relatively stable protrusions that they call dendrite-like pseudopods (DLPs). Unlike filopodia or lamellipodia, which

are entirely actin-based, DLPs contain a core of vimentin which appears to enhance their stability. Mechanistically, the authors describe a signaling cascade in which the bacterial lipopolysaccharide (LPS) activates the Toll-like receptor TLR4, inducing expression of the RhoA activating protein ARHGEF3 in an NFkB-dependent manner, all of which is necessary for DLP formation. Physiologically, they show that peritoneal injection of DLP-proficient macrophages in mice enhances clearance of peritoneal Salmonella, relative to non-DLP proficient cells. Taken together, these observations describe a novel mechanism for the clearance of Gram-negative bacteria by macrophages.

In general the experiments are thorough and well controlled, the data are clear, statistically sound and support the conclusions drawn by the authors. I have only a few relatively minor concerns that should be addressed:

1. It is unclear whether bacteria are present during the entire (typically 6h) incubation period or the authors use a standard gentamicin treatment assay in which cells are exposed to a pulse of bacteria then treated with gentamicin to kill remaining extracellular bacteria. The reason this is important is that the authors infect at an MOI of 20, but bacteria in the nutrient-rich medium would be expected to proliferate dramatically over a 6h incubation period.
2. It should be stated clearly in both the abstract and the text that this is quite a late response to LPS exposure, much later than the formation of filopodia or lamellipodia that are much more rapid (within minutes) and are driven by Cdc42 or Rac1 respectively, as opposed to RhoA.
3. It is unclear why the authors segregated cells into responder vs. non-responder (DLP-producing vs. non-producing) clonal cell lines prior to RNA-seq analysis as opposed to simply treating cells with LPS or not. Why was this necessary?
4. A related point - the authors used these clonal cell lines for their in vivo infection model, injecting responders vs. non-responders into the peritoneum after Salmonella infection. While the data are consistent with the authors' interpretation that the ability to form DLPs enhances their ability to clear infection, there are literally hundreds of other genes that are differentially expressed that might also contribute to clearance. It would seem "cleaner" to use macrophages lacking ARHGEF3 (or not) to prove their point.

Referee #1:

In the accompanying manuscript by Fan et al, the authors identify the appearance of macrophages that exhibit dendritic-like pseudopods in the presence of high numbers of bacteria, such as Salmonella. This is an interesting finding and it is surprising that these shapes have not been observed before. The authors carefully describe the dynamic appearance of these dendritic-like pseudopods in macrophages from various sources and across numerous datasets. Mechanistically, they identify the dependence of dendritic-like pseudopods on high numbers of gram-negative bacteria, TLR4 receptor signalling, the vimentin intermediate filaments, the linear actin cytoskeleton, as well as myosin, RhoA, and ARHGEF3. Overall, this manuscript is very well-written and reports an interesting finding relevant to the fields of macrophage biology, cell shape, and the cytoskeleton. Therefore, it would be a suitable fit for this journal, provided that the authors address the following aspects, including critical controls, quantifications, and statistical analysis.

Response:

We thank you for a thorough reading of our manuscript and for your critical comments aiming to improve the quality of our study. We have taken these comments very seriously and have addressed all suggestions by additional experiments and text revisions. The point-by-point explicit explanations are listed below.

Major points:

(i) Macrophages typically have elongated and branched cell types even before a bacterial infection (see e.g., see images from the intravital imaging in Fig.1d, mock). Although it is very clear from the entire manuscript that cells with dendritic-like pseudopods appear upon high numbers of bacteria, it is surprising to see that the cells - from which they appear before infection - are mostly very round throughout the datasets shown in the manuscript (e.g. Fig. 1g, l; Fig. 2f; etc...). Is this surpassing round shape a specific property of the imaging slides that the authors used? For instance, does the adhesiveness of the microenvironment play a role in the appearance of macrophages with dendritic-like pseudopods? Would macrophages with a more natural elongated and branched shape also develop additionally dendritic-like pseudopods? The authors should also mention this aspect in the manuscript.

Response:

We respectfully acknowledge your observations and kind suggestions to improve the quality of our data. We agree with you that macrophages in general exhibit morphological plasticity and show diverse shapes (i.e., round, oval, fusiform, elongated slender, with needle-like filopodia, with protrusive flat lamellipodia, etc.). In the case of THP-1 monocyte-derived macrophages, we performed experiments and quantifications after 1-2 days of differentiation. At that time, we observed and characterized the cell morphology into three distinct

shapes: type I represents circular, elliptical, fleshy, and corpulent macrophages, although we coined them all as ‘round’ without strict subdivision; type II represents fusiformed, slender macrophages, although in some cases these cells have needle-like filopodia or protrusive flat lamellipodia; type III represents the dendrite-like pseudopods (DLPs) equipped macrophages, which we found in this study. Notably, we indeed observed more various irregular shape as differentiation went further (i.e., after 8 or 10 days). The DLPs-equipped macrophages can only be observed upon infection with a large number of bacteria. We have clarified this point in the revised manuscript.

For the concerns on adhesiveness, we performed vinculin staining (a key focal adhesion marker) in DLPs-equipped macrophages. Our results demonstrate that DLPs were enriched with focal adhesions (Figure R1-1 in below), indicating that their ability to anchor to the extracellular matrix and microenvironment adhesiveness may contribute to the formation of DLPs. We have added these data to Fig. EV1N,O and updated the corresponding text in the revised manuscript.

Figure R1-1. (A) Representative image of DLP-equipped THP-1 macrophages infected with *Salmonella* and stained with vinculin antibody. Scale bar, 10 μm . (B) Quantification of mean vinculin immunofluorescence intensity in cell bodies and DLPs, respectively. Data are presented as mean \pm s.d. from three independent experiments. *** $P < 0.001$: unpaired two-tailed Student's t -test.

For the concerns on original shape of macrophages that derived DLPs, by live-cell imaging tracking and quantification, we found that approximately 90% of DLPs emerged from type I round, unpolarized macrophages, while ~10% of DLPs emerged from type II fusiformed macrophages (Figure R1-2 in below). We have added these data to Fig. EV1A and updated the text in the revised manuscript.

Figure R1-2. Pie chart quantification of the percentage of the original shape of macrophages that derived DLPs upon *Salmonella* infection (MOI=20). Blue, from shape I to shape III; Red, from shape II to shape III. A total of 322 cells from three-independent experiments were used for quantification.

(ii) The percentage of macrophages with dendritic-like pseudopods in the entire population of cells is around 30% in most of the results. Are these cells a specific subpopulation of macrophages within the population? Could the authors test this by using macrophage surface markers in combination with immunofluorescence stainings?

Response:

We thank you for this insightful suggestion. To investigate whether DLPs-forming macrophages represent a distinct subpopulation, we performed comprehensive molecular characterization. First, we isolated P+ (with excessive DLPs formation) and P- (with minimal DLPs formation) macrophages, and characterized their differential gene expression patterns by RNA sequencing. The results showed that macrophage surface markers (CD14 and F4/80) and M1 macrophage-specific surface markers (CD64 and CD32) were significantly increased upon *Salmonella* infection in P+, but not P- cells (Fig. 5D). In direct response to your suggestion, we further validated these findings through immunofluorescence staining, confirming that infected macrophages forming DLPs indeed constitute a distinct subpopulation with unique surface marker expression profiles (Figure R1-3 in below). These results have been incorporated into the revised manuscript, supporting our conclusion that DLPs formation is associated with a specific macrophage subpopulation rather than a random occurrence.

Figure R1-3. (A-C) Left panels: representative images showing the F-actin visualized by phalloidin and macrophage surface markers CD64 (A), CD32 (B), and CD14 (C) in P- and P+ iBMDMs, respectively. Scale bars, 10 μ m. Right panels: quantification of the mean intensity of macrophage surface markers CD64(A), CD32(B), and CD14(C) in P- and P+ iBMDMs, respectively. Data are presented as mean \pm s.d. calculated based on all cells from three independent experiments. * $P < 0.05$; ** $P < 0.01$: unpaired two-tailed Student's t -test.

(iii) Furthermore, the occurrence of cells exhibiting dendritic-like pseudopods is only around 30% of cells in the population at a MOI of 20 or higher. In contrast, an MOI of 5 only results in this shape in about 10% of the cells. Is an MOI of 20 physiologically relevant?

Response:

We thank you for this kind suggestion. In cell infection models, the multiplicity of infection (MOI) reflects the average number of bacteria per host cell. In *in vivo* experiments, infection load is typically measured using colony-forming units (CFU). While direct quantitative correlations between these parameters are inherently challenging due to microenvironmental differences (e.g., immune cell recruitment and tissue barriers), our experimental design aligns with

established models of *Salmonella* pathogenesis.

For instance, *in vitro* macrophage infection studies (PMID: 38528181) routinely employed an MOI of 20 to mimic the high bacterial loads observed during invasive infections, as shown by electron microscopy (Figure R1-4A). This parameter appears physiologically plausible: murine models of systemic salmonellosis (PMID: 25121751) demonstrate that infection with 5×10^7 CFU of *Salmonella* leads to enterocyte microcolonies containing ~ 20 bacteria per host cell within 12 hours (Figure R1-4B). Although direct comparisons between MOI and CFU require caution, the similarity in bacterial densities (~ 20 bacteria per cell *in vitro* vs. *in vivo*) suggests that our model captures infection intensities relevant to both localized and systemic contexts.

Furthermore, the dose-dependent increase in DLPs formation (10% at MOI=5 vs. 30% at MOI=20) supports the notion that this response reflects a regulated biological threshold rather than a nonspecific overload. We have expanded the discussion in the revised manuscript to better contextualize these findings, emphasizing the biological plausibility of our approach, while acknowledging the limitations of extrapolating *in vitro* parameters to physiological scenarios.

Figure R1-4. (A) Representative image of intracellular bacteria obtained by electron microscopy. *Salmonella* infection at an MOI of 20 for 10h. Scale bar, 5 μm . (B) Representative confocal micrographs of cecum sections from infected C57BL/6 mice 18h post-infection, showing S.Tm-G⁺ (green), host cell nuclei (gray), and EpCAM (epithelial cell marker; yellow). Lu.=Lumen. Right row image shows blow-ups of boxed regions. Scale bars, 10 μm .

(iv) The authors claim that the appearance of dendritic-like pseudopods is a specific reaction in macrophages and does not appear in other cell types. This argument is mostly based on experiments with THP-1-derived dendritic cells (Extended Data Fig. G-i). However, these data also show the appearance of protrusions in these cells, albeit shorter. While these protrusions have not exactly the same shape as in macrophages, it cannot be concluded from these data that dendritic-like pseudopods are not relevant to dendritic cells. In particular, THP-1-derived dendritic cells may also not be an ideal model for physiologically appearing dendritic cells. Thus, the authors should at least dampen their interpretation.

Response:

We thank you for raising this important point. We have dampened the

interpretation in the revised manuscript as you kindly suggested. To further address this question, we performed additional experiments using human peripheral blood mononuclear cell-derived dendritic cells (MoDC), which represent a more physiologically relevant model than THP-1-derived DCs. We found that MoDCs indeed develop membrane protrusions upon infection. However, these structures were consistently shorter than macrophage DLPs, supporting the morphological distinction between these cellular responses (Figure R1-5 in below). These findings suggested that DLPs are a prominent feature of macrophages.

Figure R1-5. Representative images of phalloidin labeled actin filaments (magenta) and nuclei (blue) in *Salmonella*-infected MoDC (MOI=20) for 6 h. Scale bar, 10 μ m.

(v) Figure 3 characterizes the cytoskeletal dependence and the dynamics of the dendritic-like pseudopods. However, the images on the dynamics shown in Figure 3 d and j - which are also the basis for quantifications shown in Figure 3d and k - show protrusion types with a wide non-branched shape that is entirely different from the described thin and branched dendritic-like pseudopods in the rest of the manuscript (see. e.g. Fig 3a). This is surprising, as it suggests that Figure 3 is based on the characterization of a different protrusion type other than dendritic-like pseudopods, questioning the results from these datasets.

Response:

We sincerely appreciate your careful observation of the protrusion morphology shown in Figure 3. We apologize for the confusion caused by the representative live-cell imaging example chosen in Fig. 3C, which may not have shown typical dendritic branching characteristics. The actin network was particularly visible in this example with a wider protrusion, which led us to initially select it as a representative image. However, as you kindly suggested, we now realize that it may distract people from the authenticity of our results. Thus, we have replaced Fig. 3C with a new example demonstrating typical DLPs features and updated the related statistics to strengthen our conclusion in the revised manuscript (Figure R1-6 in below). Figure 3J shows the phenotype of vimentin depletion with sequential ‘budding and retracting’ protrusions without successful DLPs formation.

Figure R1-6. (A) Representative time-lapse images of SiR-actin labeled actin filaments in *Salmonella* infected THP-1 macrophages (MOI=20). Scale bar, 20 μm . Magnified views show the DLPs. The red lines depict actin filaments detected by ridged detection. Scale bar, 20 μm . (B) Quantification of the DLPs length and actin filament length over time in infected THP-1 macrophages in (A). Growing, steady, and collapsing stages are denoted by green, gray and blue, respectively.

(vi) Statistics: many of the results are depicted in cake-like graphs (e.g. Fig. 1h, 2g, 3f,h etc...) that nicely illustrate the percentage of cells with dendrite-like pseudopods. However, they all lack information about statistical variance (error bars) and significance (statistical tests). It will be critical to add this information to all of these graphs of the manuscript to be able to judge the variability in cell shape types between different experiments (at least 3 replicates).

Response:

We thank you for this kind suggestion regarding the statistical presentation of our data. We agree with you that the statistical variance and significance are critical for validating our data. In response, we have now incorporated measures of statistical variance (standard error) and significance testing into all relevant percentage-based graphs throughout the manuscript. Each figure now includes error bars representing the standard error, with statistical significance clearly indicated by asterisks. The specific p values and statistical tests used are detailed in the corresponding figure legends. An updated example (Figure 1H) is shown in Figure R1-7 below. We believe that these improvements significantly enhanced the statistical rigor and reproducibility of our findings.

Figure R1-7. Pie chart quantification of the percentage of the three macrophage shapes with or without *Salmonella* infection (MOI=20). The upper panels provide the definitions of the three protrusive structures. Blue, round shape (I); Red, fusiform shape (II); Green, deformed shape with DLPs (III). Data are presented as mean \pm s.d. calculated based on all cells from three independent experiments. *** $P < 0.001$.

Minor points:

(i) Intravital imaging of macrophage shapes: it remains unclear from the text and methods how the shapes were defined as having pseudopods or not (Fig. 1e). Further, were these shapes manually quantified or automatically in a non-biased manner? The latter would be particularly important, given the image resolution shown in Fig. 1d, which makes it difficult to distinguish individual cells and thus may cause a bias in manual quantification.

Response:

We thank you for raising this important question. Most mouse peritoneal macrophages labeled with F4/80 and observed utilizing two-photon microscopy adhered to the peritoneal wall and aligned with the ECM network, exhibiting sleek and slender morphology. We defined ‘with pseudopods’ as when a macrophage derived three or more protrusions from two sides of its elongated axis, and each protrusion was over five microns in length. Reciprocally, we defined ‘without pseudopods’ as when a macrophage showed less than two protrusions. The 3D overview in Fig. 1D was obtained from automated surface reconstruction in the Imaris 10.0 software (Oxford Instruments).

For quantification method, images from the intravital microscopy were difficult to automatically analysis by the software which we have tried. To ensure objectivity, we implemented a double-blind manual quantification protocol: one researcher anonymized and randomized the Mock and infected datasets, and a second researcher, blinded to the experimental conditions, performed all morphological classifications and calculations. This approach was necessary due to the limitations of the current software in reliably analyzing such complex intravital images, particularly given the resolution constraints evident in Fig. 1D.

We have added detailed descriptions of both the morphological criteria and the blinded quantification protocol to the Methods section of our revised manuscript to enhance the methodological transparency.

(ii) The data on the shapes of macrophages upon their exposure to cellular debris from Caco2 colonic epithelial cells lack a quantification (Extended Fig. 1F).

Response:

We thank you for this kind suggestion. We have supplemented this quantification in Fig. EV1G in the revised manuscript supports the conclusion that cell debris cannot induce DLPs formation.

(iii) Single images of the appearance of cells with dendritic-like pseudopods in macrophages of different sources are shown in Figure 1I. This is an important dataset but lacks a quantification. Ideally, the quantification should be performed over time, as it is surprising that very specific time points (e.g., 62 min for BMDMs and 174 min for RAW cells) are shown.

Response:

We thank you for this kind suggestion. The very specific time points indicate the duration of each representative live-cell imaging video. We have the representative time points, and the corresponding quantifications are shown in Figure R1-8 in below. We have added these data to Fig. 1L and updated the text in the revised manuscript.

Figure R1-8. Upper panels: representative time-lapse imaging reveals DLPs formation in iBMDM (A), RAW264.7 (B), and mouse peritoneal macrophages (C) during *Salmonella* infection (MOI=20). Scale bars, 50 μ m. Middle and lower panels: pie chart quantification of the percentage of three shapes of macrophages in iBMDM (A), RAW264.7 (B), and mouse peritoneal macrophages (C) with or without *Salmonella* infection (MOI=20). Blue, round shape (I); Red, fusiform shape (II); Green, deformed shape with DLPs (III). Data are presented as

mean \pm s.d. calculated based on all cells from three independent experiments. *** $P < 0.001$.

(iv) The results from the microfluidic setup (Fig. 2c, d) need a quantification beyond the length of dendrite-like pseudopods: the number of dendrite-like pseudopods in mock vs. *Salmonella* conditions.

Response:

We thank you for this kind suggestion. We have calculated and supplemented the corresponding results in Fig. 2F,G of the revised manuscript (Figure R1-9 in below).

Figure R1-9. (A) Pie chart quantification of the percentage of three macrophage shapes located in the junction between the upper vertical microchannel (*Salmonella* infection for 6 h) / the lower vertical microchannels (Mock) and the middle horizontal chamber. Blue, round shape (I); Red, fusiform shape (II); Green, deformed shape with DLPs (III). (B) The percentage of extending ratio of DLP-positive macrophages to microchannels. Data are presented as mean \pm s.d. calculated based on all cells from three independent experiments. *** $P < 0.001$.

(v) Figure 3: definition of growing and collapsing stages: to this reviewer, it is not clear how the growing and collapsing stages were defined, as neither the growth nor the collapse is really evident in the figures and movies. Moreover, it is surprising that the protrusions grow and collapse so slowly over multiple hours, as macrophages normally can extend and retract protrusions within a few minutes.

Response:

We apologize for the confusion regarding our description of the growing and collapsing stages. We have optimized the description in the main text of the revised manuscript as follows: “(1) *the growing stage*, characterized by rapid actin assembly into bundles to fill a just-grown nascent DLP, where DLP length remains unchanged while F-actin intensity gradually increases; (2) *the steady stage*, the longest period (~67% of the cycle) during which both the protrusive structure and the internal actin bundles remained stable; and (3) *the collapsing stage*, during which the DLPs extended further, accompanied by rapid disassembly of actin bundles indicated by decreased F-actin intensity.”.

Regarding the unusually slow kinetics (hours versus minutes), we speculate it may reflect the transcriptional regulation required for DLP formation. Our results indicate the expression and trans-localization of ARHGEF3 is critical to link bacterial stimulus to DLPs formation (Fig. 5). This distinguishes DLPs from rapidly-forming protrusions like filopodia that rely solely on pre-existing signaling cascades.

(vi) Figure 3F: This dataset lacks a control (solvent of inhibitors; e.g. DMSO).
Response:

We thank you for this kind suggestion. We have supplemented the control with the solvent of inhibitors in Fig. 3F of the revised manuscript. These results confirm that the solvent alone had no effect on the formation of DLPs. We apologize for not including this in the initial submission, which we indeed carried out these control experiments.

(vii) Figure 3J and I lack a control (WT control to VIM KO). Along this line: do VIM KOs already have an altered shape before the infection with *Salmonella*?
Response:

We apologize for the confusion in the interpretation of our results in the initial submission. The wild-type controls for Fig. 3J (VIM KO cells) are presented in Fig. 3C (actin dynamics) and Fig. 1G (morphology), as we aimed to maintain a logical flow by comparing both cytoskeletal networks. Importantly, we observed no significant morphological differences between wild-type and VIM KO THP-1 macrophages prior to infection, as shown in Figure R1-10 in below. This confirms that the altered protrusion dynamics observed in VIM KO cells following *Salmonella* infection (Fig. 3I,J) are specifically related to the infection response rather than baseline morphological changes. We included these control comparisons in the revised manuscript to provide complete context for our findings.

Figure R1-10. Representative images reveal the cell shape of WT and VIM KO THP-1 macrophages. Scale bars, 20 μm .

(viii) Figure 4A: this critical dataset lacks a quantification, which be essential to see the effects of gram-negative and -positive bacteria.

Response:

We thank you for this kind suggestion. We have now included the quantification in Fig. 4A (also Figure R1-11 in below), showing the comparative effects of Gram-negative and Gram-positive bacteria on DLPs formation. This analysis has been added in the revised manuscript.

Figure R1-11. Upper panels: representative images reveal DLPs formation in THP-1 macrophages infected with *Salmonella*, *E. coli*, *Shigella*, *Bacillus*, and *Listeria* (MOI=20), respectively. Lower panels: pie chart quantification of the percentage of three macrophage shapes in THP-1 macrophages with or without infection. Blue, round shape (I); Red, fusiform shape (II); Green, deformed shape with DLPs (III). Data are presented as mean ± s.d. calculated based on all cells from three independent experiments.

(ix) Figure 4h, i: also these datasets lack a quantification.

Response:

We thank you for this kind suggestion. We have now included quantitative analyses for Fig. 4J,K (also Figure R1-12 in below) which provide statistical validation of these results. These data have been added in the revised manuscript.

Figure R1-12. Left panels: representative time-lapse imaging reveals DLPs formation in iBMDM (A) and RAW264.7 (B) during LPS treatment. Scale bars, 20 μm . Middle and right panels: pie chart quantification of the percentage of three shapes of macrophages in iBMDM (A) and RAW264.7 (B) with or without LPS treatment. Blue, round shape (I); Red, fusiform shape (II); Green, deformed shape with DLPs (III). Data are presented as mean \pm s.d. calculated based on all cells from three independent experiments. *** $P < 0.001$.

Referee #2:

The manuscript by Changyuan Fan et al., describes membrane protrusions induced by Gram-negative bacteria and LPS in macrophages. The authors use a lot of different approaches to investigate the signaling axis implicated in this process and identify a RhoGEF, ARHGEF3, acting upstream of NF- κ B.

The authors present a study that looks compelling, with a lot of experiments that rely on multiple approaches (both in vitro and in vivo, using different types of macrophages).

However, there is a major question that is never addressed concerning the invasive capacity of the bacteria used, which are *Salmonella Typhimurium*. These bacteria have been extensively studied as the archetypal invasive bacteria: they are known to inject in the host cell, via a secretion system, effectors that were described to remodel the actin cytoskeleton and induce membrane protrusions and ruffling. This large body of literature is totally ignored (work from the laboratory of Jorge Galan for instance).

In addition, the structures described seem to have several shapes depending on the experiments, from filopodia-like structures, to large membrane extensions and cone-shaped structures (figure 3).

The timing of the phenomenon observed is also unclear, as bacteria often induce quick membrane remodeling, while the process described here requires de novo transcription.

There is no proof that the described structures are involved in bacterial uptake. The two populations, P- and P+, used in vivo (figure 6), have distinct properties that could explain differences in bacterial survival.

In addition, there are major concerns regarding how data from multiple experiments are presented, as well as lack of controls and quantifications in several instances. Methods sometimes unclear or missing. English needs some work.

Response:

We sincerely appreciate your thorough evaluation of our manuscript and your valuable suggestions aiming to improve the quality of our study. We have taken these comments very seriously, and carefully addressed all of your concerns through additional experiments and extensive revisions to the text.

For bacterial secretion system, we acknowledge that the effectors of *Salmonella's* type III secretion system remodel the actin cytoskeleton and induce membrane protrusions and ruffling during infection, as highlighted in the work of Dr. Jorge Galan and many other previous studies. We generated *Salmonella* strains lacking SPI-1 (SipB mutant) or SPI-2 (PhoP mutant) to investigate whether bacterial effectors were required for DLPs formation. These mutants, previously confirmed to have defective secretion functions in our work (PMID: 36717589), were employed to block effector injection. By infecting macrophages with these mutated bacteria, we observed that DLPs were still

formed, while the actin network was perturbed within the cytoplasm (Figure 2-1 in below). These results indicate that while bacterial effectors influence cytoplasmic actin rearrangement, they are not required for DLPs formation, indicating that DLPs formation represents a distinct cellular response mechanism.

Figure R2-1. Upper panels: representative time-lapse imaging reveals DLPs formation in THP-1 macrophages during *Salmonella* (Left: wild type, Middle: SipB mutant, Right: PhoP mutant) infection (MOI=20). Scale bars, 50 μ m. Lower panels: pie chart quantification of the percentage of three macrophage shapes in THP-1 macrophages infected with different *Salmonella* strains (MOI=20).

Concerning the wider unbranched shape in Fig. 3, we sincerely appreciate your careful observation regarding the protrusion morphology. We apologize for the confusion caused by the representative live-cell imaging example chosen in Fig. 3C, which may not have shown typical dendritic branching characteristics. The actin network was particularly visible in this example with a wider protrusion, led us to initially select it as the representative image. However, as you kindly suggested, we now realize that it may distract people from the authenticity of our results. Thus, we have replaced Fig. 3C with a new example demonstrating typical DLPs features and updated the related statistics to strengthen our conclusion in the revised manuscript (Figure R2-2 in below).

Figure R2-2. (A) Representative time-lapse images of SiR-actin labeled actin filaments in DLPs. (B) Total F-Actin filaments length (μ m) and DLP length (μ m) over time.

Salmonella infected THP-1 macrophages (MOI=20). Scale bar, 20 μ m. Magnified views show the DLPs. Red lines depict the actin filaments detected by Ridged detection. Scale bar, 20 μ m. (B) Quantification of the DLPs length and actin filaments length over time in infected THP-1 macrophages in (A). Growing, steady and collapsing stages are denoted by green, gray and blue, respectively.

Concerning the slow growth and collapse of DLPs, we speculate it may reflect the requirement of the transcriptional regulation. Our results indicate the expression and trans-localization of ARHGEF3 are critical to link bacterial stimulus to DLPs formation (Fig. 5). This might explain the slow formation of DLPs compared to filopodia or lamellipodia, which only require rapid pre-transcriptional signaling cascades for induction.

Concerning P- and P+, we acknowledge that other gene differences between P+ / P- cells might have affected the credibility of our results. To address this, we conducted an *in vivo* re-infusion experiment, including P+ cells treated with ARHGEF3 siRNA and non-silencing control, respectively, which support the conclusion in the initial submission that formation of DLPs significantly inhibits bacterial infection in mice (shown in Figure R2-3). This experiment further validated the importance of DLPs in bacterial defense and strengthened the reliability of our findings.

Figure R2-3. (A) Schematic diagram of extraintestinal infection mouse model incorporating macrophage re-infusion. (B-C) Quantification of bacterial load indicated by colony forming unit (CFU) in peritoneal lavage fluid (B) and liver (C) infected with *Salmonella*. (D-F) Quantitative RT-qPCR analysis of the expression levels of inflammatory markers in the spleen. Data are presented as mean \pm s.d.. ns $P > 0.05$; * $P < 0.05$; ** $P < 0.01$: unpaired two-tailed Student's *t*-test (B-F).

We have incorporated all these new findings together with controls and quantifications in the revised manuscript, along with improved methodological descriptions. These revisions substantially strengthened our study while directly addressing your insightful critiques. We are grateful for your constructive feedback, which has undoubtedly improved the quality and clarity of our manuscript.

Additional major concerns:

1. General comments:

- Figures for which a quantification of the observed phenotype over a population of cells should be provided: Figure 1I, Figure 3c-d, Figure 4a, Figure 4h-i.

Response:

We thank you for this kind suggestion. The quantifications have been added to Fig. 1L, Fig. 3D, Fig. 4A, Fig. 4j,K. We also summarized all the quantifications as Fig. R2-4 in below. We apologize for not including these data in the initial submission (due to space constraints in the main figures), which we indeed carried out these control experiments. We are grateful for your comment highlighting the importance of including quantitative data.

Figure R2-4. (A-C) Upper panels: representative time-lapse imaging reveals DLPs formation

in iBMDM (A), RAW264.7 (B), and mouse peritoneal macrophages (C) during *Salmonella* infection (MOI=20). Scale bars, 50 μ m. Middle and lower panels: pie chart quantification of the percentage of three shapes of macrophages in iBMDM (A), RAW264.7 (B), and mouse peritoneal macrophages (C) with or without *Salmonella* infection (MOI=20). (D) Upper panels: representative images reveal DLPs formation in THP-1 macrophages infected with *Salmonella*, *E. coli*, *Shigella*, *Bacillus*, and *Listeria* (MOI=20), respectively. (E-F) Lower panels: pie chart quantification of the percentage of three shapes of THP-1 macrophages with or without infection. Left panels: representative time-lapse imaging reveals DLPs formation in iBMDM (E) and RAW264.7 (F) during LPS treatment. Scale bars, 20 μ m. Middle and right panels: pie chart quantification of the percentage of three shapes of macrophages in iBMDM (E) and RAW264.7 (F) with or without LPS treatment. Blue, round shape (I); Red, fusiform shape (II); Green, deformed shape with DLPs (III). Data are presented as mean \pm s.d. calculated based on all cells from three independent experiments. *** $P < 0.001$.

- For many figures, the legend states "n= x cells": from how many independent experiments do they come from? Statistics lacking in all the pie charts with percentages. Ideally, the percentages from each experiment should be displayed and statistical tests should be done on them.

Response:

We appreciate your suggestion regarding experimental reproducibility and statistical analysis. All quantitative data were obtained from three independent biological experiments, with cells randomly sampled from multiple fields of view. We have updated the Methods section to detail our sampling approach and have revised the figures to include standard deviations for percentages, statistical comparisons between replicates, and explicit p values in the legends. We should have labeled the variance and significance of the pie charts. The charts and legends with additional statistical information have been updated in the revised manuscript and figures. These modifications provided full transparency regarding experimental replication while maintaining our original conclusions. We appreciate the opportunity to strengthen our data presentation.

- Figure for which controls (untreated cells) are missing: Figure 3f (inhibitors could have an effect on macrophage shape even in the absence of *Salmonella* infection), Figure 3h (VIM KO could have an effect on macrophage shape even in the absence of *Salmonella* infection), Figure 4k-l (TLR4 KO could have an effect on macrophage shape even in the absence of *Salmonella* infection), Figure 4o (inhibitor could have an effect on macrophage shape even in the absence of *Salmonella* infection).

Response:

We appreciate your suggestion regarding the requirement for proper controls. In response, we have performed comprehensive morphological analyses of macrophages under all specified experimental conditions without *Salmonella* infection. Our data demonstrated that neither actin inhibitors (Figure R2-5 in below), VIM knockout (Figure R2-6 in below), TLR4 knockout (Figure R2-7 in below), nor TLR4 inhibitor treatment (Figure R2-8 in below) caused significant alterations in basic macrophage morphology compared to untreated wild-type cells. These results confirm that the DLPs formation described in our study specifically requires bacterial infection and is not an artifact of our experimental manipulation. We have included these control experiments in the revised manuscript to provide complete validation of our findings.

Figure R2-5. Representative images of actin filaments visualized by phalloidin in Veh., 0.5 μ M Latrunculin B (LatB), 10 μ M Blebbistatin, 40 μ M CK666, 15 μ M SMIFH2 and 1 μ M NP-G2-044 treated THP-1 macrophages, respectively. The white dashed lines outline the cell contours. Scale bars, 10 μ m.

Figure R2-6. Representative images reveal the cell shape in WT and VIM KO THP-1 macrophages. Scale bars, 20 μ m.

Figure R2-7. Representative images reveal the cell shape in WT and TLR4 KO THP-1 macrophages. Scale bars, 20 μ m.

Figure R2-8. Representative images reveal the cell shape in THP-1 macrophages with or without TAK242 treatment. Scale bars, 20 μ m.

2. Specific questions/comments:

- Supplementary Figure S2a-c could be in the main figure, as it is the quantification corresponding to the observations made in the images shown in Figure 2c.

Response:

We agree with you for this suggestion and have moved Fig EV2A-C to the main figure, as new Fig. 2B,C,H in the revised manuscript to better present the quantitative data alongside the corresponding experimental observations.

- Figure 2c: Could the authors provide a quantification of the percentage of pseudopods extending in the upper vs lower channel in mock vs Salmonella conditions?

Response:

We have added relevant statistics in Fig. 2C, covering both the proportion of DLPs-forming cells and the length of DLPs (Figure R2-9 in below).

Figure R2-9. (A) Pie chart quantification of the percentage of three shapes of macrophages in THP-1 macrophages in microfluidics infected with or without *Salmonella* (MOI=20). Blue, round shape (I); Red, fusiform shape (II); Green, deformed shape with DLPs (III). (B) Percentage of THP-1 macrophages in microfluidics with or without DLPs. Data are presented as mean ± s.d. calculated based on all cells from three independent experiments. *** $P < 0.001$.

- line 278: "Collectively, these findings suggest that cyclic assembly and disassembly of bundled actin and vimentin are crucial for DLPs morphogenesis." The cyclic aspect was only observed here, whether or not it is crucial was not tested.

Response:

We apologize for the imprudent description. We have now revised the text to more accurately state "Collectively, these findings suggest that assembly and disassembly of bundled actin and vimentin are crucial for DLPs morphogenesis.". This change removes the unsupported claim about cyclicity

while maintaining our core conclusion.

- Figure 5a: why a first round of LPS stimulation here? Why not seed the cells directly in the 96-well plate to establish the different clones? The two rounds of LPS stimulation might affect gene expression. In addition, what specific criteria were used to identify the P- vs P+ colonies? Was it based on the percentage of cells that adopt the type III morphology after LPS stimulation? If so, what was the threshold?

Response:

We thank you for this kind suggestion to clarify our experimental design. We sincerely apologize for the ambiguity in our original schematic diagram, which may have led to confusion regarding the experimental design. In our experiments, two parallel cell populations (#1 and #2) derived from the same batch of iBMDM macrophages were used. Parallel #1 underwent the first round of LPS stimulation to confirm the reproducibility of DLPs formation in this batch of cells. Once validated, parallel #2 was subjected to monoclonal cell seeding in a 96-well plate, followed by LPS stimulation to screen for P+ and P- colonies based on morphological criteria. The specific criteria used to identify P- vs. P+ colonies were based on the percentage of cells that were adapted to the type III morphology after LPS stimulation. More than 40% of type III cells were considered P+, while less than 10% of type III cells were considered P-. The specific clones selected for RNA sequencing represented the most extreme phenotypes within these groups, with the sequenced P+ clone showing ~52% of type III cells and the P- clone showing only ~1%. The sequencing results were verified by comparison with another pair of P+ and P- monoclonal iBMDM lines, in which the level of ARHGEF3 was also significantly increased. We have revised Fig. 5A to better illustrate this experimental scheme and added detailed descriptions of the selection criteria to the Methods section of the revised manuscript. We appreciate the opportunity to clarify these important methodological details.

- Supplementary Figure S5c: could be in the main figure?

Response:

We agree with you and have moved Figure S5C to the main figure, as shown in Fig. 5E, in the revised manuscript.

- Figure 6o-s: The authors conclude from the re-infusion experiment that DLPs help control bacterial infection; however, P- and P+ macrophages differ not only in their ability to form DLPs but also in other regards, as demonstrated by the RNA-seq analysis in Figure 5. Therefore, the conclusion should be more nuanced to reflect the possibility that other mechanisms might be at play to explain why the re-infusion of P+ macrophages is more efficient than the one of P- macrophages at controlling the infection. Alternatively, to demonstrate that

DLPs are definitely involved, the authors could perform re-infusion experiments with macrophages silenced for ARHGEF3, if they can also show that the silencing does not affect some key parameters such as pro-inflammatory cytokine production.

Response:

We acknowledge that P- and P+ macrophages differ not only in their ability to form DLPs but also in many other genes. As you kindly suggested, to directly test whether DLPs formation specifically contributes to bacterial control, we knockdown ARHGEF3 in P+ macrophages by siRNA and performed the *in vivo* re-infusion experiment in mice. The results showed that ARHGEF3 knockdown in P+ cells reversed their enhanced infection control capacity indicated by the bacterial load in the peritoneal lavage fluid of the primary infection site and liver tissue of the secondary infection site (Figure R2-3), supporting a specific role for DLPs.

Moreover, transcriptional analysis confirmed that ARHGEF3 knockdown did not alter the systemic levels of pro-inflammatory cytokines (IL-1 β , TNF α , and IFN- γ) in the spleen (Figure R2-3), suggesting that the protective effect is not due to differential cytokine responses. We have revised the manuscript to present these findings with appropriate nuances, maintaining our core conclusions while acknowledging the complexity of macrophage responses.

3. Methods that are unclear or missing:

- The methods don't indicate how the bacteria were cultured.

Response:

We thank you for this kind reminding. We have supplemented the bacterial culture details “the bacteria were cultured overnight at 37°C in LB broth, washed in PBS, and transferred 1:33 to 2 mL fresh LB medium and grown for another 2 h to logarithmic phase.”, in the methods of the revised manuscript.

- The methods for LPS and LTA stimulation are missing from the Material and Methods section. Especially, the duration of the stimulation is not clear in the legend of Figure 4.

Response:

We thank you for this kind reminding. We have now added detailed methods for LPS and LTA treatments in the revised manuscript: “macrophages were treated with LPS (100 ng/mL) or LTA (1 μ g/mL) diluted in cell culture medium, then bright-field live-cell imaging was recorded at 37°C in 5% CO₂ for up to 10 h”. These experimental details have been included in both the methods section and relevant figure legends to ensure complete documentation.

- WGA staining is not described in the Material and Methods section.

Response:

We thank you for this kind reminding. We have supplemented the WGA staining details “Fluorescent wheat germ agglutinin (WGA) conjugates are enabled to label the plasma membrane. For staining, the prewarmed (37°C) WGA (10 µg/mL)-containing medium was applied cells for 2 h, then replaced with fresh medium and followed bacterial infection to observe the cells using a fluorescence microscope.” in the methods of the revised manuscript.

- It is not clear in the Material and Methods how GTP-RhoA levels were assessed by western blot, was it with an antibody specific to the GTP-bound form of the protein?

Response:

A commercial active Rho detection kit (#8820, CST) was used. The kit uses Rhotekin-RBD beads to pull down active GTP-RhoA, which was then detected by Western blot with an anti-RhoA antibody. We have added these details to the methods of the revised manuscript.

- line 676 (bacterial infection), paragraph starting with "For infection of THP-1, RAW264.7 and iBMDMs macrophages..." How the infection was performed is not clear. How long were the macrophages exposed to the bacteria before being cultured with gentamicin?

Response:

We thank you for this comment regarding the infection protocol. We have now clarified in the methods that macrophages were first infected with suspended *Salmonella* in serum-free, antibiotic-free medium for 30 min. The supernatant medium containing bacteria was then collected, centrifuged, and resuspended in medium with 10% FBS and 2 µg/mL gentamicin for the main infection (time 0). This gentamicin concentration (tested and shown in Figure R2-10, also referred to as PMID: 36717589) prevented bacterial overgrowth while maintaining infectivity. We have added these details to the methods section in the revised manuscript.

Figure R2-10. (A) Bacterial survival rates in THP-1 macrophages after infection with *Salmonella* under varying gentamicin concentrations (0, 2, and 20 µg/mL). (B) Representative image of the CFU assay of *Salmonella* from THP-1 macrophages cultured in medium containing

2 µg/mL gentamicin.

- line 747 (transfection): How was the transfection performed? Using what reagent?

Response:

We thank you for this kind reminding. THP-1 macrophages were transiently transfected with constitutively active RhoA (CA, L63) (#15900, Addgene), dominant negative RhoA (DN, N19) (#15901, Addgene) or EGFP (negative control) by using Lipofectamine™ LTX (#15338100, Invitrogen). We have added the details to the methods of the revised manuscript.

- line 821 (co-immunoprecipitation): How were iBMDMs stably transfected?

Response:

We thank you for raising this important question. We have supplemented the transfection details “iBMDMs were transfected with pMSCV2.2-TLR4-HA-EGFP (#60206, Addgene) by lentivirus and sorted for GFP-positive cells by flow cytometry, which assumed as stable transfection after enrichment and verification by microscopy.” in the methods of the revised manuscript.

- line 867 (CRISPR KO): How were THP-1 and iBMDMs transfected? Using what reagent?

Response:

We have supplemented the transfection details “For construction of VIM-KO lines, we transfected plasmids based on pSpCas9-2A-GFP vector (#48138, Addgene) by electroporation using Bio-RadGene Pulser electroporation buffer (#1652676, Bio-Rad); For construction of TLR4-KO lines, cells were transduced with lentiCRISPRv2 (#98290, Addgene) lentivirus.” in the methods of the revised manuscript.

- line 904 (luciferase assay): More details on the experimental procedure should be provided regarding the cloning of the promoter of ARHGEF3.

Response:

We appreciate your suggestion to clarify the details of promoter cloning. We have supplemented the cloning details “The promoter region (2000 bp upstream of TSS) of ARHGEF3 containing NF-κB binding sites was amplified and inserted into the pGL3 basic vector (Promega), to generate pGL3-prom-ARHGEF3. iBMDMs at 80% confluence were transfected this construct, treated after 48 hours, and luciferase activity was quantified using the Synergy H1 Hybrid Multi-Mode Reader (BioTek).” in the methods of the revised manuscript.

- TIM: How was the 3D reconstruction done?

Response:

The 3D reconstruction in Fig.1D was generated using the automated surface

reconstruction function in Imaris 10.0 software (Oxford Instruments). We have added this details to the methods of the revised manuscript.

- Figure legends: It is not always clear for how long were the infections done.

Response:

We appreciate your comment regarding the need for clearer infection timepoints. We have now systematically added information about the duration of infection in all the corresponding figure legends of the revised manuscript.

Other concerns:

1. Inaccuracies:

- Figure 1d: It is not mentioned in the legend why some macrophages are green in the 3D reconstruction.

Response:

We appreciate your attention to this detail. The green coloration in Fig. 1D highlights the macrophages exhibiting clear pseudopod formation. We have now explicitly stated this interpretation in the revised figure legend to avoid any potential confusion.

- Figure 5p: in the text, refers to a co-IP experiment, but actually microscopy images in the figure (5p and 5r seem to be inverted).

Response:

We appreciate your attention to this detail. We have corrected the panel order in Fig. 5P-R to properly match the co-IP experimental description in the text. The revised figure now accurately represents the experimental data.

- line 436: "indicating that TLR4 also guides the spatial localization of ARHGEF3, thereby promoting DLPs formation" Could also be the other way around.

Response:

We thank you for this important suggestion. Our new experimental data with the TLR4 inhibitor TAK242 (Figure R2-11 in below) demonstrated that TLR4 activation is indeed required for ARHGEF3 recruitment to DLPs, as ARHGEF3 staining intensity significantly decreased upon TLR4 inhibition. While we cannot completely rule out some reciprocal regulation, these results strongly support our conclusion that TLR4 signaling guides ARHGEF3 spatial localization during DLPs formation. We have modified the text to reflect this interpretation more precisely, while acknowledging the complexity of these interactions.

Figure R2-11. Representative images of THP-1 macrophages with DLPs stimulated by LPS with TAK242 pre-treatment stained with ARHGEF3 (green) and TLR4 (red) antibodies. The white dashed lines depict the outline of the cell. Cropped views depict the DLPs regions. Yellow arrows depict DLPs. Bars, 20 μm (in cell images) and 5 μm (in the magnified images).

- line 655 (differentiation of human macrophages): "These monocytes were plated in tissue culture plates with 25% medium conditioned by recombinant mouse M-CSF (carrier-free, 20 ng/mL)" Why mouse M-CSF to differentiate human cells?

Response:

We apologize for the oversight. This was a typographical error. Recombinant human M-CSF was used to differentiate human macrophages. The text has been accordingly corrected in the revised manuscript.

- In the legend of Figure 1, Figure 3 and Figure 4, the authors state: "Data are presented as mean {plus minus} s.d. from three independent experiments." This statement is not correct for most of the graphs presented in those figures, where dots represent individual cells.

Response:

We thank you for this kind reminding. We apologize for the confusion caused by insufficient description in the figure legends. Dots representing individual cells were obtained from three independent experiments. We have revised the legends as "Data are presented as mean \pm s.d. calculated based on all cells from three independent experiments." in Fig. 1, 3, and 4 of the revised manuscript, respectively.

- line 1321 (Figure 3 legend): Were the macrophages analyzed in Figure 3h infected with *Salmonella*? This information should be stated in the legend.

Response:

We thank you for pointing out this omission. We confirmed that the macrophages analyzed in Fig. 3H were infected with *Salmonella*. We have now explicitly stated this in the revised figure legends.

- line 1371: "Schematic diagram shows that NF- κ B binds to ARHGEF3 promoter." That is incorrect, the diagram shows that there is a predicted NF- κ B binding site in the ARHGEF3 promoter.

Response:

We thank you for careful reading and constructive comment. We agree that our original wording overstated the experimental evidence, and we have revised the text to more accurately state: "predicted NF- κ B binding site in the ARHGEF3 promoter."

- There are 7 supplementary videos in the supplementary figure legends but only 6 files. The file for supplementary movie 6 fits the legend of supplementary video 7.

Response:

We sincerely apologize for this oversight. We have now verified and uploaded all the 7 supplementary videos corresponding to the legends in the revised manuscript.

2. writing style:

- Introduction is sometimes unnecessarily wordy.

Response:

We appreciate this helpful suggestion. We have carefully edited the Introduction to improve clarity and conciseness, while maintaining all key scientific points. The revised version removes redundant phrasing and sharpens the focus on our study's rationale and objectives.

- line 72: "morphogenetic remodeling paradigms"? Does the term "morphogenetic" really apply in this context? Maybe "morphological" instead?

Response:

We thank you for this suggestion. We have changed "morphogenetic" to "morphological" for more accurate description of cell shape changes.

- line 165: "and exposed it to macrophages" should be corrected to "and exposed macrophages to them"

Response:

Thank you. We have corrected the sentence to read: "and exposed macrophages to them" in the revised manuscript.

- line 151: "There results indicate that an increase in..."

Response:

We have removed the indicated sentence to streamline the text and enhance readability.

- line 207: "while excluded THP-1" should be corrected to "while excluding THP-1"

Response:

We agree with you. We have corrected the text to "while excluding THP-1" in the revised manuscript.

- line 342: "After a 24-hour recovery interval, single-cell colonies were established by sorting individual macrophages into 96-well plates, followed by enrichment in the second round of LPS stimulation, colonies with significant variance in their capacity for DLPs production were identified as 19 P- colonies with minimal DLPs and 5 P+ colonies with robust DLPs formation." Clarity would be improved by breaking this sentence into two.

Response:

We appreciate your helpful suggestion to improve readability. We have revised this sentence by breaking it into two sentences: "After a 24-hour recovery interval, single-cell colonies were established by sorting individual macrophages into 96-well plates, followed by enrichment in the second round of LPS stimulation. Colonies with significant variance in their capacity for DLPs production were identified as 19 P- colonies with minimal DLPs and 5 P+ colonies with robust DLPs formation."

- line 381: "implicating" should be corrected to "suggesting that"

Response:

Thank you. We have revised the text as "suggesting that".

- line 400: "suggesting that NF- κ B is the key cause of LPS-TLR4 to boost ARHGEF3 expression" should be corrected to "suggesting that NF- κ B is the key factor through which LPS-TLR4 boosts ARHGEF3 expression"

Response:

We agree with you. We have revised the text as you suggested.

- line 415: "inhibition of RhoA signaling using Y-27632, an inhibitor of..."

Response:

Thank you. We have removed the text "an inhibitor of..." in the revised manuscript to make the explanation more concise and reader-friendly.

- line 789 (mice infection): Only male mice were analyzed for the re-infusion experiments, why?

Response:

We appreciate your question regarding our use of male mice. While *Salmonella* infection does not show gender preference in mice, we used male mice in these experiments to maintain consistency with our previous experimental conditions and to align with established protocols in the field (e.g., PMID: 35969050).

- line 794: What does "partially euthanized" mean?

Response:

We thank you for catching this imprecise wording. We have revised the text to clarify: "Mice for tissue and organ collection were anesthetized and euthanized at 48 hpi, while those for body weight measurements were euthanized at 72 hpi."

- line 894: "The following drugs were used at a specified doses and times"

Response:

Thank you. We have revised the text as "The following drugs were used" in the revised manuscript to make the text clearer and easier to read.

- line 943: "Primary and secondary antibodies were diluted in fresh blocking buffer for overnight at 4{degree sign}C and for 1 h at room temperature, respectively."

Response:

We thank you for this kind reminding. To clarity, we have modified the text as "Primary and secondary antibodies were diluted in fresh blocking buffer. Primary antibody incubated with PVDF membranes overnight at 4°C and secondary antibodies incubated with PVDF membrane for 1 h at room temperature, respectively." In the revised manuscript.

- line 949: "iBMDM monoclonals" should be replaced by "monoclonal iBMDMs"

Response:

We agree with you. We have corrected the text as "monoclonal iBMDMs" in the revised manuscript.

- Title of Fig. 1 should be: "Macrophages form DLPs in response to a bacterial challenge".

Response:

Thank you. We have corrected the title of Fig. 1 as "Macrophages form DLPs in response to a bacterial challenge" in the revised manuscript.

- line 1253: "from three separate sets of experimental mice" Do the authors mean from three experiments?

Response:

We thank you for catching this unclear wording. We have revised the text to state "from three experiments" for better clarity and precision in the revised manuscript.

- line 1299 (Figure 1 legend): "Schematic diagram illustrates that macrophages directional growth requires the contact with bacteria."

Response:

Thank you. We have revised the text as "Schematic diagram illustrates that the directional growth of DLPs require the contact with bacteria" in the revised manuscript.

- There are sentences written in the present tense in most figure legends, which are a bit unnatural to read (for example, "Representative images visualize the membrane..."). These sentences should be corrected to use the gerund (for example, "Representative images showing the membrane...") or to avoid the use of a verb altogether.

Response:

We appreciate your helpful suggestion regarding figure legend style. We have systematically revised all figure legends to use gerund forms (e.g., "showing" instead of "visualize") for more natural scientific writing.

- line 1296: "membrane pore" should be plural.

Response:

Thank you. We have modified the "membrane pore" to plural form as "membrane pores" in the revised manuscript.

- lines 1313 and 1329: "stage" should be plural.

Response:

Thank you. We have modified "stage" to plural form as "stages" in the revised manuscript.

- line 1335: "TLPs" should be DLPs.

Response:

Thank you. We have revised this typo as "DLPs" in the revised manuscript.

- line 1370: "Western blot analysis of ARHGEF3 and the active levels of RhoA treated with LPS for indicated time in THP-1 macrophages" should be corrected to: "Western blot analysis of ARHGEF3 and the active levels of RhoA in THP-1 macrophages treated with LPS for indicated times".

Response:

Thank you. We have revised the text as you suggested in the revised manuscript.

- line 1378: "The diagram of RhoA-ROCK signaling pathway" should be corrected to: "Diagram of the RhoA-ROCK signaling pathway".

Response:

We agree with you. We have revised the text as "Diagram of the RhoA-ROCK signaling pathway" in the revised manuscript.

Referee #3:

This manuscript presents evidence for a previously undescribed response of macrophages to infection with Gram-negative bacteria. While many studies have reported that macrophages rapidly take up bacteria via "capture" by filopodia and subsequent engulfment in actin-rich phagocytic cups, here the authors describe a later, more prolonged response in which cells extend long, branched, relatively stable protrusions that they call dendrite-like pseudopods (DLPs). Unlike filopodia or lamellipodia, which are entirely actin-based, DLPs contain a core of vimentin which appears to enhance their stability. Mechanistically, the authors describe a signaling cascade in which the bacterial lipopolysaccharide (LPS) activates the Toll-like receptor TLR4, inducing expression of the RhoA activating protein ARHGEF3 in an NFkB-dependent manner, all of which is necessary for DLP formation. Physiologically, they show that peritoneal injection of DLP-proficient macrophages in mice enhances clearance of peritoneal *Salmonella*, relative to non-DLP proficient cells. Taken together, these observations describe a novel mechanism for the clearance of Gram-negative bacteria by macrophages.

In general, the experiments are thorough and well controlled, the data are clear, statistically sound and support the conclusions drawn by the authors. I have only a few relatively minor concerns that should be addressed:

Response:

We sincerely thank you for the thorough evaluation and positive assessment of our work. We have carefully addressed all comments to improve methodological clarity, data presentation, and interpretation. These revisions have strengthened the manuscript while maintaining our core findings about this new bacterial ingestion pathway. We believe that the revised manuscript now makes a clearer contribution to our understanding of macrophage responses to infection.

1. It is unclear whether bacteria are present during the entire (typically 6h) incubation period or the authors use a standard gentamicin treatment assay in which cells are exposed to a pulse of bacteria then treated with gentamicin to kill remaining extracellular bacteria. The reason this is important is that the authors infect at an MOI of 20, but bacteria in the nutrient-rich medium would be expected to proliferate dramatically over a 6h incubation period.

Response:

We appreciate this important question. To clarify our protocol, macrophages were first infected with suspended *Salmonella* in serum-free, antibiotic-free medium for 30 min. The supernatant medium containing bacteria were then collected, centrifuged, and resuspended in a medium containing 10% FBS and 2 µg/mL gentamicin. This gentamicin concentration, which we validated in Figure R3-1 and in previous studies (PMID: 36717589), effectively prevented bacterial overgrowth while maintaining infectivity throughout the incubation

period. Cells were re-infected with these gentamicin-added bacteria. This time point was considered as 0 h of infection. We have added these methodological details to the revised manuscript to ensure full transparency about how we maintained consistent bacterial loads during our experiments.

Figure R3-1. (A) Bacterial survival rates in THP-1 macrophages after infection with *Salmonella* under varying gentamicin concentrations (0, 2, and 20 µg/mL). (B) Representative image of the CFU assay of *Salmonella* from THP-1 macrophages cultured in medium containing 2 µg/mL gentamicin.

2. It should be stated clearly in both the abstract and the text that this is quite a late response to LPS exposure, much later than the formation of filopodia or lamellipodia that are much more rapid (within minutes) and are driven by Cdc42 or Rac1 respectively, as opposed to RhoA.

Response:

We thank you for this important suggestion. In response, we have carefully revised the text to emphasize that DLP formation represents a delayed response occurring hours after infection, in contrast to the rapid (minutes) formation of Cdc42/Rac1-dependent filopodia and lamellipodia.

Key modifications include: (1) In the abstract, we now describe DLPs as emerging "in a few hours upon severe Gram-negative bacterial infections"; (2) The introduction now contrasts the immediate actin-based protrusions with our newly described delayed response; (3) The discussion explicitly compares the transient nature of classical protrusions (minutes) with the prolonged stability of DLPs (hours). These changes better position our findings within the established timeline of macrophage responses while clearly delineating the novel aspects of DLPs biology. We appreciate this opportunity to improve the clarity and context of our work.

3. It is unclear why the authors segregated cells into responder vs. non-responder (DLP-producing vs. non-producing) clonal cell lines prior to RNA-seq analysis as opposed to simply treating cells with LPS or not. Why was this necessary?

Response:

We thank you for this kind suggestion to make our experimental design clearer. The decision to establish clonal DLP-producing (P+) and non-producing (P-) cell lines was driven by the observation that only about 30% of the wild-type population of cells was able to form DLPs upon infection. By creating these distinct populations through careful clonal selection and LPS stimulation (with P+ clones showing >40% type III morphology versus P- clones with <10%), we could more effectively identify molecular signatures specific to DLP formation without the dilution effect seen in bulk populations. This approach allowed us to identify ARHGEF3 as a key regulator through RNA sequencing comparisons between matched P+ and P- clones with maximally divergent phenotypes (~52% vs. ~1% type III cells), which were then validated in independent clones.

The multiple rounds of LPS stimulation served to first confirm DLP-forming capacity in the parental population, then select and characterize clones, and finally trigger the response prior to molecular analysis. We have expanded the explanation of this experimental design in the revised manuscript to better communicate its rationale and advantages for identifying DLP-specific mechanisms that would be challenging to detect in heterogeneous wild-type populations.

4. A related point - the authors used these clonal cell lines for their *in vivo* infection model, injecting responders vs. non-responders into the peritoneum after *Salmonella* infection. While the data are consistent with the authors' interpretation that the ability to form DLPs enhances their ability to clear infection, there are literally hundreds of other genes that are differentially expressed that might also contribute to clearance. It would seem "cleaner" to use macrophages lacking ARHGEF3 (or not) to prove their point.

Response:

We thank you for this important suggestion. To directly test the specific role of ARHGEF3 in DLP formation, we performed additional *in vivo* re-infusion experiments using ARHGEF3-silenced P+ macrophages. As shown in Figure R3-2, ARHGEF3 knockdown in P+ cells reversed their enhanced bacterial control capacity, supporting that DLPs (rather than other differentially expressed genes) mediate this protective effect.

Figure R3-2. (A) Schematic diagram of extraintestinal infection mouse model incorporating macrophage re-infusion. (B-C) Quantification of bacterial load indicated by colony forming unit (CFU) in peritoneal lavage fluid (B) and liver (C) infected with *Salmonella*. (D-F) Quantitative RT-qPCR analysis of the expression levels of inflammatory markers in the spleen. Data are presented as mean ± s.d.. ns $P > 0.05$; * $P < 0.05$; ** $P < 0.01$: unpaired two-tailed Student's t -test (B-F).

that time, we observed and characterized the cell morphology into three distinct shapes: type I represents circular, elliptical, fleshy, and corpulent macrophages, although we coined them all as 'round' without strict subdivision; type II represents fusiformed, slender macrophages, although in some cases these cells have needle-like filopodia or protrusive flat lamellipodia; type III represents the dendrite-like pseudopods (DLPs) equipped macrophages, which we found in this study. Notably, we indeed observed more various irregular shape as differentiation went further (i.e., after 8 or 10 days). The DLPs-equipped macrophages can only be observed upon infection with a large number of bacteria. We have clarified this point in the revised manuscript.

For the concerns on adhesiveness, we performed vinculin staining (a key focal adhesion marker) in DLPs-equipped macrophages. Our results demonstrate that DLPs were enriched with focal adhesions (Figure R1-1 in below), indicating that their ability to anchor to the extracellular matrix and microenvironment adhesiveness may contribute to the formation of DLPs. We have added these data to Fig. EV1N,O and updated the corresponding text in the revised manuscript.

Figure R1-1. (A) Representative image of DLP-equipped THP-1 macrophages infected with *Salmonella* and stained with vinculin antibody. Scale bar, 10 μm . (B) Quantification of mean vinculin immunofluorescence intensity in cell bodies and DLPs, respectively. Data are presented as mean \pm s.d. from three independent experiments. *** $P < 0.001$: unpaired two-tailed Student's t -test.

For the concerns on original shape of macrophages that derived DLPs, by live-cell imaging tracking and quantification, we found that approximately 90% of DLPs emerged from type I round, unpolarized macrophages, while ~10% of DLPs emerged from type II fusiformed macrophages (Figure R1-2 in below). We have added these data to Fig. EV1A and updated the text in the revised manuscript.

Figure R1-2. Pie chart quantification of the percentage of the original shape of macrophages that derived DLPs upon *Salmonella* infection (MOI=20). Blue, from shape I to shape III; Red, from shape II to shape III. A total of 322 cells from three-independent experiments were used for quantification.

(ii) The percentage of macrophages with dendritic-like pseudopods in the entire population of cells is around 30% in most of the results. Are these cells a specific subpopulation of macrophages within the population? Could the authors test this by using macrophage surface markers in combination with immunofluorescence stainings?

Response:

We thank you for this insightful suggestion. To investigate whether

DLPs-forming macrophages represent a distinct subpopulation, we performed comprehensive molecular characterization. First, we isolated P+ (with excessive DLPs formation) and P- (with minimal DLPs formation) macrophages, and characterized their differential gene expression patterns by RNA sequencing. The results showed that macrophage surface markers (CD14 and F4/80) and M1 macrophage-specific surface markers (CD64 and CD32) were significantly increased upon *Salmonella* infection in P+, but not P-cells (Fig. 5D). In direct response to your suggestion, we further validated these findings through immunofluorescence staining, confirming that infected macrophages forming DLPs indeed constitute a distinct subpopulation with unique surface marker expression profiles (Figure R1-3 in below). These results have been incorporated into the revised manuscript, supporting our conclusion that DLPs formation is associated with a specific macrophage subpopulation rather than a random occurrence.

Figure R1-3. (A-C) Left panels: representative images showing the F-actin visualized by phalloidin and macrophage surface markers CD64 (A), CD32 (B), and CD14 (C) in P- and P+ iBMDMs, respectively. Scale bars, 10 μ m. Right panels: quantification of the mean intensity of macrophage surface markers CD64(A), CD32(B), and CD14(C) in P- and P+ iBMDMs, respectively. Data are presented as mean \pm s.d. calculated based on all cells from three independent experiments. * $P < 0.05$; ** $P < 0.01$: unpaired two-tailed Student's t -test.

(iii) Furthermore, the occurrence of cells exhibiting dendritic-like pseudopods is only around 30% of cells in the population at a MOI of 20 or higher. In contrast, an MOI of 5 only results in this shape in about 10% of the cells. Is an MOI of 20 physiologically relevant?

Response:

We thank you for this kind suggestion. In cell infection models, the multiplicity of infection (MOI) reflects the average number of bacteria per host cell. In *in vivo* experiments, infection load is typically measured using colony-forming units (CFU). While direct quantitative correlations between these parameters are inherently challenging due to microenvironmental differences (e.g., immune cell recruitment and tissue barriers), our experimental design aligns with established models of *Salmonella* pathogenesis.

For instance, *in vitro* macrophage infection studies (PMID: 38528181) routinely employed an MOI of 20 to mimic the high bacterial loads observed during

invasive infections, as shown by electron microscopy (Figure R1-4A). This parameter appears physiologically plausible: murine models of systemic salmonellosis (PMID: 25121751) demonstrate that infection with 5×10^7 CFU of *Salmonella* leads to enterocyte microcolonies containing ~20 bacteria per host cell within 12 hours (Figure R1-4B). Although direct comparisons between MOI and CFU require caution, the similarity in bacterial densities (~20 bacteria per cell *in vitro* vs. *in vivo*) suggests that our model captures infection intensities relevant to both localized and systemic contexts.

Furthermore, the dose-dependent increase in DLPs formation (10% at MOI=5 vs. 30% at MOI=20) supports the notion that this response reflects a regulated biological threshold rather than a nonspecific overload. We have expanded the discussion in the revised manuscript to better contextualize these findings, emphasizing the biological plausibility of our approach, while acknowledging the limitations of extrapolating *in vitro* parameters to physiological scenarios.

Figure R1-4. (A) Representative image of intracellular bacteria obtained by electron microscopy. *Salmonella* infection at an MOI of 20 for 10h. Scale bar, 5 μ m. (B) Representative confocal micrographs of cecum sections from infected C57BL/6 mice 18h post-infection, showing S.Tm-G⁺ (green), host cell nuclei (gray), and EpCAM (epithelial cell marker; yellow). Lu.=Lumen. Right row image shows blow-ups of boxed regions. Scale bars, 10 μ m.

(iv) The authors claim that the appearance of dendritic-like pseudopods is a specific reaction in macrophages and does not appear in other cell types. This argument is mostly based on experiments with THP-1-derived dendritic cells (Extended Data Fig. G-i). However, these data also show the appearance of protrusions in these cells, albeit shorter. While these protrusions have not exactly the same shape as in macrophages, it cannot be concluded from these data that dendritic-like pseudopods are not relevant to dendritic cells. In particular, THP-1-derived dendritic cells may also not be an ideal model for physiologically appearing dendritic cells. Thus, the authors should at least dampen their interpretation.

Response:

We thank you for raising this important point. We have dampened the interpretation in the revised manuscript as you kindly suggested. To further address this question, we performed additional experiments using human peripheral blood mononuclear cell-derived dendritic cells (MoDC), which represent a more physiologically relevant model than THP-1-derived DCs. We found that MoDCs indeed develop membrane protrusions upon infection. However, these structures were consistently shorter than macrophage DLPs, supporting the morphological distinction between these cellular responses (Figure R1-5 in below). These findings suggested that DLPs are a prominent feature of macrophages.

Figure R1-5. Representative images of phalloidin labeled actin filaments (magenta) and nuclei

(blue) in *Salmonella*-infected MoDC (MOI=20) for 6 h. Scale bar, 10 μ m.

(v) Figure 3 characterizes the cytoskeletal dependence and the dynamics of the dendritic-like pseudopods. However, the images on the dynamics shown in Figure 3 d and j - which are also the basis for quantifications shown in Figure 3d and k - show protrusion types with a wide non-branched shape that is entirely different from the described thin and branched dendritic-like pseudopods in the rest of the manuscript (see. e.g. Fig 3a). This is surprising, as it suggests that Figure 3 is based on the characterization of a different protrusion type other than dendritic-like pseudopods, questioning the results from these datasets.

Response:

We sincerely appreciate your careful observation of the protrusion morphology shown in Figure 3. We apologize for the confusion caused by the representative live-cell imaging example chosen in Fig. 3C, which may not have shown typical dendritic branching characteristics. The actin network was particularly visible in this example with a wider protrusion, which led us to initially select it as a representative image. However, as you kindly suggested, we now realize that it may distract people from the authenticity of our results. Thus, we have replaced Fig. 3C with a new example demonstrating typical DLPs features and updated the related statistics to strengthen our conclusion in the revised manuscript (Figure R1-6 in below). Figure 3J shows the phenotype of vimentin depletion with sequential 'budding and retracting' protrusions without successful DLPs formation.

Figure R1-6. (A) Representative time-lapse images of SiR-actin labeled actin filaments in *Salmonella* infected THP-1 macrophages (MOI=20). Scale bar, 20 μ m. Magnified views show the DLPs. The red lines depict actin filaments detected by ridged detection. Scale bar, 20 μ m. (B) Quantification of the DLPs length and actin filament length over time in infected THP-1 macrophages in (A). Growing, steady, and collapsing stages are denoted by green, gray and blue, respectively.

(vi) Statistics: many of the results are depicted in cake-like graphs (e.g. Fig. 1h, 2g, 3f,h etc...) that nicely illustrate the percentage of cells with dendrite-like pseudopods. However, they all lack information about statistical variance (error bars) and significance (statistical tests). It will be critical to add this information to all of these graphs of the manuscript to be able to judge the variability in cell shape types between different experiments (at least 3 replicates).

Response:

We thank you for this kind suggestion regarding the statistical presentation of our data. We agree with you that the statistical variance and significance are

critical for validating our data. In response, we have now incorporated measures of statistical variance (standard error) and significance testing into all relevant percentage-based graphs throughout the manuscript. Each figure now includes error bars representing the standard error, with statistical significance clearly indicated by asterisks. The specific p values and statistical tests used are detailed in the corresponding figure legends. An updated example (Figure 1H) is shown in Figure R1-7 below. We believe that these improvements significantly enhanced the statistical rigor and reproducibility of our findings.

Figure R1-7. Pie chart quantification of the percentage of the three macrophage shapes with or without *Salmonella* infection (MOI=20). The upper panels provide the definitions of the three protrusive structures. Blue, round shape (I); Red, fusiform shape (II); Green, deformed shape with DLPs (III). Data are presented as mean \pm s.d. calculated based on all cells from three independent experiments. *** $P < 0.001$.

Minor points:

(i) Intravital imaging of macrophage shapes: it remains unclear from the text and methods how the shapes were defined as having pseudopods or not (Fig. 1e). Further, were these shapes manually quantified or automatically in a non-biased manner? The latter would be particularly important, given the image resolution shown in Fig.1d, which makes it difficult to distinguish individual cells and thus may cause a bias in manual quantification.

Response:

We thank you for raising this important question. Most mouse peritoneal macrophages labeled with F4/80 and observed utilizing two-photon microscopy adhered to the peritoneal wall and aligned with the ECM network, exhibiting sleek and slender morphology. We defined 'with pseudopods' as when a macrophage derived three or more protrusions from two sides of its elongated axis, and each protrusion was over five microns in length. Reciprocally, we defined 'without pseudopods' as when a macrophage showed less than two protrusions. The 3D overview in Fig. 1D was obtained from automated surface reconstruction in the Imaris 10.0 software (Oxford Instruments).

For quantification method, images from the intravital microscopy were difficult to automatically analysis by the software which we have tried. To ensure objectivity, we implemented a double-blind manual quantification protocol: one researcher anonymized and randomized the Mock and infected datasets, and

a second researcher, blinded to the experimental conditions, performed all morphological classifications and calculations. This approach was necessary due to the limitations of the current software in reliably analyzing such complex intravital images, particularly given the resolution constraints evident in Fig. 1D. We have added detailed descriptions of both the morphological criteria and the blinded quantification protocol to the Methods section of our revised manuscript to enhance the methodological transparency.

(ii) The data on the shapes of macrophages upon their exposure to cellular debris from Caco2 colonic epithelial cells lack a quantification (Extended Fig. 1F).

Response:

We thank you for this kind suggestion. We have supplemented this quantification in Fig. EV1G in the revised manuscript supports the conclusion that cell debris cannot induce DLPs formation.

(iii) Single images of the appearance of cells with dendritic-like pseudopods in macrophages of different sources are shown in Figure 1I. This is an important dataset but lacks a quantification. Ideally, the quantification should be performed over time, as it is surprising that very specific time points (e.g., 62 min for BMDMs and 174 min for RAW cells) are shown.

Response:

We thank you for this kind suggestion. The very specific time points indicate the duration of each representative live-cell imaging video. We have the representative time points, and the corresponding quantifications are shown in Figure R1-8 in below. We have added these data to Fig. 1L and updated the text in the revised manuscript.

Figure R1-8. Upper panels: representative time-lapse imaging reveals DLPs formation in iBMDM (A), RAW264.7 (B), and mouse peritoneal macrophages (C) during *Salmonella*

infection (MOI=20). Scale bars, 50 μ m. Middle and lower panels: pie chart quantification of the percentage of three shapes of macrophages in iBMDM (A), RAW264.7 (B), and mouse peritoneal macrophages (C) with or without *Salmonella* infection (MOI=20). Blue, round shape (I); Red, fusiform shape (II); Green, deformed shape with DLPs (III). Data are presented as mean \pm s.d. calculated based on all cells from three independent experiments. *** $P < 0.001$.

(iv) The results from the microfluidic setup (Fig. 2c, d) need a quantification beyond the length of dendrite-like pseudopods: the number of dendrite-like pseudopods in mock vs. *Salmonella* conditions.

Response:

We thank you for this kind suggestion. We have calculated and supplemented the corresponding results in Fig. 2F,G of the revised manuscript (Figure R1-9 in below).

Figure R1-9. (A) Pie chart quantification of the percentage of three macrophage shapes located in the junction between the upper vertical microchannel (*Salmonella* infection for 6 h) / the lower vertical microchannels (Mock) and the middle horizontal chamber. Blue, round shape (I); Red, fusiform shape (II); Green, deformed shape with DLPs (III). (B) The percentage of extending ratio of DLP-positive macrophages to microchannels. Data are presented as mean \pm s.d. calculated based on all cells from three independent experiments. *** $P < 0.001$.

(v) Figure 3: definition of growing and collapsing stages: to this reviewer, it is not clear how the growing and collapsing stages were defined, as neither the growth nor the collapse is really evident in the figures and movies. Moreover, it is surprising that the protrusions grow and collapse so slowly over multiple hours, as macrophages normally can extend and retract protrusions within a few minutes.

Response:

We apologize for the confusion regarding our description of the growing and collapsing stages. We have optimized the description in the main text of the revised manuscript as follows: “(1) *the growing stage*, characterized by rapid actin assembly into bundles to fill a just-grown nascent DLP, where DLP length remains unchanged while F-actin intensity gradually increases; (2) *the steady stage*, the longest period (~67% of the cycle) during which both the protrusive structure and the internal actin bundles remained stable; and (3) *the collapsing stage*, during which the DLPs extended further, accompanied by rapid disassembly of actin bundles indicated by decreased F-actin intensity.”.

Regarding the unusually slow kinetics (hours versus minutes), we speculate it is may reflect the transcriptional regulation required for DLP formation. Our results indicate the expression and trans-localization of ARHGEF3 is critical to link bacterial stimulus to DLPs formation (Fig. 5). This distinguishes DLPs from rapidly-forming protrusions like filopodia that rely solely on pre-existing signaling cascades.

(vi) Figure 3F: This dataset lacks a control (solvent of inhibitors; e.g. DMSO).

Response:

We thank you for this kind suggestion. We have supplemented the control with the solvent of inhibitors in Fig. 3F of the revised manuscript. These results confirm that the solvent alone had no effect on the formation of DLPs. We apologize for not including this in the initial submission, which we indeed carried out these control experiments.

(vii) Figure 3J and I lack a control (WT control to VIM KO). Along this line: do VIM KOs already have an altered shape before the infection with *Salmonella*?

Response:

We apologize for the confusion in the interpretation of our results in the initial submission. The wild-type controls for Fig. 3J (VIM KO cells) are presented in Fig. 3C (actin dynamics) and Fig. 1G (morphology), as we aimed to maintain a logical flow by comparing both cytoskeletal networks. Importantly, we observed no significant morphological differences between wild-type and VIM KO THP-1 macrophages prior to infection, as shown in Figure R1-10 in below. This confirms that the altered protrusion dynamics observed in VIM KO cells following *Salmonella* infection (Fig. 3I,J) are specifically related to the infection response rather than baseline morphological changes. We included these control comparisons in the revised manuscript to provide complete context for our findings.

Figure R1-10. Representative images reveal the cell shape of WT and VIM KO THP-1 macrophages. Scale bars, 20 μm .

(viii) Figure 4A: this critical dataset lacks a quantification, which be essential to see the effects of gram-negative and -positive bacteria.

Response:

We thank you for this kind suggestion. We have now included the quantification in Fig. 4A (also Figure R1-11 in below), showing the comparative effects of Gram-negative and Gram-positive bacteria on DLPs formation. This analysis has been added in the revised manuscript.

Figure R1-11. Upper panels: representative images reveal DLPs formation in THP-1 macrophages infected with *Salmonella*, *E. coli*, *Shigella*, *Bacillus*, and *Listeria* (MOI=20), respectively. Lower panels: pie chart quantification of the percentage of three macrophage shapes in THP-1 macrophages with or without infection. Blue, round shape (I); Red, fusiform shape (II); Green, deformed shape with DLPs (III). Data are presented as mean \pm s.d. calculated based on all cells from three independent experiments.

(ix) Figure 4h, i: also these datasets lack a quantification.

Response:

We thank you for this kind suggestion. We have now included quantitative

analyses for Fig. 4J,K (also Figure R1-12 in below) which provide statistical validation of these results. These data have been added in the revised manuscript.

Figure R1-12. Left panels: representative time-lapse imaging reveals DLPs formation in iBMDM (A) and RAW264.7 (B) during LPS treatment. Scale bars, 20 μ m. Middle and right panels: pie chart quantification of the percentage of three shapes of macrophages in iBMDM (A) and RAW264.7 (B) with or without LPS treatment. Blue, round shape (I); Red, fusiform shape (II); Green, deformed shape with DLPs (III). Data are presented as mean \pm s.d. calculated based on all cells from three independent experiments. *** $P < 0.001$.

Referee #2:

The manuscript by Changyuan Fan et al., describes membrane protrusions induced by Gram-negative bacteria and LPS in macrophages. The authors use a lot of different approaches to investigate the signaling axis implicated in this process and identify a RhoGEF, ARHGEF3, acting upstream of NF- κ B.

The authors present a study that looks compelling, with a lot of experiments that rely on multiple approaches (both in vitro and in vivo, using different types of macrophages).

However, there is a major question that is never addressed concerning the invasive capacity of the bacteria used, which are *Salmonella Typhimurium*. These bacteria have been extensively studied as the archetypal invasive bacteria: they are known to inject in the host cell, via a secretion system, effectors that were described to remodel the actin cytoskeleton and induce membrane protrusions and ruffling. This large body of literature is totally ignored (work from the laboratory of Jorge Galan for instance).

In addition, the structures described seem to have several shapes depending on the experiments, from filopodia-like structures, to large membrane extensions and cone-shaped structures (figure 3).

The timing of the phenomenon observed is also unclear, as bacteria often induce quick membrane remodeling, while the process described here requires de novo transcription.

There is no proof that the described structures are involved in bacterial uptake. The two populations, P- and P+, used in vivo (figure 6), have distinct properties that could explain differences in bacterial survival.

In addition, there are major concerns regarding how data from multiple experiments are presented, as well as lack of controls and quantifications in several instances. Methods sometimes unclear or missing. English needs some work.

Response:

We sincerely appreciate your thorough evaluation of our manuscript and your valuable suggestions aiming to improve the quality of our study. We have taken these comments very seriously, and carefully addressed all of your concerns through additional experiments and extensive revisions to the text.

For bacterial secretion system, we acknowledge that the effectors of *Salmonella's* type III secretion system remodel the actin cytoskeleton and induce membrane protrusions and ruffling during infection, as highlighted in the work of Dr. Jorge Galan and many other previous studies. We generated *Salmonella* strains lacking SPI-1 (SipB mutant) or SPI-2 (PhoP mutant) to investigate whether bacterial effectors were required for DLPs formation. These mutants, previously confirmed to have defective secretion functions in our work (PMID: 36717589), were employed to block effector injection. By infecting macrophages with these mutated bacteria, we observed that DLPs

were still formed, while the actin network was perturbed within the cytoplasm (Figure 2-1 in below). These results indicate that while bacterial effectors influence cytoplasmic actin rearrangement, they are not required for DLPs formation, indicating that DLPs formation represents a distinct cellular response mechanism.

Figure R2-1. Upper panels: representative time-lapse imaging reveals DLPs formation in THP-1 macrophages during *Salmonella* (Left: wild type, Middle: SipB mutant, Right: PhoP mutant) infection (MOI=20). Scale bars, 50 μ m. Lower panels: pie chart quantification of the percentage of three macrophage shapes in THP-1 macrophages infected with different *Salmonella* strains (MOI=20).

Concerning the wider unbranched shape in Fig. 3, we sincerely appreciate your careful observation regarding the protrusion morphology. We apologize for the confusion caused by the representative live-cell imaging example chosen in Fig. 3C, which may not have shown typical dendritic branching characteristics. The actin network was particularly visible in this example with a wider protrusion, led us to initially select it as the representative image. However, as you kindly suggested, we now realize that it may distract people from the authenticity of our results. Thus, we have replaced Fig. 3C with a new example demonstrating typical DLPs features and updated the related statistics to strengthen our conclusion in the revised manuscript (Figure R2-2 in below).

Figure R2-2. (A) Representative time-lapse images of SiR-actin labeled actin filaments in *Salmonella* infected THP-1 macrophages (MOI=20). Scale bar, 20 μ m. Magnified views show the DLPs. Red lines depict the actin filaments detected by Ridged detection. Scale bar, 20 μ m. (B) Quantification of the DLPs length and actin filaments length over time in infected THP-1 macrophages in (A). Growing, steady and collapsing stages are denoted by green, gray and blue, respectively.

Concerning the slow growth and collapse of DLPs, we speculate it is may reflect the requirement of the transcriptional regulation. Our results indicate the expression and trans-localization of ARHGEF3 are critical to link bacterial stimulus to DLPs formation (Fig. 5). This might explain the slow formation of DLPs compared to filopodia or lamellipodia, which only require rapid pre-transcriptional signaling cascades for induction.

Concerning P- and P+, we acknowledge that other gene differences between P+ / P- cells might have affected the credibility of our results. To address this, we conducted an *in vivo* re-infusion experiment, including P+ cells treated with ARHGEF3 siRNA and non-silencing control, respectively, which support the conclusion in the initial submission that formation of DLPs significantly inhibits bacterial infection in mice (shown in Figure R2-3). This experiment further validated the importance of DLPs in bacterial defense and strengthened the

reliability of our findings.

Figure R2-3. (A) Schematic diagram of extraintestinal infection mouse model incorporating macrophage re-infusion. (B-C) Quantification of bacterial load indicated by colony forming unit (CFU) in peritoneal lavage fluid (B) and liver (C) infected with *Salmonella*. (D-F) Quantitative RT-qPCR analysis of the expression levels of inflammatory markers in the spleen. Data are presented as mean \pm s.d.. ns $P > 0.05$; * $P < 0.05$; ** $P < 0.01$: unpaired two-tailed Student's *t*-test (B-F).

We have incorporated all these new findings together with controls and quantifications in the revised manuscript, along with improved methodological descriptions. These revisions substantially strengthened our study while directly addressing your insightful critiques. We are grateful for your constructive feedback, which has undoubtedly improved the quality and clarity of our manuscript.

Additional major concerns:

1. General comments:

- Figures for which a quantification of the observed phenotype over a population of cells should be provided: Figure 1I, Figure 3c-d, Figure 4a, Figure 4h-i.

Response:

We thank you for this kind suggestion. The quantifications have been added to Fig. 1L, Fig. 3D, Fig. 4A, Fig. 4j,K. We also summarized all the quantifications as Fig. R2-4 in below. We apologize for not including these data in the initial submission (due to space constraints in the main figures), which we indeed carried out these control experiments. We are grateful for your comment highlighting the importance of including quantitative data.

Figure R2-4. (A-C) Upper panels: representative time-lapse imaging reveals DLPs formation

in iBMDM (A), RAW264.7 (B), and mouse peritoneal macrophages (C) during *Salmonella* infection (MOI=20). Scale bars, 50 μ m. Middle and lower panels: pie chart quantification of the percentage of three shapes of macrophages in iBMDM (A), RAW264.7 (B), and mouse peritoneal macrophages (C) with or without *Salmonella* infection (MOI=20). (D) Upper panels: representative images reveal DLPs formation in THP-1 macrophages infected with *Salmonella*, *E. coli*, *Shigella*, *Bacillus*, and *Listeria* (MOI=20), respectively. (E-F) Lower panels: pie chart quantification of the percentage of three shapes of THP-1 macrophages with or without infection. Left panels: representative time-lapse imaging reveals DLPs formation in iBMDM (E) and RAW264.7 (F) during LPS treatment. Scale bars, 20 μ m. Middle and right panels: pie chart quantification of the percentage of three shapes of macrophages in iBMDM (E) and RAW264.7 (F) with or without LPS treatment. Blue, round shape (I); Red, fusiform shape (II); Green, deformed shape with DLPs (III). Data are presented as mean \pm s.d. calculated based on all cells from three independent experiments. *** $P < 0.001$.

- For many figures, the legend states "n= x cells": from how many independent experiments do they come from? Statistics lacking in all the pie charts with percentages. Ideally, the percentages from each experiment should be displayed and statistical tests should be done on them.

Response:

We appreciate your suggestion regarding experimental reproducibility and statistical analysis. All quantitative data were obtained from three independent biological experiments, with cells randomly sampled from multiple fields of view. We have updated the Methods section to detail our sampling approach and have revised the figures to include standard deviations for percentages, statistical comparisons between replicates, and explicit p values in the legends. We should have labeled the variance and significance of the pie charts. The charts and legends with additional statistical information have been updated in the revised manuscript and figures. These modifications provided full transparency regarding experimental replication while maintaining our original conclusions. We appreciate the opportunity to strengthen our data presentation.

- Figure for which controls (untreated cells) are missing: Figure 3f (inhibitors could have an effect on macrophage shape even in the absence of *Salmonella* infection), Figure 3h (VIM KO could have an effect on macrophage shape even in the absence of *Salmonella* infection), Figure 4k-l (TLR4 KO could have an effect on macrophage shape even in the absence of *Salmonella* infection), Figure 4o (inhibitor could have an effect on macrophage shape even in the absence of *Salmonella* infection).

Response:

We appreciate your suggestion regarding the requirement for proper controls. In response, we have performed comprehensive morphological analyses of macrophages under all specified experimental conditions without *Salmonella* infection. Our data demonstrated that neither actin inhibitors (Figure R2-5 in below), VIM knockout (Figure R2-6 in below), TLR4 knockout (Figure R2-7 in below), nor TLR4 inhibitor treatment (Figure R2-8 in below) caused significant alterations in basic macrophage morphology compared to untreated wild-type cells. These results confirm that the DLPs formation described in our study specifically requires bacterial infection and is not an artifact of our experimental manipulation. We have included these control experiments in the revised manuscript to provide complete validation of our findings.

Figure R2-5. Representative images of actin filaments visualized by phalloidin in Veh., 0.5 μ M Latrunculin B (LatB), 10 μ M Blebbistatin, 40 μ M CK666, 15 μ M SMIFH2 and 1 μ M NP-G2-044 treated THP-1 macrophages, respectively. The white dashed lines outline the cell contours. Scale bars, 10 μ m.

Figure R2-6. Representative images reveal the cell shape in WT and VIM KO THP-1 macrophages. Scale bars, 20 μ m.

Figure R2-7. Representative images reveal the cell shape in WT and TLR4 KO THP-1 macrophages. Scale bars, 20 μ m.

Figure R2-8. Representative images reveal the cell shape in THP-1 macrophages with or without TAK242 treatment. Scale bars, 20 μ m.

2. Specific questions/comments:

- Supplementary Figure S2a-c could be in the main figure, as it is the quantification corresponding to the observations made in the images shown in Figure 2c.

Response:

We agree with you for this suggestion and have moved Fig EV2A-C to the main figure, as new Fig. 2B,C,H in the revised manuscript to better present the quantitative data alongside the corresponding experimental observations.

- Figure 2c: Could the authors provide a quantification of the percentage of pseudopods extending in the upper vs lower channel in mock vs *Salmonella* conditions?

Response:

We have added relevant statistics in Fig. 2C, covering both the proportion of DLPs-forming cells and the length of DLPs (Figure R2-9 in below).

Figure R2-9. (A) Pie chart quantification of the percentage of three shapes of macrophages in THP-1 macrophages in microfluidics infected with or without *Salmonella* (MOI=20). Blue, round shape (I); Red, fusiform shape (II); Green, deformed shape with DLPs (III). (B) Percentage of THP-1 macrophages in microfluidics with or without DLPs. Data are presented

as mean \pm s.d. calculated based on all cells from three independent experiments. *** $P < 0.001$.

- line 278: "Collectively, these findings suggest that cyclic assembly and disassembly of bundled actin and vimentin are crucial for DLPs morphogenesis." The cyclic aspect was only observed here, whether or not it is crucial was not tested.

Response:

We apologize for the imprudent description. We have now revised the text to more accurately state "Collectively, these findings suggest that assembly and disassembly of bundled actin and vimentin are crucial for DLPs morphogenesis.". This change removes the unsupported claim about cyclicity while maintaining our core conclusion.

- Figure 5a: why a first round of LPS stimulation here? Why not seed the cells directly in the 96-well plate to establish the different clones? The two rounds of LPS stimulation might affect gene expression. In addition, what specific criteria were used to identify the P- vs P+ colonies? Was it based on the percentage of cells that adopt the type III morphology after LPS stimulation? If so, what was the threshold?

Response:

We thank you for this kind suggestion to clarify our experimental design. We sincerely apologize for the ambiguity in our original schematic diagram, which may have led to confusion regarding the experimental design. In our experiments, two parallel cell populations (#1 and #2) derived from the same batch of iBMDM macrophages were used. Parallel #1 underwent the first round of LPS stimulation to confirm the reproducibility of DLPs formation in this batch of cells. Once validated, parallel #2 was subjected to monoclonal cell seeding in a 96-well plate, followed by LPS stimulation to screen for P+ and P- colonies based on morphological criteria. The specific criteria used to identify P- vs. P+ colonies were based on the percentage of cells that were adapted to the type III morphology after LPS stimulation. More than 40% of type III cells were considered P+, while less than 10% of type III cells were considered P-. The specific clones selected for RNA sequencing represented the most extreme phenotypes within these groups, with the sequenced P+ clone showing ~52% of type III cells and the P- clone showing only ~1%. The sequencing results were verified by comparison with another pair of P+ and P- monoclonal iBMDM lines, in which the level of ARHGEF3 was also significantly increased. We have revised Fig. 5A to better illustrate this experimental scheme and added detailed descriptions of the selection criteria to the Methods section of the revised manuscript. We appreciate the opportunity to clarify these important methodological details.

- Supplementary Figure S5c: could be in the main figure?

Response:

We agree with you and have moved Figure S5C to the main figure, as shown in Fig. 5E, in the revised manuscript.

- Figure 6o-s: The authors conclude from the re-infusion experiment that DLPs help control bacterial infection; however, P- and P+ macrophages differ not only in their ability to form DLPs but also in other regards, as demonstrated by the RNA-seq analysis in Figure 5. Therefore, the conclusion should be more nuanced to reflect the possibility that other mechanisms might be at play to explain why the re-infusion of P+ macrophages is more efficient than the one of P- macrophages at controlling the infection. Alternatively, to demonstrate that DLPs are definitely involved, the authors could perform re-infusion experiments with macrophages silenced for ARHGEF3, if they can also show that the silencing does not affect some key parameters such as pro-inflammatory cytokine production.

Response:

We acknowledge that P- and P+ macrophages differ not only in their ability to form DLPs but also in many other genes. As you kindly suggested, to directly test whether DLPs formation specifically contributes to bacterial control, we knockdown ARHGEF3 in P+ macrophages by siRNA and performed the *in vivo* re-infusion experiment in mice. The results showed that ARHGEF3 knockdown in P+ cells reversed their enhanced infection control capacity indicated by the bacterial load in the peritoneal lavage fluid of the primary infection site and liver tissue of the secondary infection site (Figure R2-3), supporting a specific role for DLPs.

Moreover, transcriptional analysis confirmed that ARHGEF3 knockdown did not alter the systemic levels of pro-inflammatory cytokines (IL-1 β , TNF α , and IFN- γ) in the spleen (Figure R2-3), suggesting that the protective effect is not due to differential cytokine responses. We have revised the manuscript to present these findings with appropriate nuances, maintaining our core conclusions while acknowledging the complexity of macrophage responses.

3. Methods that are unclear or missing:

- The methods don't indicate how the bacteria were cultured.

Response:

We thank you for this kind reminding. We have supplemented the bacterial culture details “the bacteria were cultured overnight at 37°C in LB broth, washed in PBS, and transferred 1:33 to 2 mL fresh LB medium and grown for another 2 h to logarithmic phase.”, in the methods of the revised manuscript.

- The methods for LPS and LTA stimulation are missing from the Material and Methods section. Especially, the duration of the stimulation is not clear in the

legend of Figure 4.

Response:

We thank you for this kind reminding. We have now added detailed methods for LPS and LTA treatments in the revised manuscript: "macrophages were treated with LPS (100 ng/mL) or LTA (1 µg/mL) diluted in cell culture medium, then bright-field live-cell imaging was recorded at 37°C in 5% CO₂ for up to 10 h". These experimental details have been included in both the methods section and relevant figure legends to ensure complete documentation.

- WGA staining is not described in the Material and Methods section.

Response:

We thank you for this kind reminding. We have supplemented the WGA staining details "Fluorescent wheat germ agglutinin (WGA) conjugates are enabled to label the plasma membrane. For staining, the prewarmed (37°C) WGA (10 µg/mL)-containing medium was applied cells for 2 h, then replaced with fresh medium and followed bacterial infection to observe the cells using a fluorescence microscope." in the methods of the revised manuscript.

- It is not clear in the Material and Methods how GTP-RhoA levels were assessed by western blot, was it with an antibody specific to the GTP-bound form of the protein?

Response:

A commercial active Rho detection kit (#8820, CST) was used. The kit uses Rhotekin-RBD beads to pull down active GTP-RhoA, which was then detected by Western blot with an anti-RhoA antibody. We have added these details to the methods of the revised manuscript.

- line 676 (bacterial infection), paragraph starting with "For infection of THP-1, RAW264.7 and iBMDMs macrophages..." How the infection was performed is not clear. How long were the macrophages exposed to the bacteria before being cultured with gentamicin?

Response:

We thank you for this comment regarding the infection protocol. We have now clarified in the methods that macrophages were first infected with suspended *Salmonella* in serum-free, antibiotic-free medium for 30 min. The supernatant medium containing bacteria was then collected, centrifuged, and resuspended in medium with 10% FBS and 2 µg/mL gentamicin for the main infection (time 0). This gentamicin concentration (tested and shown in Figure R2-10, also referred to as PMID: 36717589) prevented bacterial overgrowth while maintaining infectivity. We have added these details to the methods section in the revised manuscript.

Figure R2-10. (A) Bacterial survival rates in THP-1 macrophages after infection with *Salmonella* under varying gentamicin concentrations (0, 2, and 20 µg/mL). (B) Representative

image of the CFU assay of *Salmonella* from THP-1 macrophages cultured in medium containing 2 µg/mL gentamicin.

- line 747 (transfection): How was the transfection performed? Using what reagent?

Response:

We thank you for this kind reminding. THP-1 macrophages were transiently transfected with constitutively active RhoA (CA, L63) (#15900, Addgene), dominant negative RhoA (DN, N19) (#15901, Addgene) or EGFP (negative control) by using Lipofectamine™ LTX (#15338100, Invitrogen). We have added the details to the methods of the revised manuscript.

- line 821 (co-immunoprecipitation): How were iBMDMs stably transfected?

Response:

We thank you for raising this important question. We have supplemented the transfection details “iBMDMs were transfected with pMSCV2.2-TLR4-HA-EGFP (#60206, Addgene) by lentivirus and sorted for GFP-positive cells by flow cytometry, which assumed as stable transfection after enrichment and verification by microscopy.” in the methods of the revised manuscript.

- line 867 (CRISPR KO): How were THP-1 and iBMDMs transfected? Using what reagent?

Response:

We have supplemented the transfection details “For construction of VIM-KO lines, we transfected plasmids based on pSpCas9-2A-GFP vector (#48138, Addgene) by electroporation using Bio-RadGene Pulser electroporation buffer (#1652676, Bio-Rad); For construction of TLR4-KO lines, cells were transduced with lentiCRISPRv2 (#98290, Addgene) lentivirus.” in the methods of the revised manuscript.

- line 904 (luciferase assay): More details on the experimental procedure should be provided regarding the cloning of the promoter of ARHGEF3.

Response:

We appreciate your suggestion to clarify the details of promoter cloning. We have supplemented the cloning details “The promoter region (2000 bp upstream of TSS) of ARHGEF3 containing NF-κB binding sites was amplified and inserted into the pGL3 basic vector (Promega), to generate pGL3-prom-ARHGEF3. iBMDMs at 80% confluence were transfected this construct, treated after 48 hours, and luciferase activity was quantified using the Synergy H1 Hybrid Multi-Mode Reader (BioTek).” in the methods of the revised manuscript.

- TIM: How was the 3D reconstruction done?

Response:

The 3D reconstruction in Fig.1D was generated using the automated surface reconstruction function in Imaris 10.0 software (Oxford Instruments). We have added this details to the methods of the revised manuscript.

- Figure legends: It is not always clear for how long were the infections done.

Response:

We appreciate your comment regarding the need for clearer infection timepoints. We have now systematically added information about the duration of infection in all the corresponding figure legends of the revised manuscript.

Other concerns:

1. Inaccuracies:

- Figure 1d: It is not mentioned in the legend why some macrophages are green in the 3D reconstruction.

Response:

We appreciate your attention to this detail. The green coloration in Fig. 1D highlights the macrophages exhibiting clear pseudopod formation. We have now explicitly stated this interpretation in the revised figure legend to avoid any potential confusion.

- Figure 5p: in the text, refers to a co-IP experiment, but actually microscopy images in the figure (5p and 5r seem to be inverted).

Response:

We appreciate your attention to this detail. We have corrected the panel order in Fig. 5P-R to properly match the co-IP experimental description in the text. The revised figure now accurately represents the experimental data.

- line 436: "indicating that TLR4 also guides the spatial localization of ARHGEF3, thereby promoting DLPs formation" Could also be the other way around.

Response:

We thank you for this important suggestion. Our new experimental data with the TLR4 inhibitor TAK242 (Figure R2-11 in below) demonstrated that TLR4 activation is indeed required for ARHGEF3 recruitment to DLPs, as ARHGEF3 staining intensity significantly decreased upon TLR4 inhibition. While we cannot completely rule out some reciprocal regulation, these results strongly support our conclusion that TLR4 signaling guides ARHGEF3 spatial localization during DLPs formation. We have modified the text to reflect this interpretation more precisely, while acknowledging the complexity of these interactions.

Figure R2-11. Representative images of THP-1 macrophages with DLPs stimulated by LPS with TAK242 pre-treatment stained with ARHGEF3 (green) and TLR4 (red) antibodies. The white dashed lines depict the outline of the cell. Cropped views depict the DLPs regions. Yellow arrows depict DLPs. Bars, 20 μm (in cell images) and 5 μm (in the magnified images).

- line 655 (differentiation of human macrophages): "These monocytes were plated in tissue culture plates with 25% medium conditioned by recombinant mouse M-CSF (carrier-free, 20 ng/mL)" Why mouse M-CSF to differentiate human cells?

Response:

We apologize for the oversight. This was a typographical error. Recombinant human M-CSF was used to differentiate human macrophages. The text has been accordingly corrected in the revised manuscript.

- In the legend of Figure 1, Figure 3 and Figure 4, the authors state: "Data are presented as mean {plus minus} s.d. from three independent experiments." This statement is not correct for most of the graphs presented in those figures, where dots represent individual cells.

Response:

We thank you for this kind reminding. We apologize for the confusion caused by insufficient description in the figure legends. Dots representing individual cells were obtained from three independent experiments. We have revised the legends as "Data are presented as mean \pm s.d. calculated based on all cells from three independent experiments." in Fig. 1, 3, and 4 of the revised manuscript, respectively.

- line 1321 (Figure 3 legend): Were the macrophages analyzed in Figure 3h infected with *Salmonella*? This information should be stated in the legend.

Response:

We thank you for pointing out this omission. We confirmed that the macrophages analyzed in Fig. 3H were infected with *Salmonella*. We have now explicitly stated this in the revised figure legends.

- line 1371: "Schematic diagram shows that NF- κ B binds to ARHGEF3 promoter." That is incorrect, the diagram shows that there is a predicted NF- κ B binding site in the ARHGEF3 promoter.

Response:

We thank you for careful reading and constructive comment. We agree that our original wording overstated the experimental evidence, and we have revised the text to more accurately state: "predicted NF- κ B binding site in the

ARHGEF3 promoter.".

- There are 7 supplementary videos in the supplementary figure legends but only 6 files. The file for supplementary movie 6 fits the legend of supplementary video 7.

Response:

We sincerely apologize for this oversight. We have now verified and uploaded all the 7 supplementary videos corresponding to the legends in the revised manuscript.

2. writing style:

- Introduction is sometimes unnecessarily wordy.

Response:

We appreciate this helpful suggestion. We have carefully edited the Introduction to improve clarity and conciseness, while maintaining all key scientific points. The revised version removes redundant phrasing and sharpens the focus on our study's rationale and objectives.

- line 72: "morphogenetic remodeling paradigms"? Does the term "morphogenetic" really apply in this context? Maybe "morphological" instead?

Response:

We thank you for this suggestion. We have changed "morphogenetic" to "morphological" for more accurate description of cell shape changes.

- line 165: "and exposed it to macrophages" should be corrected to "and exposed macrophages to them"

Response:

Thank you. We have corrected the sentence to read: "and exposed macrophages to them" in the revised manuscript.

- line 151: "There results indicate that an increase in..."

Response:

We have removed the indicated sentence to streamline the text and enhance readability.

- line 207: "while excluded THP-1" should be corrected to "while excluding THP-1"

Response:

We agree with you. We have corrected the text to "while excluding THP-1" in the revised manuscript.

- line 342: "After a 24-hour recovery interval, single-cell colonies were established by sorting individual macrophages into 96-well plates, followed by

enrichment in the second round of LPS stimulation, colonies with significant variance in their capacity for DLPs production were identified as 19 P- colonies with minimal DLPs and 5 P+ colonies with robust DLPs formation." Clarity would be improved by breaking this sentence into two.

Response:

We appreciate your helpful suggestion to improve readability. We have revised this sentence by breaking it into two sentences: "After a 24-hour recovery interval, single-cell colonies were established by sorting individual macrophages into 96-well plates, followed by enrichment in the second round of LPS stimulation. Colonies with significant variance in their capacity for DLPs production were identified as 19 P- colonies with minimal DLPs and 5 P+ colonies with robust DLPs formation."

- line 381: "implicating" should be corrected to "suggesting that"

Response:

Thank you. We have revised the text as "suggesting that".

- line 400: "suggesting that NF- κ B is the key cause of LPS-TLR4 to boost ARHGEF3 expression" should be corrected to "suggesting that NF- κ B is the key factor through which LPS-TLR4 boosts ARHGEF3 expression"

Response:

We agree with you. We have revised the text as you suggested.

- line 415: "inhibition of RhoA signaling using Y-27632, an inhibitor of..."

Response:

Thank you. We have removed the text "an inhibitor of..." in the revised manuscript to make the explanation more concise and reader-friendly.

- line 789 (mice infection): Only male mice were analyzed for the re-infusion experiments, why?

Response:

We appreciate your question regarding our use of male mice. While *Salmonella* infection does not show gender preference in mice, we used male mice in these experiments to maintain consistency with our previous experimental conditions and to align with established protocols in the field (e.g., PMID: 35969050).

- line 794: What does "partially euthanized" mean?

Response:

We thank you for catching this imprecise wording. We have revised the text to clarify: "Mice for tissue and organ collection were anesthetized and euthanized at 48 hpi, while those for body weight measurements were euthanized at 72 hpi."

- line 894: "The following drugs were used at a specified doses and times"

Response:

Thank you. We have revised the text as "The following drugs were used" in the revised manuscript to make the text clearer and easier to read.

- line 943: "Primary and secondary antibodies were diluted in fresh blocking buffer for overnight at 4{degree sign}C and for 1 h at room temperature, respectively."

Response:

We thank you for this kind reminding. To clarity, we have modified the text as "Primary and secondary antibodies were diluted in fresh blocking buffer. Primary antibody incubated with PVDF membranes overnight at 4°C and secondary antibodies incubated with PVDF membrane for 1 h at room temperature, respectively." In the revised manuscript.

- line 949: "iBMDM monoclonals" should be replaced by "monoclonal iBMDMs"

Response:

We agree with you. We have corrected the text as "monoclonal iBMDMs" in the revised manuscript.

- Title of Fig. 1 should be: "Macrophages form DLPs in response to a bacterial challenge".

Response:

Thank you. We have corrected the title of Fig. 1 as "Macrophages form DLPs in response to a bacterial challenge" in the revised manuscript.

- line 1253: "from three separate sets of experimental mice" Do the authors mean from three experiments?

Response:

We thank you for catching this unclear wording. We have revised the text to state "from three experiments" for better clarity and precision in the revised manuscript.

- line 1299 (Figure 1 legend): "Schematic diagram illustrates that macrophages directional growth requires the contact with bacteria."

Response:

Thank you. We have revised the text as "Schematic diagram illustrates that the directional growth of DLPs require the contact with bacteria" in the revised manuscript.

- There are sentences written in the present tense in most figure legends, which are a bit unnatural to read (for example, "Representative images

visualize the membrane..."). These sentences should be corrected to use the gerund (for example, "Representative images showing the membrane...") or to avoid the use of a verb altogether.

Response:

We appreciate your helpful suggestion regarding figure legend style. We have systematically revised all figure legends to use gerund forms (e.g., "showing" instead of "visualize") for more natural scientific writing.

- line 1296: "membrane pore" should be plural.

Response:

Thank you. We have modified the "membrane pore" to plural form as "membrane pores" in the revised manuscript.

- lines 1313 and 1329: "stage" should be plural.

Response:

Thank you. We have modified "stage" to plural form as "stages" in the revised manuscript.

- line 1335: "TLPs" should be DLPs.

Response:

Thank you. We have revised this typo as "DLPs" in the revised manuscript.

- line 1370: "Western blot analysis of ARHGEF3 and the active levels of RhoA treated with LPS for indicated time in THP-1 macrophages" should be corrected to: "Western blot analysis of ARHGEF3 and the active levels of RhoA in THP-1 macrophages treated with LPS for indicated times".

Response:

Thank you. We have revised the text as you suggested in the revised manuscript.

- line 1378: "The diagram of RhoA-ROCK signaling pathway" should be corrected to: "Diagram of the RhoA-ROCK signaling pathway".

Response:

We agree with you. We have revised the text as "Diagram of the RhoA-ROCK signaling pathway" in the revised manuscript.

Referee #3:

This manuscript presents evidence for a previously undescribed response of macrophages to infection with Gram-negative bacteria. While many studies have reported that macrophages rapidly take up bacteria via "capture" by filopodia and subsequent engulfment in actin-rich phagocytic cups, here the authors describe a later, more prolonged response in which cells extend long, branched, relatively stable protrusions that they call dendrite-like pseudopods (DLPs). Unlike filopodia or lamellipodia, which are entirely actin-based, DLPs contain a core of vimentin which appears to enhance their stability. Mechanistically, the authors describe a signaling cascade in which the bacterial lipopolysaccharide (LPS) activates the Toll-like receptor TLR4, inducing expression of the RhoA activating protein ARHGEF3 in an NFkB-dependent manner, all of which is necessary for DLP formation. Physiologically, they show that peritoneal injection of DLP-proficient macrophages in mice enhances clearance of peritoneal *Salmonella*, relative to non-DLP proficient cells. Taken together, these observations describe a novel mechanism for the clearance of Gram-negative bacteria by macrophages. In general, the experiments are thorough and well controlled, the data are clear, statistically sound and support the conclusions drawn by the authors. I have only a few relatively minor concerns that should be addressed:

Response:

We sincerely thank you for the thorough evaluation and positive assessment of our work. We have carefully addressed all comments to improve methodological clarity, data presentation, and interpretation. These revisions have strengthened the manuscript while maintaining our core findings about this new bacterial ingestion pathway. We believe that the revised manuscript now makes a clearer contribution to our understanding of macrophage responses to infection.

1. It is unclear whether bacteria are present during the entire (typically 6h) incubation period or the authors use a standard gentamicin treatment assay in which cells are exposed to a pulse of bacteria then treated with gentamicin to kill remaining extracellular bacteria. The reason this is important is that the authors infect at an MOI of 20, but bacteria in the nutrient-rich medium would be expected to proliferate dramatically over a 6h incubation period.

Response:

We appreciate this important question. To clarify our protocol, macrophages were first infected with suspended *Salmonella* in serum-free, antibiotic-free medium for 30 min. The supernatant medium containing bacteria were then collected, centrifuged, and resuspended in a medium containing 10% FBS and 2 µg/mL gentamicin. This gentamicin concentration, which we validated in Figure R3-1 and in previous studies (PMID: 36717589), effectively prevented bacterial overgrowth while maintaining infectivity throughout the incubation

period. Cells were re-infected with these gentamicin-added bacteria. This time point was considered as 0 h of infection. We have added these methodological details to the revised manuscript to ensure full transparency about how we maintained consistent bacterial loads during our experiments.

Figure R3-1. (A) Bacterial survival rates in THP-1 macrophages after infection with *Salmonella* under varying gentamicin concentrations (0, 2, and 20 $\mu\text{g}/\text{mL}$). (B) Representative image of the CFU assay of *Salmonella* from THP-1 macrophages cultured in medium containing 2 $\mu\text{g}/\text{mL}$ gentamicin.

2. It should be stated clearly in both the abstract and the text that this is quite a late response to LPS exposure, much later than the formation of filopodia or lamellipodia that are much more rapid (within minutes) and are driven by Cdc42 or Rac1 respectively, as opposed to RhoA.

Response:

We thank you for this important suggestion. In response, we have carefully revised the text to emphasize that DLP formation represents a delayed response occurring hours after infection, in contrast to the rapid (minutes) formation of Cdc42/Rac1-dependent filopodia and lamellipodia.

Key modifications include: (1) In the abstract, we now describe DLPs as emerging "in a few hours upon severe Gram-negative bacterial infections"; (2) The introduction now contrasts the immediate actin-based protrusions with our newly described delayed response; (3) The discussion explicitly compares the transient nature of classical protrusions (minutes) with the prolonged stability of DLPs (hours). These changes better position our findings within the established timeline of macrophage responses while clearly delineating the novel aspects of DLPs biology. We appreciate this opportunity to improve the clarity and context of our work.

3. It is unclear why the authors segregated cells into responder vs. non-responder (DLP-producing vs. non-producing) clonal cell lines prior to RNA-seq analysis as opposed to simply treating cells with LPS or not. Why was this necessary?

Response:

We thank you for this kind suggestion to make our experimental design clearer. The decision to establish clonal DLP-producing (P+) and non-producing (P-) cell lines was driven by the observation that only about 30% of the wild-type population of cells was able to form DLPs upon infection. By creating these distinct populations through careful clonal selection and LPS stimulation (with P+ clones showing >40% type III morphology versus P- clones with <10%), we could more effectively identify molecular signatures specific to DLP formation without the dilution effect seen in bulk populations. This approach allowed us to identify ARHGEF3 as a key regulator through RNA sequencing comparisons

between matched P+ and P- clones with maximally divergent phenotypes (~52% vs. ~1% type III cells), which were then validated in independent clones.

The multiple rounds of LPS stimulation served to first confirm DLP-forming capacity in the parental population, then select and characterize clones, and finally trigger the response prior to molecular analysis. We have expanded the explanation of this experimental design in the revised manuscript to better communicate its rationale and advantages for identifying DLP-specific mechanisms that would be challenging to detect in heterogeneous wild-type populations.

4. A related point - the authors used these clonal cell lines for their *in vivo* infection model, injecting responders vs. non-responders into the peritoneum after *Salmonella* infection. While the data are consistent with the authors' interpretation that the ability to form DLPs enhances their ability to clear infection, there are literally hundreds of other genes that are differentially expressed that might also contribute to clearance. It would seem "cleaner" to use macrophages lacking ARHGEF3 (or not) to prove their point.

Response:

We thank you for this important suggestion. To directly test the specific role of ARHGEF3 in DLP formation, we performed additional *in vivo* re-infusion experiments using ARHGEF3-silenced P+ macrophages. As shown in Figure R3-2, ARHGEF3 knockdown in P+ cells reversed their enhanced bacterial control capacity, supporting that DLPs (rather than other differentially expressed genes) mediate this protective effect.

Figure R3-2. (A) Schematic diagram of extraintestinal infection mouse model incorporating macrophage re-infusion. (B-C) Quantification of bacterial load indicated by colony forming unit (CFU) in peritoneal lavage fluid (B) and liver (C) infected with *Salmonella*. (D-F) Quantitative RT-qPCR analysis of the expression levels of inflammatory markers in the spleen. Data are presented as mean \pm s.d.. ns $P > 0.05$; * $P < 0.05$; ** $P < 0.1$: unpaired two-tailed Student's *t*-test (B-F).

Dear Yaming,

Thank you for submitting a revised version of your manuscript. I sincerely apologise for the long delay in the processing of your manuscript due to conference travel and the resulting backlog, as well as due to pre-publication quality control checks.

We have now received input from two of the original reviewers. While reviewer #1 is satisfied with the revision, reviewer #3 finds that further clarification of several experimental aspects is needed, also in response to the initial comments by reviewer #2, who was not able to assess the revised version.

Based on this input, I would invite you to include responses and clarification to the comments by reviewer #3 in a final revision.

Additionally, there are a few editorial points that need addressing before I can extend official acceptance of the manuscript:

1. Please submit a complete author checklist, which you can download from our author guidelines (<https://www.embopress.org/pb-assets/embo-site/EMBO%20Press%20Author%20Checklist-1642513524327.xlsx>). Please insert information in the checklist that is also reflected in the manuscript. The completed author checklist will also be part of the Review Process File.
2. Please limit the number of keywords to maximally five.
3. Please check if the correct emails have been provided for all authors. Specifically, emails sent to Hao Zhang (zhanghao1@shanghaitech.edu.cn) and Jie Mei (jmei@siii.csa.cn) did not reach the recipients.
4. We are missing the ORCID iD for the co-corresponding author Tao Xu. In order to link the ORCID iD to the account in our manuscript tracking system, the author in question has to do the following:
 - Click the 'Modify Profile' link at the bottom of your homepage in our system.
 - On the next page you will see a box halfway down the page titled ORCID*. Below this box is red text reading 'To Register/Link to ORCID, click here'. Please follow that link: you will be taken to ORCID where you can log in to your account (or create an account if you don't have one)
 - You will then be asked to authorise Wiley to access your ORCID information. Once you have approved the linking, you will be brought back to our manuscript system.Unfortunately, we cannot do this linking on the author's behalf for security reasons.
5. Please ensure that the funding information is correct and identical both in the manuscript Acknowledgments section and our online system.
6. CRediT has replaced the traditional author contributions section because it offers a systematic, machine-readable author contributions format that allows for more effective research assessment. Please remove the Authors Contributions from the manuscript and use the free text boxes beneath each contributing author's name in our online submission system to add specific details on the author's contribution. More information is available in our guide to authors.
7. Please rename the movies into Movie EV1-EV7 and update the callouts accordingly. The legends should be removed from the manuscript text file and zipped with each movie file. Further information is available here: <https://www.embopress.org/page/journal/14602075/authorguide#expandedview>
8. Please remove the Reagents and Tools Table from the manuscript text file and upload it as a separate file choosing the file type "Reagent Table" and using the template available in our author guidelines: <https://www.embopress.org/page/journal/14602075/authorguide#structuredmethods>.
9. Please add the oligonucleotide sequences in the Reagents and Tools Table in the appropriate section.
10. Please move "Acknowledgements and Disclosure Statement" after Discussion and before References.
11. In the Data Availability section, please add a resolvable link to the RNA sequencing dataset. More information about the format of this section can be found here: <https://www.embopress.org/page/journal/14602075/authorguide#dataavailability>.
12. There is a reference to "data not shown" on page 25. According to our policy, which does not permit references to "data not shown", please include this information in the Appendix. Please see also <https://www.embopress.org/page/journal/14602075/authorguide#unpublisheddata>.
13. Please upload source data files as one zip folder per figure that includes both images and numerical data.
14. In our standard source data check, we have noted unexplained numerical duplications in the source data for a couple of figures: Fig 1I, 6J-M, EV1G, EV2B, EV3B, EV4W, EV5H-I. I have attached the corresponding file with the detected duplications labelled in colour. Please take a look and correct if needed. Please provide a brief explanation to nay changes. I appreciate that these duplications can also occur due to specific measurement or calculation methods used.
15. There are some mismatches between the source data and that shown in the figures:
 - figure 1D (mock 3D) source data contain a green signal that is not reflected in the figure.
 - Figure 3G, WB for Vimentin does not match the main figure.
 - Figure 4J - WB for TLR does not match the main figure.
 - 4M NF-kappaB blot does not match the main figure.
 - 5I GAPDH blot does not match the main figure.
 - 5K - ARHGEF3, GAPDH blots do not match the main figure.
 - 5P - full blots would be needed as source data, only small sections are submitted.Please check and correct as necessary, while providing an explanation to the changes.

16. Our data editors have flagged the following issues in figure legends that need correcting:

- Please provide the exact p values in the legends of figures 1 E, F, H, I, J, K, L, O; 2F, H, K; 3F, 4A, C, D, E, F, G, I, L, O; 5G, J, L, N, O, R; 6F, H, I, J, K, L, M, O, P; EV1 B, C, D, E, H, J, O; EV2 H, L; EV3 B, EV4 E, F, G, H, L; EV5 F, H, I, K, L, M, N, O.
- Please indicate the statistical test used for data analysis in the legends of figures 5C, E, G; 6O, EV4 A.
- Please define the red arrow heads in the legend of figure 2J.
- Please define the yellow arrow heads in the legend of figures 4D, EV4 T, U.
- Please define the white arrow heads in the legend of figure 6B.

17. Papers published in The EMBO Journal are accompanied online by a 'Synopsis' to enhance discoverability of the manuscript. It consists of A) a short (1-2 sentences) summary of the findings and their significance, B) 3-4 bullet points highlighting key results and C) a synopsis image that is 550x300-600 pixels large (width x height, jpeg or png format). You can either show a model or key data in the synopsis image. Please note that the image size is rather small and that text needs to be readable at the final size.

With kind regards,

leva

leva Gailite, PhD
Senior Scientific Editor
The EMBO Journal
Meyerhofstrasse 1
D-69117 Heidelberg
Tel: +4962218891309
i.gailite@embojournal.org

We realize that it is difficult to revise to a specific deadline. In the interest of protecting the conceptual advance provided by the work, we recommend a revision within 3 months (8th Sep 2025). Please discuss the revision progress ahead of this time with the editor if you require more time to complete the revisions.

Referee #1:

The authors have addressed all my comments. Congratulations for the findings.

Referee #3:

EMBOJ-2024-119940R, corr. author Prof. Jiu
"Macrophage-derived dendrite-like pseudopods enhance bacterial ingestion"

This manuscript presents evidence for a previously undescribed response of macrophages to infection with Gram-negative bacteria. While many studies have reported that macrophages rapidly take up bacteria via "capture" by filopodia and subsequent engulfment in actin-rich phagocytic cups, here the authors describe a later, more prolonged response in which cells extend long, branched, relatively stable protrusions that they call dendrite-like pseudopods (DLPs). Unlike filopodia or lamellipodia, which are entirely actin-based, DLPs contain a core of vimentin which appears to enhance their stability. Mechanistically, the authors describe a signaling cascade in which the bacterial lipopolysaccharide (LPS) activates the Toll-like receptor TLR4, inducing expression of the RhoA activating protein ARHGEF3 in an NFkB-dependent manner, all of which is necessary for DLP formation. Physiologically, they show that peritoneal injection of DLP-proficient macrophages in mice enhances clearance of peritoneal

Salmonella, relative to ARHGEF3-deficient (and DLP deficient) cells. Taken together, these observations describe a novel mechanism for the clearance of Gram-negative bacteria by macrophages.

In general, the authors have done a thorough job of addressing previous reviewer concerns. However, I have several questions arising from their response that remain unresolved:

1. In their vivo, two-photon imaging, mice were only infected for 20 min intraperitoneally prior to imaging. This is in direct contrast with the authors' statement that DLPs require hours to be induced. How do they reconcile this?
2. How were those bacteria cultured prior to intraperitoneal infection? If in log phase (as their methods section suggests) these are SPI-1 (invasion-inducing) conditions that would not be relevant for an intraperitoneal infection. Typically SPI-1 genes are downregulated during passage through the intestinal epithelium and would not be active in subepithelial tissues
3. One of the other reviewers questioned whether an moi of 20 is physiologically relevant. This may be true for intraperitoneal infection, but this is not the natural route of infection for Salmonella. Is there any evidence for such a high moi during oral infection?
4. Several questions about the in vitro infection model: 1) Bacteria are cultured under invasion-inducing conditions (log phase growth). Extensive evidence indicates that the T3SS1 translocon is cytotoxic to macrophages and kills them within 1h of infection. Is an moi of 20 sufficient to avoid this? At what moi do the bacteria induce cytotoxicity? 2) After an initial 30 min infection, bacteria are continuously present for several hours in the presence of 2 ug/ml gentamicin. The authors state that this low dose of gentamicin prevents bacterial overgrowth but that they remain "infective". What is the evidence for this?
5. The authors state that they infected macrophages with bacteria deficient in either the SPI1 or SPI2 machinery/effectors and that they saw no differences in the induction of DLPs. Were there differences in internalization efficiency? One might imagine that invasive bacteria are more effectively internalized than non-invasive ones. Similarly, the SPI2 effectors are necessary for intracellular survival, especially over a time course of 6 hours. Presumably there was a difference in the number of intracellular bacteria at this time point? Does the number of surviving bacteria matter to the induction of DLPs?

Referee #1:

The authors have addressed all my comments. Congratulations for the findings.

Response:

We thank you very much for your positive feedback on our manuscript.

Referee #3:

This manuscript presents evidence for a previously undescribed response of macrophages to infection with Gram-negative bacteria. While many studies have reported that macrophages rapidly take up bacteria via "capture" by filopodia and subsequent engulfment in actin-rich phagocytic cups, here the authors describe a later, more prolonged response in which cells extend long, branched, relatively stable protrusions that they call dendrite-like pseudopods (DLPs). Unlike filopodia or lamellipodia, which are entirely actin-based, DLPs contain a core of vimentin which appears to enhance their stability. Mechanistically, the authors describe a signaling cascade in which the bacterial lipopolysaccharide (LPS) activates the Toll-like receptor TLR4, inducing expression of the RhoA activating protein ARHGEF3 in an NFkB-dependent manner, all of which is necessary for DLP formation. Physiologically, they show that peritoneal injection of DLP-proficient macrophages in mice enhances clearance of peritoneal Salmonella, relative to ARHGEF3-deficient (and DLP deficient) cells. Taken together, these observations describe a novel mechanism for the clearance of Gram-negative bacteria by macrophages.

In general, the authors have done a thorough job of addressing previous reviewer concerns. However, I have several questions arising from their response that remain unresolved:

1. In their vivo, two-photon imaging, mice were only infected for 20 min intraperitoneally prior to imaging. This is in direct contrast with the authors' statement that DLPs require hours to be induced. How do they reconcile this?

Response:

As shown in Figure 1A (also Figure R1 in below), we started infection 2 hours prior to imaging (not 20 min), the labeling of macrophages with F4/80 was 20 min prior to imaging and mice were anesthetized 10 min prior to imaging.

Figure R1. (A) Schematic diagram of the two-photo intravital microscopy (TIM) experimental procedure.

2. How were those bacteria cultured prior to intraperitoneal infection? If in log phase (as their methods section suggests) these are SPI-1 (invasion-inducing) conditions that would not be relevant for an intraperitoneal infection. Typically SPI-1 genes are downregulated during passage through the intestinal epithelium and would not be active in subepithelial tissues

Response:

We sincerely thank you for your question. Peritoneal macrophages, which reside in the peritoneal cavity or differentiate from monocytes, adhere to the cavity wall (PMID: 33674464). Peritoneal *Salmonella* can directly contact these macrophages without crossing the intestinal epithelium. Thus, we believe peritoneal macrophage deformation in Figure 1B-E is directly driven by *Salmonella* infection.

3. One of the other reviewers questioned whether an moi of 20 is physiologically relevant. This may be true for intraperitoneal infection, but this is not the natural route of infection for *Salmonella*. Is there any evidence for such a high moi during oral infection?

Response:

Thank you for raising the question. When responding to Reviewer 1's question during revision, we referred data from an oral gavage mouse model. This parameter appears physiologically plausible: murine models of systemic salmonellosis (PMID: 25121751) demonstrate that infection with 5×10^7 CFU (bacterial load causing inflammation) of *Salmonella* leads to enterocyte microcolonies containing 10-20 bacteria per host cell within 12 hours (Figure R2). Therefore, we believe that high moi is achievable in the case of oral infection.

Figure R2. (A) Representative confocal micrographs of cecum sections from infected C57BL/6 mice 18h post-infection, showing S.Tm-G⁺ (green), host cell nuclei (gray), and EpCAM (epithelial cell marker; yellow). Lu.=Lumen. Right row image shows blow-ups of boxed regions. Scale bars, 10 μ m.

4. Several questions about the in vitro infection model: 1) Bacteria are cultured under invasion-inducing conditions (log phase growth). Extensive evidence indicates that the T3SS1 translocon is cytotoxic to macrophages and kills them within 1h of infection. Is an moi of 20 sufficient to avoid this? At what moi do the bacteria induce cytotoxicity? 2) After an initial 30 min infection, bacteria are continuously present for several hours in the presence of 2 ug/ml gentamicin. The authors state that this low dose of gentamicin prevents bacterial overgrowth but that they remain "infective". What is the evidence for this?

Response:

We sincerely thank you for your question. We agree with you and recognized that the T3SS1 channel is cytotoxic to macrophages (PMID: 10348855, in which the infection condition was **without antibiotics** and **MOI=25**). We thus measured the cell viability of THP-1 macrophages infected with *Salmonella* by LDH assay. Under MOI=20 infection with relatively low-concentration of gentamicin for 6h, cell viability did not decrease significantly, except when infected at MOI=100. This may be because that persistent low-concentration of gentamicin reduced bacterial protein synthesis, leading to lower bacterial toxicity. But it is speculated that a higher infection multiplicity and longer infection time would induce cytotoxicity (Figure R3-1).

Figure R3-1. Quantification of the relative cell viability without or with *Salmonella* infection for 6h (MOI=10/20/100). ns P > 0.05; *P < 0.05; one-way ANOVA with Sidak's analysis.

We tested bacterial growth in maintenance medium with varied concentrations of gentamicin. With 2 µg/mL gentamicin (referred as low concentration), bacterial growth was significantly inhibited compared to the control. With 20 µg/mL gentamicin, bacteria were completely killed (Figure R3-2A). After 2 µg/mL gentamicin-containing medium treatment, bacteria were collected and infected with macrophages. The CFU assay showed that the treated bacteria could still be found within macrophages (Figure R3-2A). We thus speculate that low-concentration gentamicin inhibits bacterial growth but could not eliminate infectivity.

Figure R3-2. (A) Bacterial survival rates in THP-1 macrophages after infection with *Salmonella* under varying gentamicin concentrations (0, 2, and 20 µg/mL). (B) Representative image of the CFU assay of *Salmonella* from THP-1 macrophages cultured in medium containing 2 µg/mL gentamicin.

5. The authors state that they infected macrophages with bacteria deficient in either the SPI1 or SPI2 machinery/ effectors and that they saw no differences in the induction of DLPs. Were there differences in internalization efficiency? One might imagine that invasive bacteria are more effectively internalized than non-invasive ones. Similarly, the SPI2 effectors are necessary for intracellular survival, especially over a time course of 6 hours. Presumably there was a difference in the number of intracellular bacteria at this time point? Does the number of surviving bacteria matter to the induction of DLPs?

Response:

We sincerely thank you for your question. we infected macrophages with bacteria lacking either SPI1 or SPI2 effectors and observed no apparent differences in DLPs formation. There are key points to consider:

For SPI1-deficient bacteria, the internalization efficiency into macrophages was significantly lower than that of the wild-type strain as reported (PMID: 33563986). This aligns with the expectation that invasive bacteria are more effectively internalized than non-invasive ones. Despite this difference in internalization, the induction of DLPs showed no significant variation between SPI1-deficient and wild-type bacteria. We speculate that the signaling transduction for DLPs formation is active and manage to perform the deformation.

For SPI2-deficient bacteria, as the reviewer pointed out, SPI2 effectors are crucial for intracellular survival, especially over a 6-hour time course. Our results confirmed that SPI2-deficient bacteria had a lower intracellular survival rate compared to the wild-type strain, consistent with previous reports (PMID: 33563986). However, similar to the SPI1-deficient case, this difference in intracellular bacterial numbers did not translate into a significant difference in DLPs induction. Our work confirms that bacterial LPS is the key factor causing DLP formation in macrophages via activating the signaling cascade we found. The T3SS is closely correlated with bacterial infection and replication, but it is not very much correlated to trigger DLPs formation.

Dear Yaming,

Thank you for addressing the final editorial issues. I am sorry to hear about the poor health of Dr. Xu. In this case, we will exceptionally proceed without an associated ORCID number in our system, as we cannot add it to his account on our side.

I am now pleased to inform you that your manuscript has been accepted for publication. Congratulations on a great study!

Finally, we would like to promote your manuscript among the Chinese readership. Therefore, we would like to invite you to prepare a short summary of the manuscript in Chinese (1500-2000 Chinese characters), which we will promote on the WeChat platform 'BioArt' with more than 610,000 followers.

If you are interested in this opportunity, we recommend covering the article very close to its online publication date. Thus, ideally we would very much appreciate if you could send us a draft within the next 7 working days. Please let us know whether or not you would be interested in contributing such a short summary in Chinese.

I have included below some general guidelines on how to prepare a summary and a link to recent examples for your reference. Please let me know if you have any questions about this.

If you have any questions, please do not hesitate to contact the Editorial Office. Thank you once more for your contribution to The EMBO Journal!

With best wishes,

Ieva

General Guidelines

1. These summary articles are meant to be targeting general audience so please limit the use of specialized technical terms, acronyms and jargon.
2. A summary usually starts with brief background information of the reported work, which is followed by explaining the findings in some detail, and ends with a short review of the conclusions as well as the implications of the work and future directions for the research.
3. The summary should contain a visual abstract, which can be the one provided in the paper.
4. Please provide ONE SINGLE document containing all text and graphical materials, ideally as a Word.docx or .doc file. Please DO NOT provide the document as a .pdf file.
5. Please DO NOT publicly release the document before the paper is officially published online.

Summary Examples

EMBO Journal | 灵珠与魔丸：昆虫miRNA调控病毒感染虫媒和植物的双重作用

EMBO Mol Med | 陈良/舒红兵合作揭示毛花甘C通过STUB1-FOXP3促进抗肿瘤免疫的机制

EMBO Rep | 王一飞/郑楷/刘凯胜团队揭示HDAC6调控cGAS-STING介导的抗病毒免疫的分子机制
